# Estimates of Late Cenozoic climate change relevant to Earth surface processes in tectonically active orogens

Sebastian G. Mutz[1], Todd A. Ehlers[1], Martin Werner[2], Gerrit Lohmann[2], Christian Stepanek[2], Jingmin Li[1,3]

[1] Department of Geosciences, University Tübingen, D-72074 Tübingen, Germany

[2] Department of Paleoclimate Dynamics, Alfred Wegener Institute, Helmholtz Centre for Polar and Marine Research, D-27570 Bremerhaven, Germany

[3] now at Institute for Geography and Geology, University of Würzburg, Würzburg, D-97074 Germany

*Correspondence to*: Sebastian G. Mutz (sebastian.mutz@uni-tuebingen.de)

**Abstract**

The denudation history of active orogens is often interpreted in the context of modern climate gradients. Here we address the validity of this approach and ask the question: what are the spatial and temporal variations in palaeoclimate for a latitudinally diverse range of active orogens? We do this using high-resolution (T159, ca. 80 x 80 km at the equator) palaeoclimate simulations from the ECHAM5 global Atmospheric General Circulation Model and a statistical cluster analysis of climate over different orogens (Andes, Himalaya, SE Alaska, Pacific NW USA). Time periods and boundary conditions considered include the Pliocene (PLIO, ~3 Ma), the Last Glacial Maximum (LGM, ~21 ka), Mid Holocene (MH, ~6 ka) and Pre-Industrial (PI, reference year 1850). The regional simulated climates of each orogen are described by means of cluster analyses based on the variability of precipitation, 2m air temperature, the intra-annual amplitude of these values, and monsoonal wind speeds where appropriate. Results indicate the largest differences to the PI climate existed for the LGM and PLIO climates in the form of widespread cooling and reduced precipitation in the LGM and warming and enhanced precipitation during the PLIO. The LGM climate shows the largest deviation in annual precipitation from the PI climate, and shows enhanced precipitation in the temperate Andes, and coastal regions for both SE Alaska and the US Pacific Northwest . Furthermore, LGM precipitation is reduced in the western Himalayas and enhanced in the eastern Himalayas, resulting in a shift of the wettest regional climates eastward along the orogen. The cluster-analysis results also suggest more climatic variability across latitudes east of the Andes in the PLIO climate than in other time-slice experiments conducted here. Taken together, these results highlight significant changes in Late Cenozoic regional climatology over the last ~3 Ma. Comparison of simulated climate with proxy-based reconstructions for the MH and LGM reveal satisfactory to good performance of the model in reproducing precipitation changes, although in some cases discrepancies between neighbouring proxy observations highlight contradictions between proxy observations themselves. Finally, we document regions where the largest magnitudes of Late Cenozoic

changes in precipitation and temperature occur and offer the highest potential for future observational studies that
quantify the impact of climate change on denudation and weathering rates.
**Keywords:** Cenozoic climate, ECHAM5, Last Glacial Maximum, Mid-Holocene, Pliocene, cluster analysis, Himalaya,
Tibet, Andes, Alaska, Cascadia

## 1. Introduction

Interpretation of orogen denudation histories in the context of climate and tectonic interactions is often hampered
by a paucity of terrestrial palaeoclimate proxy data needed to reconstruct spatial variations in palaeoclimate. While it is
self-evident that palaeoclimate changes could influence palaeodenudation rates, it is not always self-evident what the
magnitude of climate change over different geologic time scales is, or what geographic locations offer the greatest
potential to investigate palaeoclimate impacts on denudation. Palaeoclimate reconstructions are particularly beneficial
when denudation rates are determined using geo- and thermo-chronology techniques that integrate over timescales of
$10^3$-$10^{6+}$ years (e.g. cosmogenic radionuclides or low-temperature thermochronology) [e.g., Kirchner et al., 2001;
Schaller et al., 2002; Bookhagen et al., 2005; Moon et al., 2011; Thiede and Ehlers, 2013; Lease and Ehlers, 2013].
However, few studies using denudation rate determination methods that integrate over longer timescales have access to
information about past climate conditions that could influence these palaeo-denudation rates. Palaeoclimate modelling
offers an alternative approach to sparsely available proxy data for understanding the spatial and temporal variations in
precipitation and temperature in response to changes in orography [e.g. Takahashi and Battisti, 2007a, b; Insel et al.,
2010; Feng et al., 2013] and global climate change events [e.g. Salzmann, 2011; Jeffery et al., 2013]. In this study, we
characterise the climate at different times in the Late Cenozoic, and the magnitude of climate change for a range of
active orogens.  Our emphasis is on identifying changes in climate parameters relevant to weathering and catchment
denudation to illustrate the potential importance of various global climate change events on surface processes.
Previous studies of orogen-scale climate change provide insight into how different tectonic or global climate
change events influence regional climate change.  For example, sensitivity experiments demonstrated significant
changes in regional and global climate in response to landmass distribution and topography of the Andes, including
changes in moisture transport, the north-south asymmetry of the Intertropical Convergence Zone [e.g. Takahashi and
Battisti, 2007a, ; Insel et al., 2010] and (tropical) precipitation [Maroon et al., 2015, ; 2016]. Another example is the
regional and global climate changes induced by the Tibetan Plateau surface uplift due to its role as a physical obstacle to
circulation [Raymo and Ruddiman, 1992; Kutzbach et al., 1993; Thomas, 1997; Bohner, 2006; Molnar et al., 2010;
Boos and Kuang, 2010]. The role of tectonic uplift in long term regional and global climate change remains a focus of
research and continues to be assessed with geologic datasets [e.g. Dettman et al., 2003; Caves et al., 2017; Kent-
Corson et al., 2006; Lechler et al., 2013; Lechler and Niemi, 2011; Licht et al., 2016; Methner et al.,
2016; Mulch et al., 2015, 2008; Pingel et al., 2016] and climate modelling [e.g. Kutzbach et al., 1989; Kutzbach
et al., 1993; Zhisheng, 2001; Bohner, 2006; Takahashi and Battisti, 2007a; Ehlers and Poulsen, 2009; Insel et al., 2010;
Boos and Kuang, 2010]. Conversely, climate influences tectonic processes through erosion [e.g. Molnar and England,
1990; Whipple et al., 1999; Montgomery et al., 2001; Willett et al., 2006; Whipple, 2009]. Quaternary climate change
between glacial and interglacial conditions [e.g. Braconnot et al., 2007; Harrison et al., 2013] resulted in not only the
growth and decay of glaciers and glacial erosion [e.g. Yanites and Ehlers, 2012; Herman et al., 2013; Valla et al., 2011]
but also global changes in precipitation and temperature [e.g. Otto-Bliesner et al., 2006; Li et al., 2017] that could
influence catchment denudation in non-glaciated environments [e.g. Schaller and Ehlers, 2006; Glotzbach et al., 2013;
Marshall et al., 2015]. These dynamics highlight the importance of investigating how much climate has changed over
orogens that are the focus of studies of climate-tectonic interactions and their impact on erosion.
Despite recognition by previous studies that climate change events relevant to orogen denudation are prevalent
throughout the Late Cenozoic, few studies have critically evaluated how different climate change events may, or may
not, have affected the orogen climatology, weathering and erosion. Furthermore, recent controversy exists concerning
the spatial and temporal scales over which geologic and geochemical observations can record climate-driven changes in
weathering and erosion [e.g. Whipple, 2009; von Blanckenburg et al., 2015; Braun, 2016]. For example, the previous
studies highlight that although palaeoclimate impacts on denudation rates are evident in some regions and measurable
with some approaches, they are not always present (or detectable) and the spatial and temporal scale of climate change
influences our ability to record climate sensitive denudation histories. This study contributes to our understanding of
the interactions between climate, weathering, and erosion by bridging the gap between the palaeoclimatology and
surface processes communities by documenting the magnitude and distribution of climate change over tectonically
active orogens.
Motivated by the need to better understand climate impacts on Earth surface processes, especially the denudation
of orogens, we model palaeoclimate for four time slices in the Late Cenozoic, use descriptive statistics to identify the
extent of different regional climates, quantify changes in temperature and precipitation, and discuss the potential
impacts on fluvial and/or hillslope erosion. In this study, we employ the ECHAM5 global Atmospheric General
Circulation Model and document climate and climate change for time slices ranging between the Pliocene (PLIO, ~3
Ma) to pre-industrial (PI) times for the St. Elias Range of South East Alaska, the US Pacific Northwest (Olympic and
Cascade Range), western South America (Andes) and South Asia (incl. parts of Central- and East Asia). Our approach is
two-fold and includes:
1. An empirical characterisation of palaeo-climates in these regions based on the covariance and spatial
clustering of monthly precipitation and temperature, the monthly change in precipitation and temperature magnitude,
and wind speeds where appropriate.

2. Identification of changes in annual mean precipitation and temperature in selected regions for four time

periods: (PLIO,  Last Glacial Maximum (LGM), the Mid-Holocene (MH) and PI) and subsequent validation of the
simulated precipitation changes for MH and LGM.
Our focus is on documenting climate and climate change in different locations with the intent of informing past and
ongoing palaeodenudation studies of these regions. The results presented here also provide a means for future work to
formulate testable hypotheses and investigations into whether or not regions of large palaeoclimate change produced a
measurable signal in denudation rates or other Earth surface processes. More specifically, different aspects of the
simulated palaeoclimate may be used as boundary conditions for vegetation and landscape evolution models, such as
LPJ-GUESS and Landlab, to bridge the gap between climate change and quantitative estimates for Earth surface system
responses. In this study, we intentionally refrain from applying predicted palaeoclimate changes to predict denudation
rate changes. Such a prediction is beyond the scope of this study because a convincing (and meaningful) calculation of
climate-driven transients in fluvial erosion (e.g. via the kinematic wave equation), variations in frost cracking intensity,
or changes in hillslope sediment production and transport at the large regional scales considered here is not tractable
within a single manuscript, and instead is the focus of our ongoing work. Merited discussion of climatically induced
changes in glacial erosion, as is important in the Cenozoic, is also beyond the scope of this study. Instead, our emphasis
lies on providing and describing a consistently setup GCM simulation framework for future investigations of Earth
surface processes, and to identify regions in which Late Cenozoic climate changes potentially have a significant impact
on fluvial and hillslope erosion.

**2. Methods: Climate modelling and cluster analyses for climate characterisation**

**2.1 ECHAM5 simulations**

The global Atmospheric General Circulation Model ECHAM5 [Roeckner et al., 2003] has been developed at the

Max Planck Institute for Meteorology and is based on the spectral weather forecast model of the ECMWF [Simmons et
al., 1989]. In the context of palaeoclimate applications, the model has been used mostly at lower resolution (T31, ca.
3.75°x3.75°; T63, ca. 1.9°x1.9° in case of Feng et al. [2016] and T106 in the case of  Li et al. [2016] and Feng and
Poulsen [2016]) . The performed studies are not limited to the last millenium [e.g. Jungclaus et al., 2010] but also
include research in the field of both warmer and colder climates, at orbital [e.g. Gong et al., 2013; Lohmann et al., 2013;
Pfeiffer and Lohmann, 2016; Zhang et al., 2013a; Zhang et al., 2014; Wei and Lohmann, 2012] and tectonic time scales
[e.g. Knorr et al., 2011; Stepanek and Lohmann, 2012], and under anthropogenic influence [Gierz et al., 2015].
Here, the ECHAM5 simulations were conducted at a T159 spatial resolution (horizontal grid size ca. 80 km x 80
km at the equator) with 31 vertical levels (between the surface and 10hPa). This high model resolution is admittedly not
required for all of the climatological questions investigated in this study, and it should be noted that the skill of GCM's
in predicting orographic precipitation remains limited at this scale [e.g. Meehl et al. 2007].  However, simulations were
conducted at this resolution so that future work can apply the results in combination with different dynamical and
statistical downscaling methods to quantify changes at large catchment to orogen scales. The output frequency is
relatively high (1 day) to enhance the usefulness of our simulations as input for landscape evolution and other models
that may benefit from daily input. The simulations were conducted for five different time periods: present-day (PD), PI,
MH, LGM and PLIO.
A PD simulation (not shown here) was used to establish confidence in the model performance before conducting
palaeosimulations and has been compared with the following observation-based datasets: European Centre for Medium-
Range Weather Forecasts (ECMWF) re-analyses [ERA40, Uppala et al., 2005], National Centers for Environmental
Prediction and National Center for Atmospheric Research (NCEP/NCAR) re-analyses [Kalnay et al., 1996; Kistler et
al., 2001], NCEP Regional Reanalysis (NARR) [Mesinger et al., 2006], the Climate Research Unit (CRU) TS3.21
dataset [Harris et al., 2013], High Asia Refined Analysis (HAR30) [Maussion et al., 2014] and the University of
Delaware dataset (UDEL v3.01) [Legates et al., 1990]. (See Mutz et al. [2016] for a detailed comparison with a lower
resolution model).
The PI climate simulation is an ECHAM5 experiment with PI (reference year 1850) boundary conditions. Sea
Surface Temperatures (SST) and Sea Ice Concentration (SIC) are derived from transient coupled ocean-atmosphere
simulations [Lorenz and Lohmann, 2004; Dietrich et al., 2013].  Following Dietrich et al. [2013], greenhouse gas
(GHG) concentrations (CO2: 280 ppm) are taken from ice core based reconstructions of $CO_2$ [Etheridge et al., 1996],
$CH_4$ [Etheridge et al., 1998] and N2O [Sowers et al., 2003]. Sea surface boundary conditions for MH originate from a
transient, low-resolution, coupled atmosphere-ocean simulation of the mid (6 ka) Holocene [Wei and Lohmann, 2012;
Lohmann et al, 2013], where the GHG concentrations (CO2: 280 ppm) are taken from ice core reconstructions of
GHG's by Etheridge et al. [1996], Etheridge et al. [1998] and Sowers et al. [2003]. GHG's concentrations for the LGM
(CO2: 185 ppm) have been prescribed following Otto-Bliesner et al. [2006]. Orbital parameters for MH and LGM are
set according to Dietrich et al. [2013] and Otto-Bliesner et al. [2006], respectively. LGM land-sea distribution and ice
sheet extent and thickness are set based on the PMIP III (Palaeoclimate Modelling Intercomparison Project, phase 3)
guidelines (elaborated on by Abe-Ouchi et al [2015]). Following Schäfer-Neth and Paul [2003], SST and SIC for the
LGM are based on GLAMAP [Sarnthein et al. 2003] and CLIMAP [CLIMAP project members, 1981] reconstructions
for the  Atlantic and Pacific/Indian Ocean, respectively. Global MH and LGM vegetation are based on maps of plant
functional types by the BIOME 6000 / Palaeovegetation Mapping Project [Prentice et al., 2000; Harrison et al., 2001;
Bigelow et al., 2003; Pickett et al., 2004] and model predictions by Arnold et al. [2009]. Boundary conditions for the
PLIO simulation, including GHG concentrations ($CO_2$: 405), orbital parameters and surface conditions (SST, SIC, sea
land mask, topography and ice cover) are taken from the PRISM (Pliocene Research, Interpretation and Synoptic
Mapping) project [Haywood et al., 2010; Sohl et al., 2009; Dowsett et al., 2010], specifically PRISM3D. The PLIO
vegetation boundary condition was created by converting the PRISM vegetation reconstruction to the JSBACH plant
functional types as described by Stepanek and Lohmann [2012], but the built-in land surface scheme was used

SST reconstructions can be used as an interface between oceans and atmosphere [e.g. Li et al. 2016] instead of

conducting the computationally more expensive fully coupled Atmosphere-Ocean GCM experiments. While the use of
SST climatologies comes at the cost of capturing decadal-scale variability, and the results are ultimately biased towards
the SST reconstructions the model is forced with, the simulated climate more quickly reaches an equilibrium state and
the means of atmospheric variables used in this study do no change significantly after the relatively short spin-up
period.  The palaeoclimate simulations (PI, MH, LGM, PLIO) using ECHAM5 are therefore carried out for 17 model
years, of which the first two years are used for model spin up. The monthly long-term averages (multi-year means for
individual months) for precipitation, temperature, as well as precipitation and temperature amplitude, i.e. the mean
difference between the hottest and coldest months, have been calculated from the following 15 model years for the
analysis presented below.

For further comparison between the simulations, the investigated regions were subdivided (Fig. 1). Western

South America was subdivided into four regions: parts of tropical South America (80°-60° W, 23.5-5° S), temperate
South America (80°-60° W, 50°-23.5° S), tropical Andes (80°-60° W, 23.5-5° S; high-pass filtered), i.e. most of the
Peruvian Andes, Bolivian Andes and northernmost Chilean Andes, and temperate Andes (80°-60° W, 50°-23.5° S, high-
pass filtered). South Asia was subdivided into three regions: tropical South Asia (40°-120°E, 0°-23.5°N), temperate
South Asia (40°-120°E, 23.5°-60°N), and high altitude South Asia (40°-120°E, 0°-60°N; high-pass filtered).

Our approach of using a single GCM (ECHAM5) for our analysis is motivated by, and differs from, previous

studies where inter-model variability exists from the use of different GCMs due to different parameterisations in each
model. The variability in previous inter-model GCM comparisons exists despite the use of the same forcings [e.g. see
results highlighted in IPCC AR5]. Similarities identified between these palaeoclimate simulations conducted with
different GCMs using similar boundary conditions can establish confidence in the models when in agreement with
proxy reconstructions.  However, differences identified in inter-model GCM comparisons highlight biases by all or
specific GCMs, or reveal sensitivities to one changed parameter, such as model resolution. Given these limitations of
GCM modelling, we present in this study a comparison of a suite of ECHAM5 simulations to proxy-based
reconstructions (where possible) and, to a lesser degree, comment on general agreement or disagreement of our
ECHAM5 results with other modelling studies. A detailed inter-model comparison of our results with other GCMs is
beyond the scope of this study, and better suited for a different study in a journal with a different focus and audience.
Rather, by using the same GCM and identical resolution for the time slice experiments, we reduce the number of
parameters (or model parameterisations) varying between simulations and thereby remove potential sources of error or
uncertainty that would otherwise have to be considered when comparing output from different models with different
parameterisations of processes, model resolution, and in some cases model forcings (boundary conditions).
Nevertheless, the reader is advised to use these model results with the GCM's shortcoming and uncertainties in
boundary condition reconstructions in mind. For example, precipitation results may require dynamical or statistical
downscaling to increase accuracy where higher resolution precipitation fields are required. Furthermore, readers are
advised to familiarise themselves with the palaeogeography reconstruction initiatives and associated uncertainties. For
example, while Pliocene ice sheet volume can be estimated, big uncertainties pertaining to their locations remain
[Haywood et al. 2010].

**2.2 Cluster analysis to document temporal and spatial changes in climatology**
The aim of the clustering approach is to group climate model surface grid boxes together based on similarities in
climate. Cluster analyses are statistical tools that allow elements (i) to be grouped by similarities in the elements'
attributes. In this study, those elements are spatial units, the elements' attributes are values from different climatic
variables, and the measure of similarity is given by a statistical distance. The four basic variables used as climatic
attributes of these spatial elements are: near-surface (2m) air temperature, seasonal 2m air temperature amplitude,
precipitation rate, and seasonal precipitation rate amplitude. Since monsoonal winds are a dominant feature of the
climate in the South Asia region, near surface (10m) speeds of u-wind and v-wind (zonal and meridional wind
components, respectively) during the monsoon season (July) and outside the monsoon season (January) are included as
additional variables in our analysis of that region. Similarly, u-wind and v-wind speeds during (January) and outside
(July) the monsoon season in South America are added to the list of considered variables to take into account the South
American Monsoon System (SASM) in the cluster analysis for this region. The long-term monthly means of those
variables are used in a hierarchical clustering method, followed by a non-hierarchical k-means correction with
randomised re-groupment [Mutz et al., 2016; Wilks, 2011; Paeth, 2004; Bahrenberg et al., 1992].
The hierarchical part of the clustering procedure starts with as many clusters as there are elements (ni), then
iteratively combines the most similar clusters to form a new cluster using centroids for the linkage procedure for
clusters containing multiple elements. The procedure is continued until the desired number of clusters (k) is reached.
One disadvantage of a pure hierarchical approach is that elements cannot be re-categorised once they are assigned to a
cluster, even though the addition of new elements to existing clusters changes the clusters' defining attributes and could
warrant a re-categorisation of elements. We address this problem by implementation of a (non-hierarchical) k-means
clustering correction [e.g. Paeth, 2004]. Elements are re-categorizsed based on the multivariate centroids determined by
the hierarchical cluster analysis in order to minimisze the sum of deviations from the cluster centroids. The Mahalanobis
distance [e.g. Wilks, 2011] is used as a measure of similarity or distance between the cluster centroids, since it is a
statistical distance and thus not sensitive to different variable units. The Mahalanobis distance also accounts for possible
multi-collinearity between variables.

The end results of the cluster analyses are subdivisions of the climate in the investigated regions into $k$

subdomains or clusters based on multiple climate variables. The region-specific $k$ has to be prescribed before the
analyses. A large $k$ may result in redundant additional clusters describing very similar climates, thereby defeating the
purpose of the analysis to identify and describe the dominant, distinctly different climates in the region and their
geographical coverage. Since it is not possible to know a priori the ideal number of clusters, $k$ was varied between 3 and
10 for each region and the results presented below identify the optimal number of visibly distinctly different clusters
from the analysis. Optimal $k$ was determined by assessing the distinctiveness and similarities between the climate
clusters in the systematic process of increasing $k$ from 3 to 10. Once an increase in $k$ no longer resulted in the addition
of another cluster that was climatologically distinctly different from the others, and instead resulted in at least two
similar clusters, $k$ of the previous iteration was chosen as the optimal $k$ for the region.

The cluster analysis ultimately results in a description of the geographical extent of a climate (cluster)

characterised by a certain combination of mean values for each of the variables associated with the climate. For
example, climate cluster 1 may be the most tropical climate in a region and thus be characterised by a high precipitation
values, high temperature values and low seasonal temperature amplitude. Each of the results (consisting of the
geographical extent of climates and mean vectors describing the climate) can be viewed as an optimal classification for
the specific region and time. It serves primarily as a means for providing an overview of the climate in each of the
regions at different times, reduces dimensionality of the raw simulation output, and identify regions of climatic
homogeneity that is difficult to notice by viewing simple maps of each climate variable. Its synoptic purpose is similar
to that of the widely known Köppen-Geiger classification scheme [Peel et al., 2007], but we allow for optimal
classification rather than prescribe classes, and our selection of variables is more restricted and made in accordance with
the focus of this study.

**3. Results**

Results from our analysis are first presented for general changes in global temperature and precipitation for the

different time slices (Fig. 2, 3), which is then followed by an analysis of changes in the climatology of selected orogens.
A more detailed description of temperature and precipitation changes in our selected orogens is presented in subsequent
subsections (Fig. 4 and following). All differences in climatology are expressed relative to the PI control run. Changes
relative to the PI rather than PD conditions are presented to avoid interpreting an anthropogenic bias in the results and
focusing instead on pre-anthropogenic variations in climate. For brevity, near-surface (2m) air temperature and total
precipitation rate are referred to as temperature and precipitation.

**3.1 Global differences in mean annual temperature**
This section describes the differences between simulated MH, LGM, and PLIO annual mean temperature anom-
alies with respect to PI shown in Fig. 2b, and PI temperature absolute values shown in Fig. 2a. Most temperature differ-
ences between the PI and MH climate are within -1°C to 1°C. Exceptions to this are the Hudson Bay, Weddell Sea and
Ross Sea regions which experience warming of 1-3°C, 1-5°C and 1-9°C respectively. Continental warming is mostly re-
stricted to low-altitude South America, Finland, western Russia, the Arabian Peninsula (1-3°C) and subtropical north
Africa (1-5°C). Simulation results show that LGM and PLIO annual mean temperature deviate from the PI means the
most. The global PLIO warming and LGM cooling trends are mostly uniform in direction, but the magnitude varies re-
gionally. The strongest LGM cooling is concentrated in regions where the greatest change in ice extent occurs (as indic-
ated on Fig. 2), i.e. Canada, Greenland, the North Atlantic, Northern Europe and Antarctica. Central Alaska shows no
temperature changes, whereas coastal South Alaska experiences cooling of ≤ 9°C. Cooling in the US Pacific northwest
is uniform and between 11 and 13°C. Most of high-altitude South America experiences mild cooling of 1-3°C, 3-5°C in
the central Andes and ≤ 9°C in the south. Along the Himalayan orogen, LGM temperature values are 5-7°C below PI
values. Much of central Asia and the Tibetan plateau cools by 3-5°C, and most of India, low-altitude China and south-
east Asia by 1-3°C.
In the PLIO climate, parts of Antarctica, Greenland and the Greenland Sea experience the greatest temperature
increase (≤ 19°C). Most of southern Alaska warms by 1-5°C and ≤ 9°C near McCarthy, Alaska. The US Pacific northw-
est warms by 1-5°C. The strongest warming in South America is concentrated at the Pacific west coast and the Andes
(1-9°C), specifically between Lima and Chiclayo, and along the Chilean-Argentinian Andes south of Bolivia (≤ 9°C).
Parts of low-altitude South America to the immediate east of the Andes experience cooling of 1-5°C. The Himalayan
orogen warms by 3-9°C, whereas Myanmar, Bangladesh, Nepal, northern India and northeast Pakistan cool by 1-9°C.

**3.2 Global differences in mean annual precipitation**
Notable differences occur between simulated MH, LGM, PLIO annual mean precipitation anomalies with re-
spect to PI shown in Fig. 3b, and the PI precipitation absolute values shown in Fig. 3a. Of these, MH precipitation devi-
ates the least from PI values. The differences between MH and PI precipitation on land appear to be largest in northern
tropical Africa (increase ≤1200 mm/a) and along the Himalayan orogen (increase ≤2000 mm/a) and in central Indian
states (decrease) ≤500mm. The biggest differences in western South America are precipitation increases in central Chile
between Santiago and Puerto Montt. The LGM climate shows the largest deviation in annual precipitation from the PI
climate, and precipitation on land mostly decreases. Exceptions are increases in precipitation rates in North American
coastal regions, especially in coastal South Alaska ($\leq$2300 mm/a) and the US Pacific Northwest ($\leq$1700 mm/a). Further
exceptions are precipitation increases in low-altitude regions immediately east of the Peruvian Andes ($\leq$1800 mm/a),
central Bolivia ($\leq$1000 mm/a), most of Chile ($\leq$1000 mm/a) and northeast India ($\leq$1900 mm/a). Regions of notable pre-
cipitation decrease are northern Brazil ($\leq$1700 mm/a), southernmost Chile and Argentina ($\leq$1900 mm/a), coastal south
Peru ($\leq$700 mm/a), central India ($\leq$2300 mm/a) and Nepal ($\leq$1600 mm/a).
Most of the precipitation on land in the PLIO climate is higher than those in the PI climate. Precipitation is en-
hanced by ca. 100-200 mm/a in most of the Atacama desert, by $\leq$1700 mm/a south of the Himalayan orogen and by
$\leq$1400 mm/a in tropical South America. Precipitation significantly decreases in central Peru ($\leq$2600mm), southernmost
Chile ($\leq$2600mm) and from eastern Nepal to northernmost northeast India ($\leq$2500mm).

**3.3 Palaeoclimate characterisation from the cluster analysis and changes in regional climatology**
In addition to the above described global changes, the PLIO to PI regional climatology changes substantially in
the four investigated regions of: South Asia (section 3.3.1), the Andes (section 3.3.2), South Alaska (section 3.3.3) and
the Cascade Range (section 3.3.4). Each climate cluster defines separate distinct climate that is characterized by the
mean values of the different climate variables used in the analysis. The clusters are calculated by taking the arithmetic
means of all the values (climatic means) calculated for the grid boxes within each region. The regional climates are
referred to by their cluster number $C_1$, $C_2$, …, $C_k$, where $k$ is the number of clusters specified for the region. The clusters
for specific palaeoclimates are mentioned in the text as $C_{i(t)}$, where $i$ corresponds to the cluster number ($i$=1, …, $k$) and $t$
to the simulation time period ($t$=PI, MH, LGM, PLIO). The descriptions first highlight the similarities and then the
differences in regional climate. The cluster means of seasonal near-surface temperature amplitude and seasonal
precipitation amplitude are referred to as temperature and precipitation amplitude. The median, 25$^{th}$ percentile, 75$^{th}$
percentile, minimum and maximum values for annual mean precipitation are referred to as $P_{md}$, $P_{25}$, $P_{75}$, $P_{min}$ and $P_{max}$
respectively. Likewise, the same statistics for temperature are referred to as $T_{md}$, $T_{25}$, $T_{75}$, $T_{min}$ and $T_{max}$. These are
presented as boxplots of climate variables in different time periods. When the character of a climate cluster is described
as "high", "moderate" and "low", the climatic attribute's values are described relative to the value range of the specific
region in time, thus high PLIO precipitation rates may be higher than high LGM precipitation rates. The character is
presented a raster plots, to allow compact visual representation of it. The actual mean values for each variable in every
time-slice and region-specific cluster are included in tables in the supplementary material.

**3.3.1 Climate change and palaeoclimate characterisation in South Asia, Central- and East Asia**

This section describes the regional climatology of the four investigated Cenozoic time slices and how

precipitation and temperature changes from PLIO to PI times in tropical, temperate and high altitude regions. LGM and
PLIO simulations show the largest simulated temperature and precipitation deviations (Fig. 4b) from PI temperature and
precipitation (Fig. 4a) in the South Asia region. LGM temperatures are 1-7°C below PI temperatures and the direction
of deviation is uniform across the study region. PLIO temperature is mostly above PI temperatures by 1-7°C. The
cooling of 3-5°C in the region immediately south of the Himalayan orogen represents one of the few exceptions.
Deviations of MH precipitation from PI precipitation in the region are greatest along the eastern Himalayan orogeny,
which experiences an increase in precipitation (≤2000 mm/a). The same region experiences a notable decrease in
precipitation in the LGM simulation, which is consistent in direction with the prevailing precipitation trend on land
during the LGM. PLIO precipitation on land is typically higher than PI precipitation.

Annual means of precipitation and temperature spatially averaged for the regional subdivisions and the different

time slice simulations have been compared. The value range $P_{25}$ to $P_{75}$ of precipitation is higher for tropical South Asia
than for temperate and high altitude South Asia (Fig. 5 a-c). The LGM values for $P_{25}$, $P_{md}$ and $P_{75}$ are lower than for the
other time slice simulations, most visibly for tropical South Asia (ca. 100 mm/a). The temperature range (both $T_{75}$-$T_{25}$
and $T_{max}$- $T_{min}$) is smallest in the hot (ca. 21°C) tropical South Asia, wider in the high altitude (ca. -8°C) South Asia, and
widest in the temperate (ca. 2°C) South Asia region (Fig. 5 d-f). $T_{md}$, $T_{25}$ and $T_{75}$ values for the LGM are ca. 1°C, 1-2°C
and 2°C below PI and MH temperatures in tropical, temperate and high altitude South Asia respectively, whereas the
same temperature statistics for the PLIO simulation are ca. 1°C above PI and MH values in all regional subdivisions
(Fig. 5 d-f). With respect to PI and MH values, precipitation and temperature are generally lower in the LGM and higher
in the PLIO in tropical, temperate and high altitude South Asia.

In all time periods, the wettest climate cluster $C_1$ covers an area along the southeastern Himalayan orogen (Fig. 6

a-d) and is defined by the highest precipitation amplitude (dark blue, Fig. 6 e-h). $C_{5(PI)}$, $C_{3(MH)}$, $C_{4(LGM)}$ and $C_{5(PLIO)}$ are
characterized by (dark blue, Fig. 6e-h) the highest temperatures, u-wind and v-wind speeds during the summer monsoon
in their respective time periods, whereas $C_{4(PI)}$, $C_{5(MH)}$, and $C_{6(LGM)}$ are defined by low temperatures and highest
temperature amplitude, u-wind and v-wind speeds outside the monsoon season (in January) in their respective time
periods (Fig. 6 e-h). The latter 3 climate classes cover much of the more continental, northern landmass in their
respective time periods and represents a cooler climate affected more by seasonal temperature fluctuations (Fig. 6 a-d).
The two wettest climate clusters $C_1$ and $C_2$ are more restricted to the eastern end of the Himalayan orogen in the LGM
than during other times, indicating that the LGM precipitation distribution over the South Asia landmass is more
concentrated in this region than in other time slice experiments.

**3.3.2 Climate change and palaeoclimate characterisation in the Andes, Western South America**
This section describes the cluster analysis based regional climatology of the four investigated Late Cenozoic
time slices and illustrates how precipitation and temperature changes from PLIO to PI in tropical and temperate low-
and high altitude (i.e. Andes) regions in western South America (Fig. 7-9).
LGM and PLIO simulations show the largest simulated deviations (Fig. 7b) from PI temperature and
precipitation (Fig. 7a) in western South America. The direction of LGM temperature deviations from PI temperatures is
negative and uniform across the region. LGM temperatures are typically 1-3°C below PI temperatures across the region,
and 1-7°C below PI values in the Peruvian Andes, which also experience the strongest and most widespread increase in
precipitation during the LGM ($\leq$1800 mm/a). Other regions, such as much of the northern Andes and tropical South
America, experience a decrease of precipitation in the same experiment. PLIO temperature is mostly elevated above PI
temperatures by 1-5°C. The Peruvian Andes experience a decrease in precipitation ($\leq$2600mm), while the northern
Andes are wetter in the PLIO simulation compared to the PI control simulation.
PI, MH, LGM and PLIO precipitation and temperature means for regional subdivisions have been compared.
The $P_{25}$ to $P_{75}$ range is smallest for the relatively dry temperate Andes and largest for tropical South America and the
tropical Andes (Fig. 8 a-d). $P_{max}$ is lowest in the PLIO in all four regional subdivisions even though $P_{md}$, $P_{25}$ and $P_{75}$ in
the PLIO simulation are similar to the same statistics calculated for PI and MH time slices. $P_{md}$, $P_{25}$ and $P_{75}$ for the LGM
are ca. 50 mm/a lower in tropical South America and ca. 50 mm/a higher in the temperate Andes. Average PLIO
temperatures are slightly warmer and LGM temperatures are slightly colder than PI and MH temperatures in tropical
and temperate South America (Fig. 8 e and f). These differences are more pronounced in the Andes, however. $T_{md}$, $T_{25}$
and $T_{75}$ are ca. 5°C higher in the PLIO climate than in PI and MH climates in both temperate and tropical Andes,
whereas the same temperatures for the LGM are ca. 2-4°C below PI and MH values (Fig. 8 g and h).
For the LGM, the model computes drier-than-PI conditions in tropical South America and tropical Andes,
enhanced precipitation in the temperate Andes, and a decrease in temperature that is most pronounced in the Andes. For
the PLIO, the model predicts precipitation similar to PI, but with lower precipitation maxima. PLIO temperatures
generally increase from PI temperatures, and this increase is most pronounced in the Andes.
The climate variability in the region is described by six different clusters (Fig. 9 a-d), which have similar
attributes in all time periods. The wettest climate $C_1$ is also defined by moderate to high precipitation amplitudes, low
temperatures and moderate to high u-wind speeds in summer and winter in all time periods (dark blue, Fig. 9 e-h). $C_{2(PI)}$,
$C_{2(MH)}$, $C_{3(LGM)}$ and $C_{2(PLIO)}$ are characterized by high temperatures and low seasonal temperature amplitude (dark blue,
Fig. 9 e-h), geographically cover the north of the investigated region, and represent a more tropical climate. $C5_{(PI)}$,
$C5_{(MH)}$, $C6_{(LGM)}$ and $C6_{(PLIO)}$ are defined by low precipitation and precipitation amplitude, high temperature amplitude and
high u-wind speeds in winter (Fig. 9 e-h), cover the low-altitude south of the investigated region (Fig. 9 a-d) and
represent dry, extra-tropical climates with more pronounced seasonality. In the PLIO simulation, the lower-altitude east
of the region has four distinct climates, whereas the analysis for the other time slice experiments only yield three
distinct climates for the same region.

**3.3.3 Climate change and palaeoclimate characterisation in the St. Elias Range, Southeast Alaska**
This section describes the changes in climate and the results from the cluster analysis for South Alaska (Fig. 10-12). As
is the case for the other study areas, LGM and PLIO simulations show the largest simulated deviations (Fig. 10b) from
PI temperature and precipitation (Fig. 10a). The sign of LGM temperature deviations from PI temperatures is negative
and uniform across the region. LGM temperatures are typically 1-9°C below PI temperatures, with the east of the study
area experiencing largest cooling. PLIO temperatures are typically 1-5°C above PI temperatures and the warming is
uniform for the region. In comparison to the PI simulation, LGM precipitation is lower on land, but higher ($\leq$2300mm)
in much of the coastal regions of South Alaska. Annual PLIO precipitation is mostly higher ($\leq$800mm) than for PI.
$P_{md}$, $P_{25}$, $P_{75}$, $P_{min}$ and $P_{max}$ for South Alaskan mean annual precipitation do not differ much between PI, MH and
PLIO climates, while $P_{md}$, $P_{25}$, $P_{75}$ and $P_{min}$ decrease by ca. 20-40 mm/a and $P_{max}$ increases during the LGM (Fig. 11a).
The Alaskan PLIO climate is distinguished from the PI and MH climates by its higher (ca. 2°C) regional temperature
means, $T_{25}$, $T_{75}$ and $T_{md}$ (Fig. 11b). Mean annual temperatures, $T_{25}$, $T_{75}$, $T_{min}$ and $T_{max}$ are lower in the LGM than in any
other considered time period (Fig. 11b), and about 3-5°C lower than during the PI and MH.
Distinct climates are present in the PLIO to PI simulations for Southeast Alaska. Climate cluster $C_1$ is always
geographically restricted to coastal southeast Alaska (Fig. 12 a-d) and characterized by the highest precipitation,
precipitation amplitude, temperature, and by relatively low temperature amplitude (dark blue, Fig. 12 e-h). Climate $C_2$ is
characterized by moderate to low precipitation, precipitation amplitude, temperature, and by low temperature amplitude.
$C_2$ is either restricted to coastal southeast Alaska (in MH and LGM climates) or coastal southern Alaska (in PI and PLIO
climates). Climate $C_3$ is described by low precipitation, precipitation amplitude, temperature, and moderate temperature
amplitude in all simulations. It covers coastal western Alaska and separates climate $C_1$ and $C_2$ from the northern $C_4$
climate. Climate $C_4$ is distinguished by the highest mean temperature amplitude, by low temperature and precipitation
amplitude, and by lowest precipitation.
The geographical ranges of PI climates $C_1$- $C_4$ and PLIO climates $C_1$- $C_4$ are similar. $C_{1(PI/PLIO)}$ and $C_{2(PI/PLIO)}$ spread
over a larger area than $C_{1(MH/LGM)}$ and $C_{2(MH/LGM)}$. $C_{2(PI/PLIO)}$ are not restricted to coastal southeast Alaska, but also cover the
coastal southwest of Alaska. The main difference in characterization between PI and PLIO climates $C_1$- $C_4$ lies in the
greater difference (towards lower values) in precipitation, precipitation amplitude and temperature from $C_{1(PLIO)}$ to
$C_{2(PLIO)}$ compared to the relatively moderate decrease in those means from $C_{1(PI)}$ to $C_{2(PI)}$.

**3.3.4 Climate change and palaeoclimate characterisation in the Cascade Range, US Pacific Northwest**
This section describes the character of regional climatology in the US Pacific Northwest and its change over time
(Fig. 13-15). The region experiences cooling of typically 9-11°C on land during the LGM, and warming of 1-5°C
during the PLIO (Fig. 13b) when compared to PI temperatures (Fig. 13a). LGM precipitation increases over water,
decreases on land by ≤800 mm/a in the North and in the vicinity of Seattle and increases on land by ≤1400 mm/a on
Vancouver Island, around Portland and the Olympic Mountains, whereas PLIO precipitation does not deviate much
from PI values over water and varies in the direction of deviation on land. MH temperature and precipitation deviation
from PI values are negligible.
$P_{md}$, $P_{25}$, $P_{75}$, $P_{min}$ and $P_{max}$ for the Cascade Range do not notably differ between the four time periods (Fig. 14a).
The LGM range of precipitation values is slightly larger than that of the PI and MH with slightly increased $P_{md}$, while
the respective range is smaller for simulation PLIO. The $T_{md}$, $T_{25}$, $T_{75}$ and $T_{max}$ values for the PLIO climate are ca. 2°C
higher than those values for PI and MH (Fig. 14b). All temperature statistics for the LGM are notably (ca. 13°C) below
their analogues in the other time periods (Fig. 14b).
PI, LGM and PLIO clusters are similar in both their geographical patterns (Fig. 15 a, c, d) and their
characterization by mean values (Fig. 15 e, g, h). $C_1$ is the wettest cluster and shows the highest amplitude in
precipitation. The common characteristics of the $C_2$ cluster are moderate to high precipitation and precipitation
amplitude. $C_4$ is characterized by the lowest precipitation and precipitation amplitudes, and the highest temperature
amplitudes. Regions assigned to clusters $C_1$ and $C_2$ are in proximity to the coast, whereas $C_4$ is geographically restricted
to more continental settings.
In the PI and LGM climates, the wettest cluster $C_1$ is also characterized by high temperatures (Fig 10 e, g).
However, virtually no grid boxes were assigned to $C_{1(LGM)}$. $C_{1(MH)}$ differs from other climate state's $C_1$ clusters in that it is
also described by moderate to high near surface temperature and temperature amplitude (Fig 10 f), and in that it is
geographically less restricted and, covering much of Vancouver Island and the continental coastline north of it (Fig 10
b). Near surface temperatures are highest for $C_2$ in PI, LGM and PLIO climates (Fig 10 e, g, h) and low for $C_{2(MH)}$ (Fig
10 f). $C_{2(MH)}$ is also geographically more restricted than $C_2$ clusters in PI, LGM and PLIO climates (Fig 10 a-d). $C_{2(PI)}$,
$C_{2(MH)}$ and $C_{2(LGM)}$ have a low temperature amplitude (Fig 10 e-g), whereas $C_{2(PLIO)}$ is characterized by a moderate
temperature amplitude (Fig 10 h).

**4. Discussion**
In the following, we synthesise our results and compare to previous studies that investigate the effects of
temperature and precipitation change on erosion. Since our results do not warrant merited discussion of subglacial
processes without additional work that is beyond the scope of this study, we instead advise caution in interpreting the
presented precipitation and temperature results in an erosional context where the regions are covered with ice. For
convenience, ice cover is indicated on figures 2,3,47,10 and 13, and a summary of ice cover used as boundary
conditions for the different time slice experiments is included in the supplemental material. Where possible, we relate
the magnitude of climate change predicted in each geographical study area with terrestrial proxy data.

**4.1 Synthesis of temperature changes**

**4.1.1 Temperature changes and implications for weathering and erosion**

Changes in temperature can affect physical weathering due to temperature-induced changes in periglacial

processes and promote frost cracking and frost creep [e.g., Matsuoka, 2001; Schaller et al., 2002; Matsuoka and
Murton, 2008; Delunel et al., 2010; Andersen et al., 2015; Marshall et al., 2015], and also biotic weathering and erosion
[e.g. Moulton et al., 1998; Banfield et al., 1999; Dietrich and Perron, 2006]. Quantifying and understanding past
changes in temperature is thus vital for our understanding of denudation histories. In the following, we highlight regions
in the world where future observational studies might be able to document significant warming or cooling that would
influence temperature related changes in physical and chemical weathering over the last ~3 Ma.

Simulated MH temperatures show little deviation (typically < 1°C) from PI temperatures in the investigated

regions (Fig. 2b), suggesting little difference in MH temperature-related weathering. The LGM experiences widespread
cooling, which is accentuated at the poles. , increasin the equator-to-pole pressure gradient and consequently
strengthens global atmospheric circulation. Despite this global trend, cooling in coastal South Alaska is higher (≤ 9°C)
than in central Alaska (0±1°C). The larger temperature difference in South Alaska geographically coincides with ice
cover (Fig. 10b), and should thus be interpreted in context of a different erosional regime. Cooling in most of the lower-
latitude regions in South America and central to southeast Asia is relatively mild. The greatest temperature differences
in South America are observed for western Patagonia, which was mostly covered by glaciers. The Tibetan plateau
experiences more cooling (3-5°C) than adjacent low-altitude regions (1-3°C) during the LGM.

The PLIO simulation is generally warmer, and temperature differences are  accentuated warming at the poles.

Warming in simulation PLIO is greatest in parts of Canada,  Greenland and Antarctica (up to 19°C), which
geographically coincides with the presence of ice in the PI reference simulation and thus may be attributed to
differences in ice cover. It should therefore also be regarded as areas in which process domain shifted from glacial to
non-glacial.  The warming in simulation PLIO in South Alaska and the US Pacific northwest is mostly uniform and in
the range of 1-5°C. As before, changes in ice cover reveal that the greatest warming may be associated with the absence
of glaciers relative to the PI simulation. Warming in South America is concentrated at the Pacific west coast and the
Andes between Lima and Chiclayo, and along the Chilean-Argentinian Andes south of Bolivia (≤ 9°C).

Overall, annual mean temperatures in the MH simulation show little deviation from PI values.  The more

significant temperature deviations of the colder LGM and of the warmer PLIO simulations are accentuated at the poles
leading to higher and lower equator-to-pole temperature gradients respectively. The largest temperature-related changes
(relative to PI conditions) in weathering and subsequent erosion, in many cases through a shift in the process domain
from glacial to non-glacial or vice versa, are therefore to be expected in the LGM and PLIO climates.

**4.1.2 Temperature comparison to other studies**
LGM cooling is accentuated at the poles, thus increases the equator-to-pole pressure gradient and consequently
strengthens global atmospheric circulation, and is in general agreement with studies such as Otto-Bliesner et al. [2006]
and Braconnot et al. [2007]. The PLIO simulation shows little to no warming in the tropics and accentuated warming at
the poles, as do findings of Salzmann et al. [2011] and Robinson [2009] and Ballantyne [2010] respectively. This would
reduce the equator-to-pole sea and land surface temperature gradient, as also reported by Dowsett et al. [2010], and also
weaken global atmospheric circulation. Agreement with proxy-based reconstructions, as is the case of the relatively
little warming in lower latitudes, is not surprising given that sea surface temperature reconstructions (derived from
previous coarse resolution coupled ocean-atmosphere models) are prescribed in this uncoupled atmosphere simulation.
It should be noted that coupled ocean-atmosphere simulations do predict more low-latitude warming [e.g. Stepanek and
Lohmann 2012; Zhang et al. 2013b]. The PLIO warming in parts of Canada and Greenland (up to 19°C) and consistent
with values based on multi-proxy studies [Ballantyne et al., 2010]. Due to a scarcity of palaeobotanical proxies in
Antarctica, reconstruction-based temperature and ice-sheet extent estimates for a PLIO climate have high uncertainties
[Salzmann et al., 2011], making model validation difficult. Furthermore, controversy about relatively little warming in
the south polar regions compared to the north polar regions remains [e.g. Hillenbrand and Fütterer, 2002; Wilson et al.,
2002]. Mid-latitude PLIO warming is mostly in the 1-3°C range with notable exceptions of cooling in the northern
tropics of Africa and on the Indian subcontinent, especially south of the Himalayan orogen.

**4.2 Synthesis of precipitation changes**

**4.2.1 Precipitation and implications for weathering and erosion**
Changes in precipitation affects erosion through river incision, sediment transport, and erosion due to extreme
precipitation events and storms [e.g. Whipple and Tucker, 1999; Hobley et al., 2010]. Furthermore, vegetation type and
cover also co-evolve with variations in precipitation and with changes in geomorphology [e.g. Marston 2010; Roering
et al., 2010]. These vegetation changes in turn modify hillslope erosion by increasing root mass and canopy cover, and
decreasing water-induced erosion via surface runoff [e.g. Gyssels et al., 2005]. Therefore, understanding and
quantifying changes in precipitation in different palaeoclimates is necessary for a more complete reconstruction of
orogen denudation histories. A synthesis of predicted precipitation changes is provided below, and highlights regions
where changes in river discharge and hillslope processes might be impacted by climate change over the last ~3 Ma.
Most of North Africa is notably wetter during the MH, which is characteristic of the African Humid Period
[Sarnthein 1978]. This pluvial regional expression of the Holocene Climatic Optimum is attributed to sudden changes in
the strength of the African monsoon caused by orbital-induced changes in summer insolation [e.g. deMenocal et al.
2000]. Southern Africa is characterised by a wetter climate to the east and drier climate to the west of the approximate
location of the Congo Air Boundary (CAB), the migration of which has previously been cited as a cause for
precipitation changes in East Africa [e.g. Juninger et al. 2014]. In contrast, simulated MH precipitation rates show little
deviation from the PI in most of the investigated regions, suggesting little difference in MH precipitation-related
erosion. The Himalayan orogen is an exception and shows a precipitation increase of up to 2000 mm/a. The climate's
enhanced erosion potential, that could result from such a climatic change, should be taken into consideration when
palaeo-erosion rates estimated from the geological record in this area are interpreted [e.g. Bookhagen et al., 2005].
Specifically, higher precipitation rates (along with differences in other rainfall-event parameters) could increase the
probability of mass movement events on hillslopes, especially where hillslopes are close to the angle of failure [e.g.
Montgomery, 2001], and modify fluxes to increase shear stresses exerted on river beds and increase stream capacity to
enhance erosion on river beds (e.g. by abrasion).
Most precipitation on land is decreased during the LGM due to large-scale cooling and decreased evaporation
over the tropics, resulting in an overall decrease in inland moisture transport [e.g., Braconnot et al. 2007]. North
America, south of the continental ice sheets, is an exception and experiences increases in precipitation. For example, the
investigated US Pacific Northwest and the southeastern coast of Alaska experience experience strongly enhanced
precipitation of ≤1700 mm/a and ≤2300 mm/a, respectively. These changes geographically coincide with differences in
ice extent. An increase in precipitation in these regions may have had direct consequences on the glaciers' mass balance
and equilibrium line altitudes, where the glaciers' effectiveness in erosion is highest [e.g. Egholm et al., 2009; Yanites
and Ehlers, 2012]. The differences in the direction of precipitation changes, and accompanying changes in ice cover
would likely result in more regionally differentiated variations in precipitation-specific erosional processes in the St.
Elias Range rather than causing systematic offsets for the LGM. Although precipitation is significantly reduced along
much of the Himalayan orogen (≤1600 mm/a), , northeast India experiences strongly enhanced precipitation (≤1900
mm/a). This could have large implications for studies of uplift and erosion at orogen syntaxes, where highly localized
and extreme denudation has been documented [e.g. Koons et al., 2013; Bendick and Ehlers, 2014].
Overall, the PLIO climate is wetter than the PI climate, in particular in the (northern) mid-latitudes, and possibly
related to a northward shift of the northern Hadley cell boundary that is ultimately the result of a reduced equator-to-
pole temperature gradient [e.g. Haywood et al. 2000, 2013; Dowsett et al. 2010]. Most of the PLIO precipitation over
land increases , esp. at the Himalayan orogen by ≤1400 mm/a, and decreases from eastern Nepal to Namcha Barwa
(≤2500 mm/a). Most of the Atacama Desert experiences an increase in precipitation by 100-200 mm/a, which may have
to be considered in erosion and uplift history reconstructions for the Andes. A significant increase (~2000 mm/a) in
precipitation from simulation PLIO to modern conditions is simulated for the eastern margin of the Andean Plateau in
Peru and for northern Bolivia. This is consistent with recent findings of a pulse of canyon incision in these locations in
the last ~3 Ma [Lease and Ehlers, 2013].

Overall, the simulated MH precipitation varies least from PI precipitation. The LGM is generally drier than the

PI simulation, even though pockets of a wetter-than-PI climate do exist, such as much of coastal North America. Extra-
tropical increased precipitation of the PLIO simulation and decreased precipitation of the LGM climate may be the
result of decreased and increased equator-to-pole temperature gradients, respectively.

**4.2.2 Precipitation comparison to other studies**

The large scale LGM precipitation decrease on land, related to cooling and decreased evaporation over the

tropics, and greatly reduced precipitation along much of the Himalayan orogeny, is consistent with previous studies by,
(for example) Braconnot et al. [2007]. The large scale PLIO precipitation increase due to a reduced equator-to-pole
temperature gradient, has previously been pointed out by e.g. Haywood et al. [2000, 2013] and Dowsett et al. [2010]. A
reduction of this gradient by ca. 5°C is indeed present in the PLIO simulation of this study (Fig. 2b). This precipitation
increase over land agrees well with simulations performed at a lower spatial model resolution [cf. Stepanek and
Lohmann, 2012]. Section 4.4 includes a more in-depth discussion of how simulated MH and LGM precipitation
differences compare with proxy-based reconstructions in South Asia and South America.

**4.3 Trends in Late Cenozoic changes in regional climatology**

This section describes the major changes in regional climatology and highlights their possible implications on

erosion rates.

*Himalaya-Tibet, South Asia*

In South Asia, cluster-analysis based categorization and description of climates (Fig. 6) remains similar

throughout time. However, the two wettest climates ($C_1$ and C2) are geographically more restricted to the eastern
Himalayan orogen in the LGM simulation. Even though precipitation over the South Asia region is generally lower, this
shift indicates that rainfall on land is more concentrated in this region and that the westward drying gradient along the
orogen is more accentuated than during other time periods investigated here. While there is limited confidence in the
global Atmospheric General Circulation Model's abilities to accurately represent meso-scale precipitation patterns [e.g.
Cohen 1990], the simulation warrants careful consideration of possible, geographically non-uniform offsets in
precipitation in investigations of denudation and uplift histories.

MH precipitation and temperature in tropical, temperate and high-altitude South Asia is similar to PI

precipitation and temperature, whereas LGM precipitation and temperatures are generally lower (by ca. 100 mm/a and
1-2°C respectively), possibly reducing precipitation-driven erosion and enhancing frost-driven erosion in areas pushed
into a near-zero temperature range during the LGM.

*Andes, South America*

Clusters in South America (Fig. 9), which are somewhat reminiscent of the Köppen and Geiger classification

[Kraus, 2001], remain mostly the same over the last 3 Ma. In the PLIO simulation, the lower-altitude east of the region
is characterized by four distinct climates, which suggests enhanced latitudinal variability in the PLIO climate compared
to PI with respect temperature and precipitation.

The largest temperature deviations from PI values are derived for the PLIO simulation in the (tropical and

temperate) Andes, where temperatures exceed PI values by 5°C. On the other hand, LGM temperatures in the Andes are
ca. 2-4°C below PI values in the same region (Fig 7 g and h). In the LGM simulation, tropical South America
experiences ca. 50 mm/a less precipitation, the temperate Andes receive ca. 50 mm/a more precipitation than in PI and
MH simulations. These latitude-specific differences in precipitation changes ought to be considered in attempts to
reconstruct precipitation-specific palaeoerosion rates in the Andes on top of longitudinal climate gradients highlighted
by, e.g., Montgomery et al. [2001].

*St. Elias Range, South Alaska*

South Alaska is subdivided into two wetter and warmer clusters in the south, and two drier, colder clusters in the

north. The latter are characterised by increased seasonal temperature variability due to being located at higher latitudes
(Fig. 12). The different equator-to-pole temperature gradients for LGM and PLIO may affect the intensity of the Pacific
North American Teleconnection (PNA) [Barnston and Livzey, 1987], which has significant influence on temperatures
and precipitation, especially in southeast Alaska, and may in turn result in changes in regional precipitation and
temperature patterns and thus on glacier mass balance. Changes in the Pacific Decadal Oscillation, which is related to
the PNA pattern, has previously been connected to differences in Late Holocene precipitation [Barron and Anderson,
2011]. While this climate cluster pattern appears to be a robust feature for the considered climate states, and hence over
the recent geologic history, the LGM sets itself apart from PI and MH climates by generally lower precipitation (20-40
mm) and lower temperatures (3-5°C, Fig. 10, 11), which may favour frost driven weathering during glacial climate
states [e.g. Andersen et al., 2015; Marshall et al. 2015] in unglaciated areas, whereas glacial processes would have
dominated most of this region as it was covered by ice. Simulation PLIO is distinguished by temperatures that exceed
PI and MH conditions by ca. 2°C, and by larger temperature and precipitation value ranges, possibly modifying
temperature- and precipitation-dependent erosional processes in the region of South Alaska.

*Cascade Range, US Pacific Northwest*
In all time slices, the geographic climate patterns, based on the cluster analysis (Fig. 15), represents an increase
in the degree of continentality from the wetter coastal climates to the further inland located climates with greater
seasonal temperature amplitude and lower precipitation and precipitation amplitude (Fig 15 e-h). The most notable
difference between the time slices is the strong cooling during the LGM, when temperatures are ca. 13°C (Fig. 13, 14)
below those of other time periods. Given that the entire investigated region was covered by ice (Fig 13), we can assume
a shift to glacially dominated processes.

**4.4  Comparison of simulated and observed precipitation differences**
The predicted precipitation differences reported in this study were compared with observed (proxy record)
palaeoprecipitation change.  Proxy based precipitation reconstructions for the MH and LGM  are presented for South
Asia and South America for the purpose of assessing ECHAM5 model performance, and for identifying inconsistencies
between neighbouring proxy data. Due to the repeated glaciations, detailed terrestrial proxy records for the time slices
investigated here are not available, to the best of our knowledge, for the Alaskan and Pacific NW USA studies.
Although marine records and records of glacier extent are available in these regions, the results from them do not
explicitly provide estimates of wetter/drier, or colder/warmer conditions that can be spatially compared to the
simulation estimates.  For these two areas with no available records, the ECHAM5 predicted results therefore provide
predictions from which future studies can formulate testable hypotheses to evaluate.
The palaeoclimate changes in terrestrial proxy records compiled here are reported as "wetter than today", "drier
than today" or "the same as today" for each of the study locations, and plotted on top of the simulation-based difference
maps as upward facing blue triangles, downward facing red triangles and grey circles respectively (Fig. 16, 17). The
numbers listed next to those indicators are the ID numbers assigned to the studies compiled for this comparison and are
associated with a citation provided in the figure captions.
In South Asia, 14/26 results from local studies agree with the model predicted precipitation changes for the MH.
The model seems able to reproduce the predominantly wetter conditions on much of the Tibetan plateau, but predicts
slightly drier conditions north of Chengdu, which is not reflected in local reconstructions. The modest mismatch
between ECHAM5 predicted and proxy-based MH climate change in south Asia was also documented by Li et al.,
[2017], whose simulations were conducted at a coarser (T106) resolution. Despite these model-proxy differences, we
note that there are significant discrepancies between the proxy data themselves in neighbouring locations in the MH,
highlighting caution in relying solely upon these data for regional palaeoclimate reconstructions. These differences
could result from either poor age-constraints in the reported values, or systematic errors in the transfer functions used to
convert proxy measurements to palaeoclimate conditions.  The widespread drier conditions on the Tibetan Plateau and
immediately north of Laos are confirmed by 7/7 of the palaeoprecipitation reconstructions. 23/39 of the reconstructed
precipitation changes agree with model predictions for South America during the MH. The model predicted wetter
conditions in the central Atacama desert, as well as the drier conditions northwest of Santiago are confirmed by most of
the reconstructions. The wetter conditions in southernmost Peru and the border to Bolivia and Chile cannot be
confirmed by local studies. 11/17 of the precipitation reconstructions for the LGM are in agreement with model
predictions. These include wetter conditions in most of Chile. The most notable disagreement can be seen in northeast
Chile at the border to Argentina and Bolivia, where model predicted wetter conditions are not confirmed by reported
reconstructions from local sites.
Model performance is, in general, higher for the LGM than for the MH and overall satisfactory given that it
cannot be expected to resolve sub-grid scale differences in reported palaeoprecipitation reconstructions. However, as
mentioned above, it should be noted that some locations (MH of south Asia, and MH of norther Chile) discrepancies
exist between neighbouring proxy samples and highlight the need for caution in how these data are interpreted. Other
potential sources of error resulting in disagreement of simulated and proxy-based precipitation estimates are the model's
shortcomings in simulating orographic precipitation at higher resolutions, and uncertainties in palaeoclimate
reconstructions at the local sites. In summary, although some differences are evident in both the model-proxy data
comparison and between neighbouring proxy data themselves, the above comparison highlights an overall good
agreement between the model and data for the south Asia and South American study areas. Thus, although future
advances in GCM model parameterisations and new or improved palaeoclimate proxy techniques are likely, the
palaeoclimate changes documented here are found to be in general robust and provide a useful framework for future
studies investigating how these predicted changes in palaeoclimate impact denudation.

**4.5 Conclusions**
We present a statistical cluster-analysis-based description of the geographic coverage of possible distinct
regional expressions of climates from four different time slices (Fig. 6, 9, 12, 15). These are determined with respect to
a selection of variables that characterize the climate of the region and may be relevant to weathering and erosional
processes. While the geographic distribution of climate remains similar throughout time (as indicated by results of four
different climate states representative for the climate of the last 3 Ma), results for the PLIO simulation suggests more
climatic variability east of the Andes (with respect to near-surface temperature, seasonal temperature amplitude,
precipitation, seasonal precipitation amplitude and seasonal u-wind and v-wind speeds). Furthermore, the wetter
climates in the South Asia region retreat eastward along the Himalayan orogen for the LGM simulation, this is due to
decreased precipitation along the western part of the orogen and enhanced precipitation on the eastern end, possibly
signifying more localised high erosion rates.

Most global trends of the high-resolution LGM and PLIO simulations conducted here are in general agreement

with previous studies [Otto-Bliesner et al., 2006; Braconnot et al., 2007; Wei and Lohmann, 2012; Lohmann et al.,
2013; Zhang et al., 2013b, 2014; Stepanek and Lohmann, 2012]. The MH does not deviate notably from the PI, the
LGM is relatively dry and cool, while the PLIO is comparably wet and warm. While the simulated regional changes in
temperature and precipitation usually agree with the sign (or direction) of the simulated global changes, there are
region-specific differences in the magnitude and direction. For example, the LGM precipitation of the Tropical Andes
does not deviate significantly from PI precipitation, whereas LGM precipitation in the Temperate Andes is enhanced.

Comparisons to local, proxy-based reconstructions of MH and LGM precipitation in South Asia and South

America reveal satisfactory performance of the model in simulating the reported differences. The model performs better
for the LGM than the MH. We note however that compilations of proxy data such as we present here, also identify
inconsistences between neighbouring proxy data themselves, warranting caution in the extent to which both proxy data
and palaeoclimate models are interpreted for MH climate change in south Asia, and western South America.

The changes in regional climatology presented here are manifested, in part, by small to large magnitude changes

in fluvial and hillslope relevant parameters such as precipitation and temperature. For the regions investigated here we
find that precipitation differences between the PI, MH, LGM, and PLIO are in many areas around +/- 200-600 mm/yr,
and locally can reach maximums of +/- 1000-2000 mm/yr (Figs. 4, 7, 10, 13). In areas where significant precipitation
increases are accompanied by changes in ice extent, such as parts of southern Alaska during the LGM, we would expect
a shift in the erosional regime to glacier dominated processes. Temperature differences between these same time periods
are around 1-4 °C in many places, but reach maximum values of 8-10 °C. Many of these maxima in the temperature
differences geographically coincide with changes in ice sheet extent and must therefore be interpreted as part of a
different erosional process domains. However, we also observe large temperature differences (~5°C) in unglaciated
areas that would be affected by hillslope, frost cracking, and fluvial processes. The magnitude of these differences are
not trival, and will likely impact fluvial and hillslope erosion and sediment transport, as well as biotic and abiotic
weathering. The regions of large magnitude changes in precipitation and temperature documented here (Figs. 4, 7, 10,
13) offer the highest potential for future observational studies interested in quantifying the impact of climate change on
denudation and weathering rates.


**Acknowledgements**

The model simulations presented in this study are freely available to interested persons by contacting S. Mutz or T. Ehlers. We note however that the data files are very large (~4 TB, and too large to archive in journal supplementary material) and require familiarity in reading/plotting NetCDF formatted files. European Research Council (ERC) Consolidator Grant number 615703 provided support for S. Mutz. Additional support is acknowledged from the German science foundation (DFG) priority research program 1803 (EarthShape: Earth Surface Shaping by Biota; grants EH329/14-1 and EH329/17-1). We thank B. Adams and J. Starke for constructive discussions. We also thank the reviewers (incl. A. Wickert) for their constructive feedback on this manuscript, which helped to significantly improve it. The DKRZ is thanked for computer time used for some of the simulations presented here. C. Stepanek, M. Werner, and G. Lohmann acknowledge funding by the Helmholtz Climate Initiative Reklim and the Alfred Wegener Institute's research programme Marine, Coastal and Polar Systems.

**Figure Captions**

**Figure 1** Topography for regions (a) tropical South Asia, (b) temperate South Asia, (c) high altitude South Asia, (d) temperate South America, (e) tropical South America, (f) temperate Andes, (g) tropical Andes, SE Alaska and Cascadia.

**Figure 2** Global PI annual mean near-surface temperatures (a), and deviations of MH, LGM and PLIO annual mean near-surface temperatures from PI values (b). Units are °C and insignificant (p < 99%) differences (as determined by a t-test) are greyed out.

**Figure 3** Global PI annual mean precipitation (a), and deviations of MH, LGM and PLIO annual mean near-surface temperatures from PI values (b). Units are mm/yr.

**Figure 4** PI annual mean near-surface temperatures (a), and deviations of MH, LGM and PLIO annual mean near-surface temperatures from PI values (b) for the South Asia region. Insignificant (p < 99%) differences (as determined by a t-test) are greyed out.

**Figure 5** PI, MH, LGM and PLIO annual mean precipitation in (a) tropical South Asia, (b) temperate South Asia, and (c) high-altitude South Asia; PI, MH, LGM and PLIO annual mean temperatures in (d) tropical South Asia, (e) tem-

perate South Asia, and (f) high-altitude South Asia. For each time slice, the minimum, lower 25th percentile, median,
upper 75th percentile and maximum are plotted.
**Figure 6** Geographical coverage and characterization of climate classes $C_1$- $C_6$ based on cluster-analysis of 8 variables
(near surface temperature, seasonal near surface temperature amplitude, total precipitation, seasonal precipitation
amplitude, u-wind in January and July, v-wind in January and July) in the South Asia region. The geographical cov-
erage of the climates $C_1$- $C_6$ is shown on the left for PI (a), MH (b), LGM (c) and PLIO (d); the complementary,
time-slice specific characterization of $C_1$- $C_6$ for PI (e), MH (f), LGM (g) and PLIO (h) is shown on the right.
**Figure 7** PI annual mean near-surface temperatures (a), and deviations of MH, LGM and PLIO annual mean near-sur-
face temperatures from PI values (b) for western South America. Insignificant (p < 99%) differences (as determined
by a t-test) are greyed out.
**Figure 8** PI, MH, LGM and PLIO annual mean precipitation in (a) tropical South America, (b) temperate South Amer-
ica, (c) tropical Andes, and (d) temperate Andes; PI, MH, LGM and PLIO annual mean temperatures in (e) tropical
South America, (f) temperate South America, (g) tropical Andes, and (h) temperate Andes. For each time slice, the
minimum, lower 25th percentile, median, upper 75th percentile and maximum are plotted.
**Figure 9** Geographical coverage and characterization of climate classes $C_1$- $C_6$ based on cluster-analysis of 8 variables
(near surface temperature, seasonal near surface temperature amplitude, precipitation, seasonal precipitation amp-
litude, u-wind in January and July, v-wind in January and July) in western South America. The geographical cover-
age of the climates $C_1$- $C_6$ is shown on the left for PI (a), MH (b), LGM (c) and PLIO (d); the complementary, time-
slice specific characterization of $C_1$- $C_6$ for PI (e), MH (f), LGM (g) and PLIO (h) is shown on the right.
**Figure 10** PI annual mean near-surface temperatures (a), and deviations of MH, LGM and PLIO annual mean near-sur-
face temperatures from PI values (b) for the South Alaska region. Insignificant (p < 99%) differences (as determined
by a t-test) are greyed out.
**Figure 11** PI, MH, LGM and PLIO annual mean precipitation (a), and mean annual temperatures (b) in South Alaska.
For each time slice, the minimum, lower 25th percentile, median, upper 75th percentile and maximum are plotted.
**Figure 12** Geographical coverage of climate classes $C_1$- $C_4$ based on cluster-analysis of 4 variables (near surface tem-
perature, seasonal near surface temperature amplitude, total precipitation, seasonal total precipitation amplitude) in
southern Alaska. The geographical coverage of the climates $C_1$- $C_4$ is shown on the left for PI (a,), MH (b), LGM (c)
and PLIO (d); the complementary, time-slice specific characterization of $C_1$- $C_6$ for PI (e), MH (f), LGM (g) and
PLIO (h) is shown on the right.
**Figure 13** PI annual mean near-surface temperatures (a), and deviations of MH, LGM and PLIO annual mean near-sur-
face temperatures from PI values (b) for the US Pacific Northwest. Insignificant (p < 99%) differences (as determ-
ined by a t-test) are greyed out.
**Figure 14** PI, MH, LGM and PLIO annual mean precipitation (a), and annual mean temperatures (b) in the Cascades,
US Pacific Northwest. For each time slice, the minimum, lower 25[th] percentile, median, upper 75[th] percentile and
maximum are plotted.
**Figure 15** Geographical coverage and characterization of climate classes $C_1$- $C_4$ based on cluster-analysis of 4 variables
(near surface temperature, seasonal near surface temperature amplitude, total precipitation, seasonal total precipita-
tion amplitude) in the Cascades, US Pacific Northwest. The geographical coverage of the climates $C_1$- $C_4$ is shown
on the left for PI (a), MH (b), LGM (c) and PLIO (d); the complementary, time-slice specific characterization of $C_1$-
$C_6$ for PI (e), MH (f), LGM (g) and PLIO (h) is shown on the right.
**Figure 16** Simulated annual mean precipitation deviations of MH (left) and LGM (right) from PI values in South Asia,
and temporally corresponding proxy-based reconstructions, indicating wetter (upward facing blue triangles), drier
(downward facing red triangles) or similar (grey circles) conditions in comparison with modern climate. MH proxy-
based precipitation differences are taken from Mügler et al. (2010) (**66**), Wischnewski et al. (2011) (**67**), Mischke et
al. (2008), Wischnewski et al. (2011), Herzschuh et al. (2009) (**68**), Yanhong et al. (2006) (**69**), Morrill et al. (2006)
(**70**), Wang et al. (2002) (**71**), Wuennemann et al. (2006) (**72**), Zhang et al. (2011), Morinaga et al. (1993),
Kashiwaya et al. (1995) (**73**), Shen et al. (2005) (**74**), Liu et al. (2014) (**75**), Herzschuh et al. (2006) (**76**), Zhang and
Mischke (2009)  (**77**), Nishimura et al. (2014) (**78**), Yu and Lai (2014) (**79**), Gasse et al. (1991) (**80**), Van Campo et
al. (1996) (**81**), Demske et al. (2009) (**82**), Kramer et al. (2010) (**83**), Herzschuh et al. (2006) (**84**), Hodell et al.
(1999)(**85**), Hodell et al. (1999) (**86**), Shen et al. (2006) (**87**), Tang et al. (2000) (**88**), Tang et al. (2000) (**89**), Zhou et
al. (2002) (**90**), Liu et al. (1998) (**91**), Asashi (2010)(**92**), Kotila et al. (2009) (**93**), Kotila et al. (2000) (**94**), Wang et
al. (2002) (**95**), Hu et al. (2014) (**96**), Hodell et al. (1999) (**97**), Hodell et al. (1999) (**98**).
**Figure 17** Simulated annual mean precipitation deviations of MH (left) and LGM (right) from PI values in South Amer-
ica, and temporally corresponding proxy-based reconstructions, indicating wetter (upward facing blue triangles),
drier (downward facing red triangles) or similar (grey circles) conditions in comparison with modern climate. MH
proxy-based precipitation differences are taken from Bird et al. (2011) (**1**), Hansen et al (1994) (**2**), Hansen et al
(1994) (**3**), Hansen et al (1994) (**4**), Hansen et al (1994) (**5**), Hansen et al (1994) (**6**), Hillyer et al. (2009) (**7**),
D'Agostino et al. (2002) (**8**), Baker et al. (2001) (**9**), Schwalb et al (1999) (**10**), Schwalb et al (1999) (**11**), Schwalb
et al (1999) (**12**), Schwalb et al (1999) (**13**), Moreno et al (2009) (**14**), Pueyo et al (2011) (**15**), Mujica et al (2015)
(**16**), Fritz et al. (2004) (**17**), Gayo et al. (2012) (**18**), Latorre et al. (2006) (**19**), Latorre et al. (2003) (**20**), Quade et al
(2008) (**21**), Bobst et al. (2001) (**22**), Grosjean et al. (2001) (**23**), Betancourt et al. (2000) (**24**), Latorre et al. (2002)
(**25**), Rech et al. (2003) (**26**), Diaz et al. (2012) (**27**), Maldonado et al (2005) (**28**), Diaz et al. (2012) (**29**), Lamy et
al. (2000) (**30**), Kaiser et al. (2008) (**31**), Maldonado et al. (2010) (**32**), Villagrán et al. (1990) (**33**), Méndez et al.
(2015) (**34**), Maldonado et al. (2006) (**35**), Lamy et al. (1999) (**36**), Jenny et al. (2002) (**37**), Jenny et al. (2002b)
(**38**), Villa-Martínez et al. (2003) (**39**), Bertrand et al. (2008) (**40**), De Basti et al. (2008) (**41**), Lamy et al. (2009)
(**42**), Lamy et al. (2002) (**43**), Szeicz et al. (2003) (**44**), de Porras et al. (2012) (**45**), de Porras et al. (2014) (**46**),
Markgraf et al. (2007) (**47**), Siani et al. (2010) (**48**), Gilli et al. (2001) (**49**), Markgraf et al. (2003) (**50**), Stine et al.
(1990) (**51**).

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

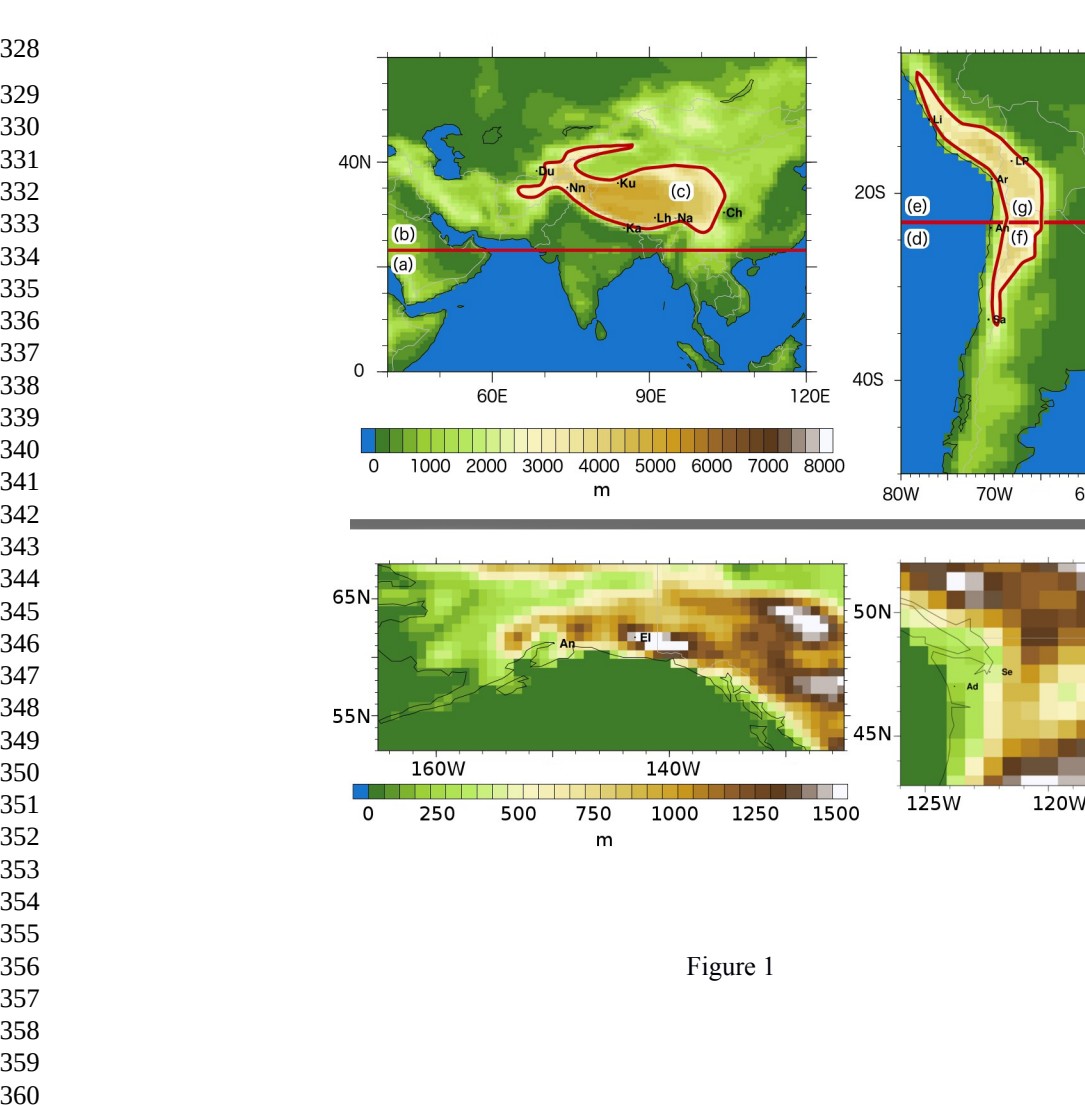

Figure 1

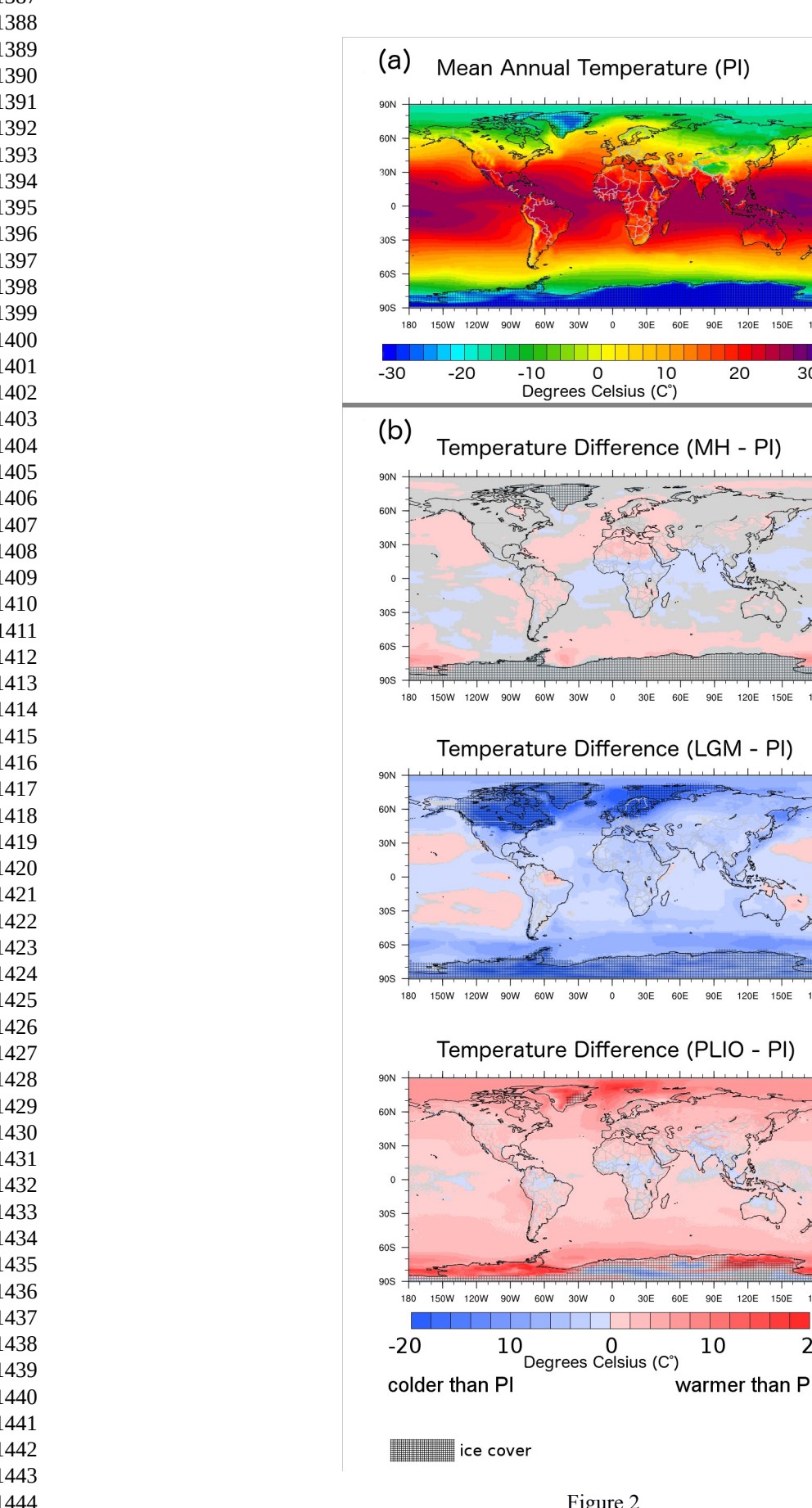

Figure 2

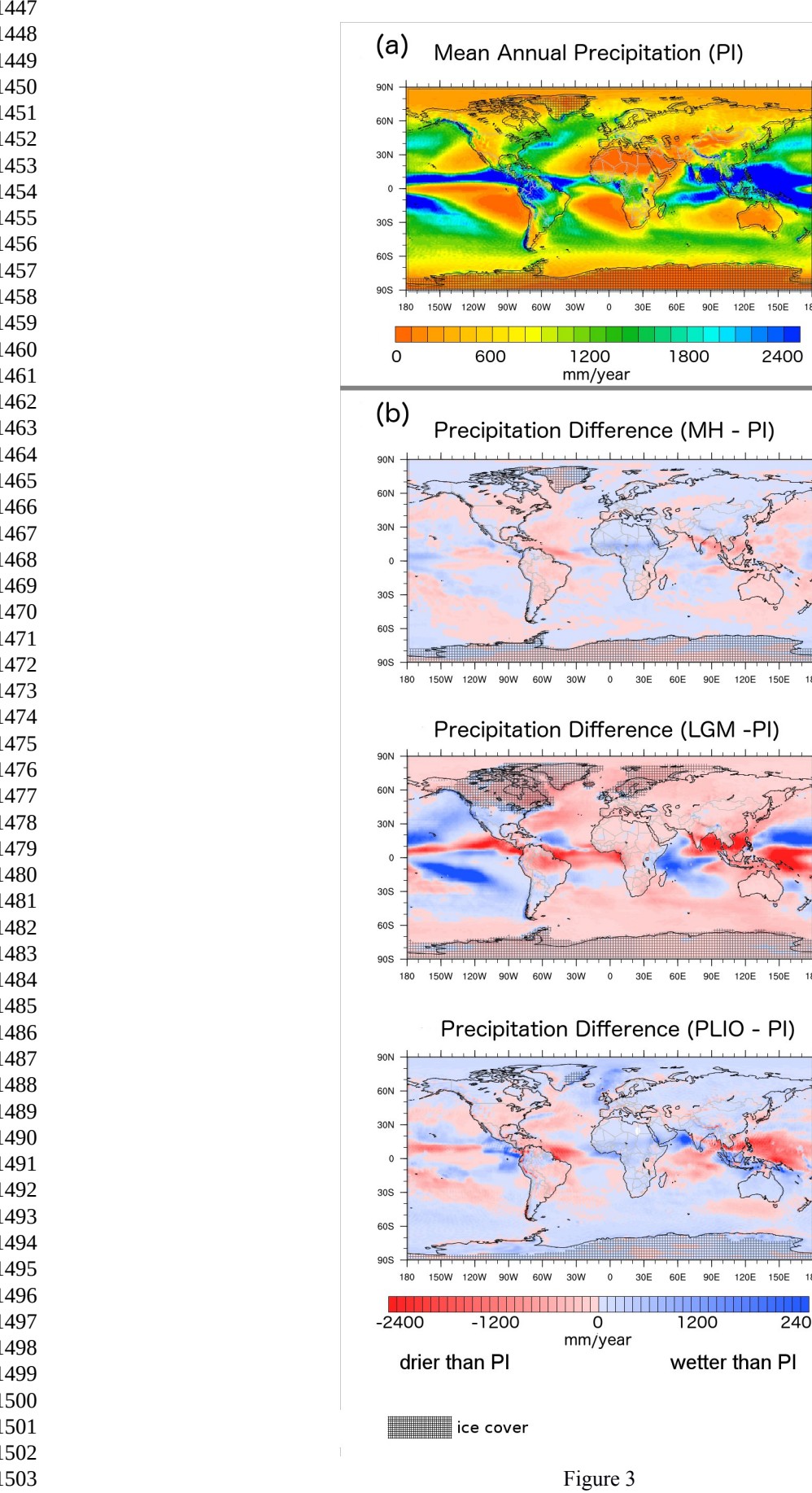

Figure 3

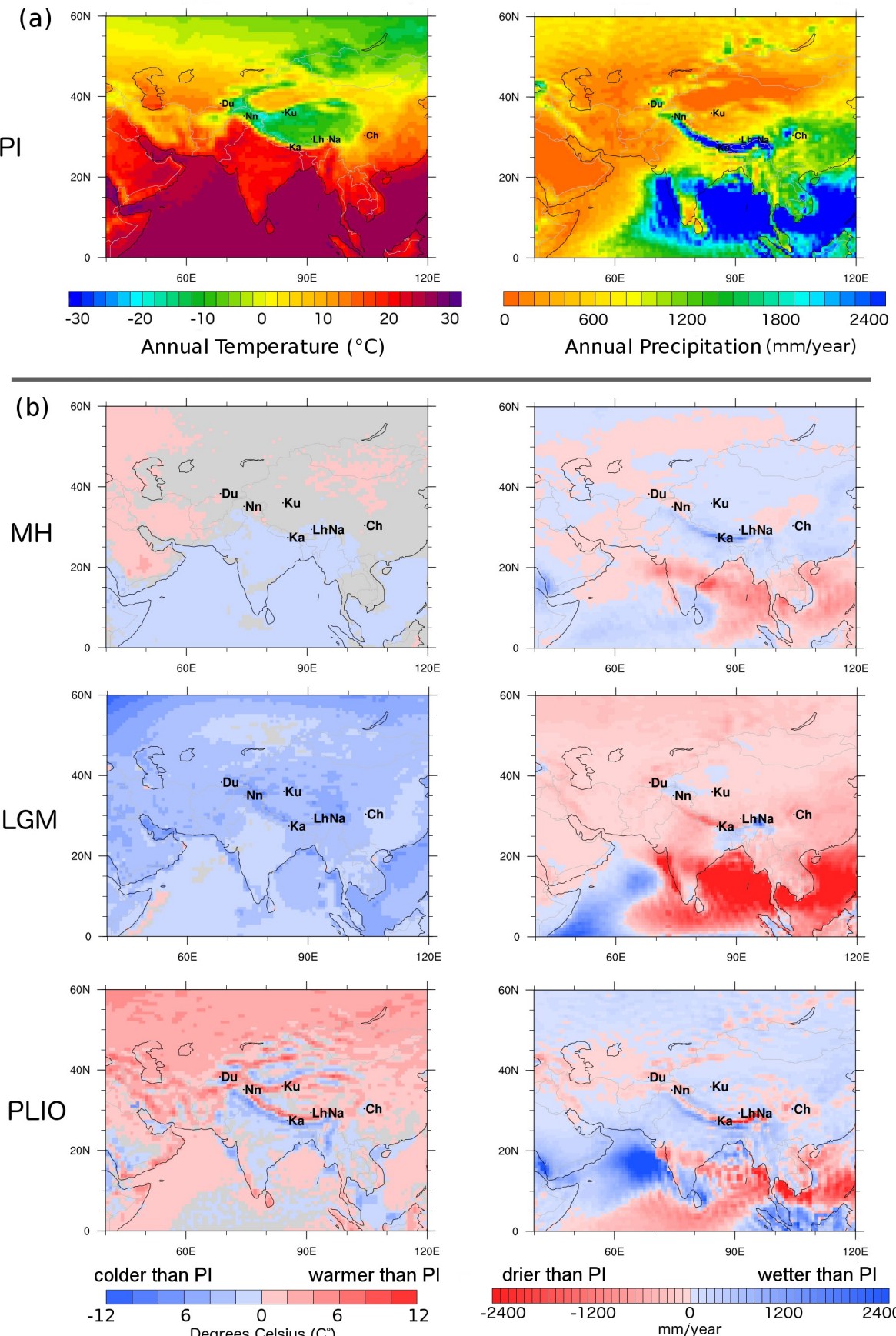

Figure 4

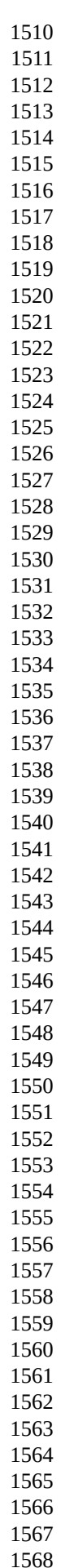

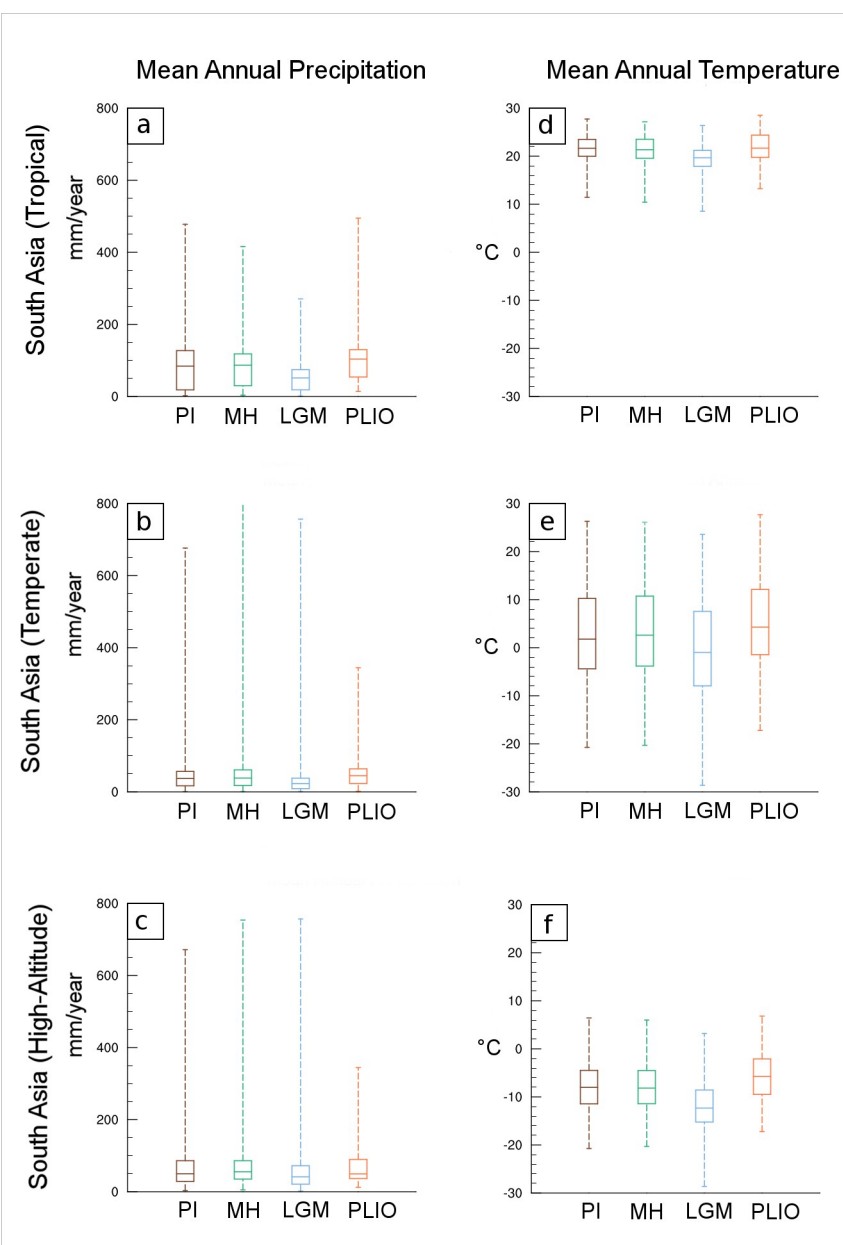

Figure 5

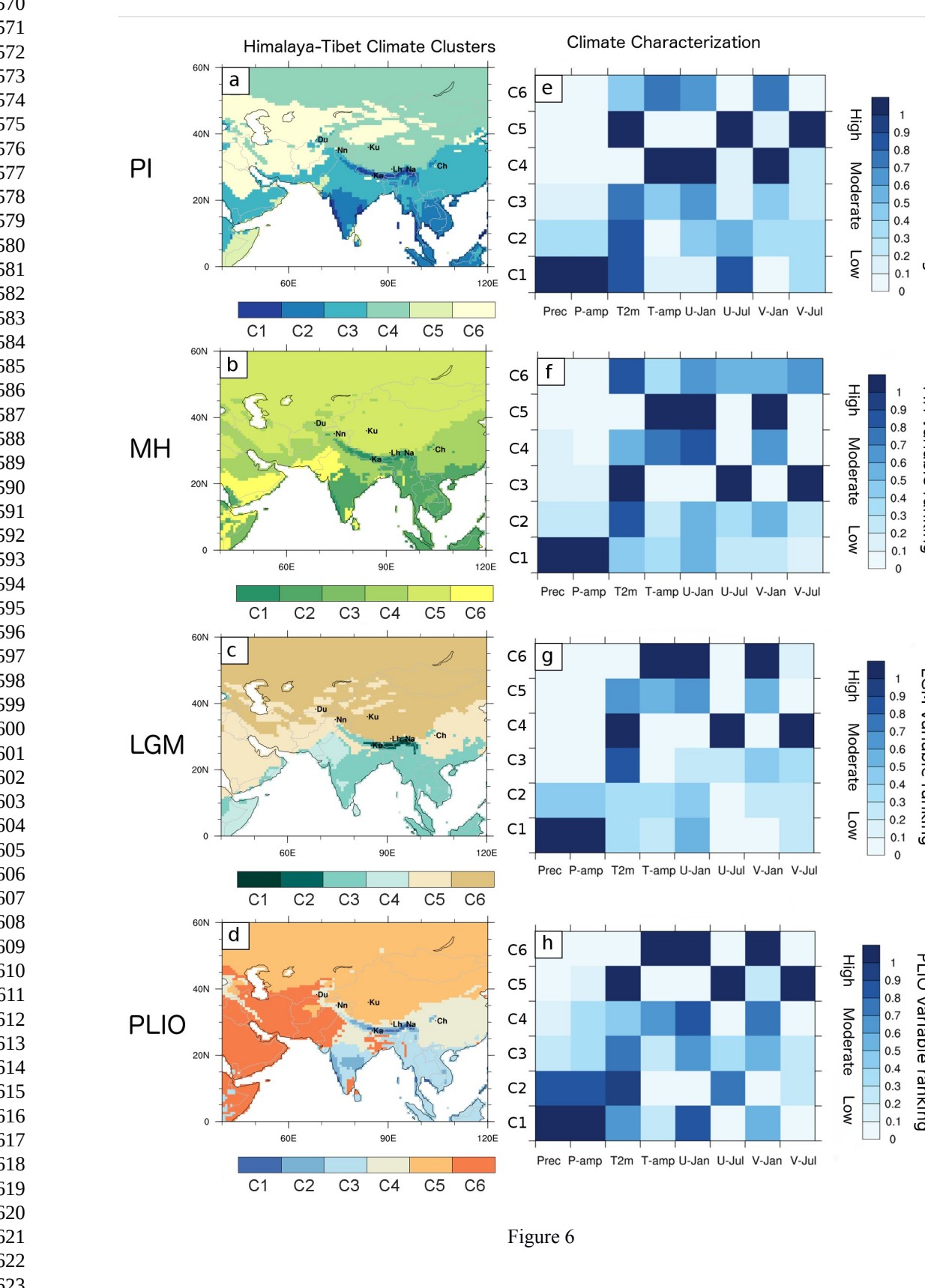

Figure 6

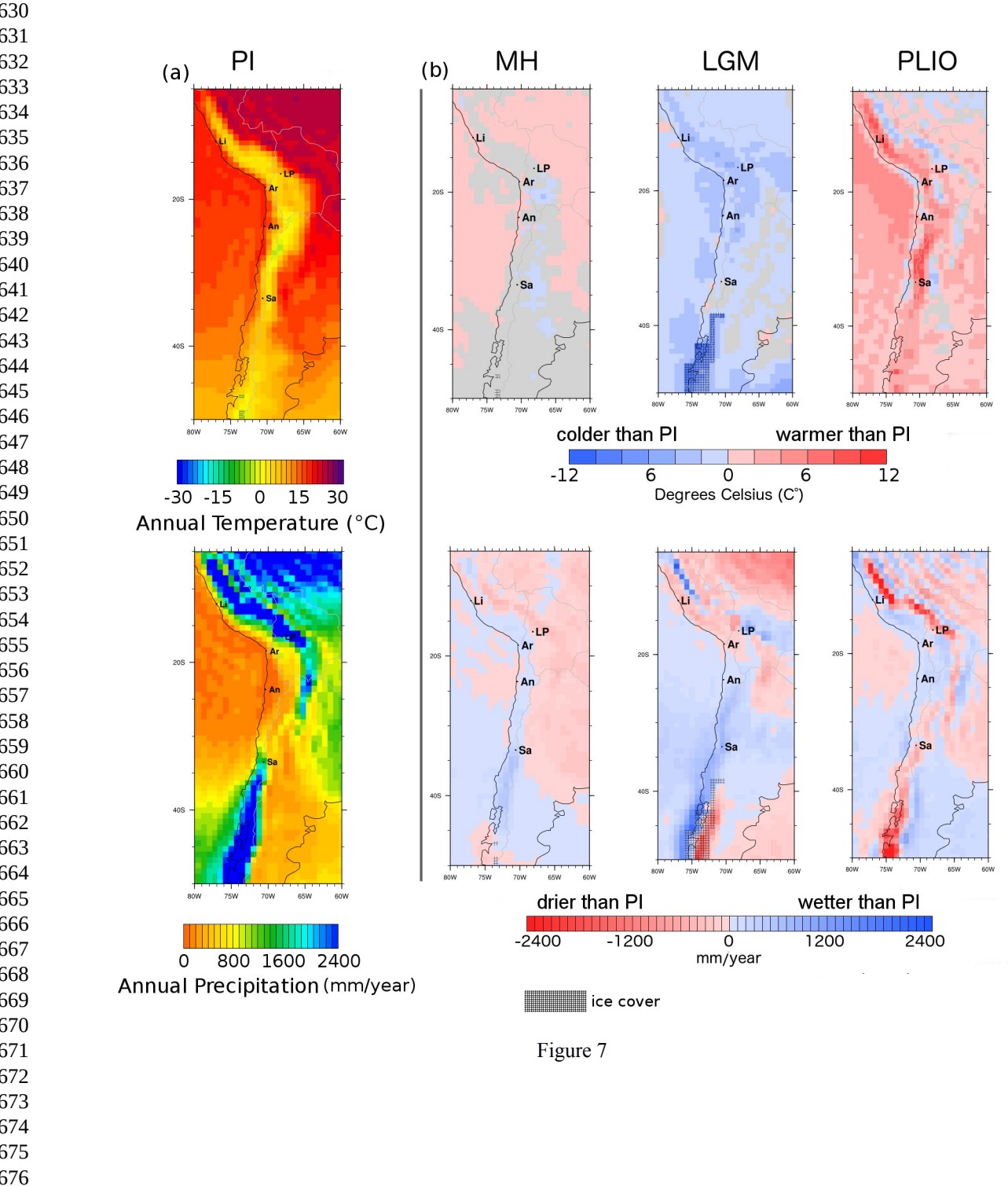

Figure 7

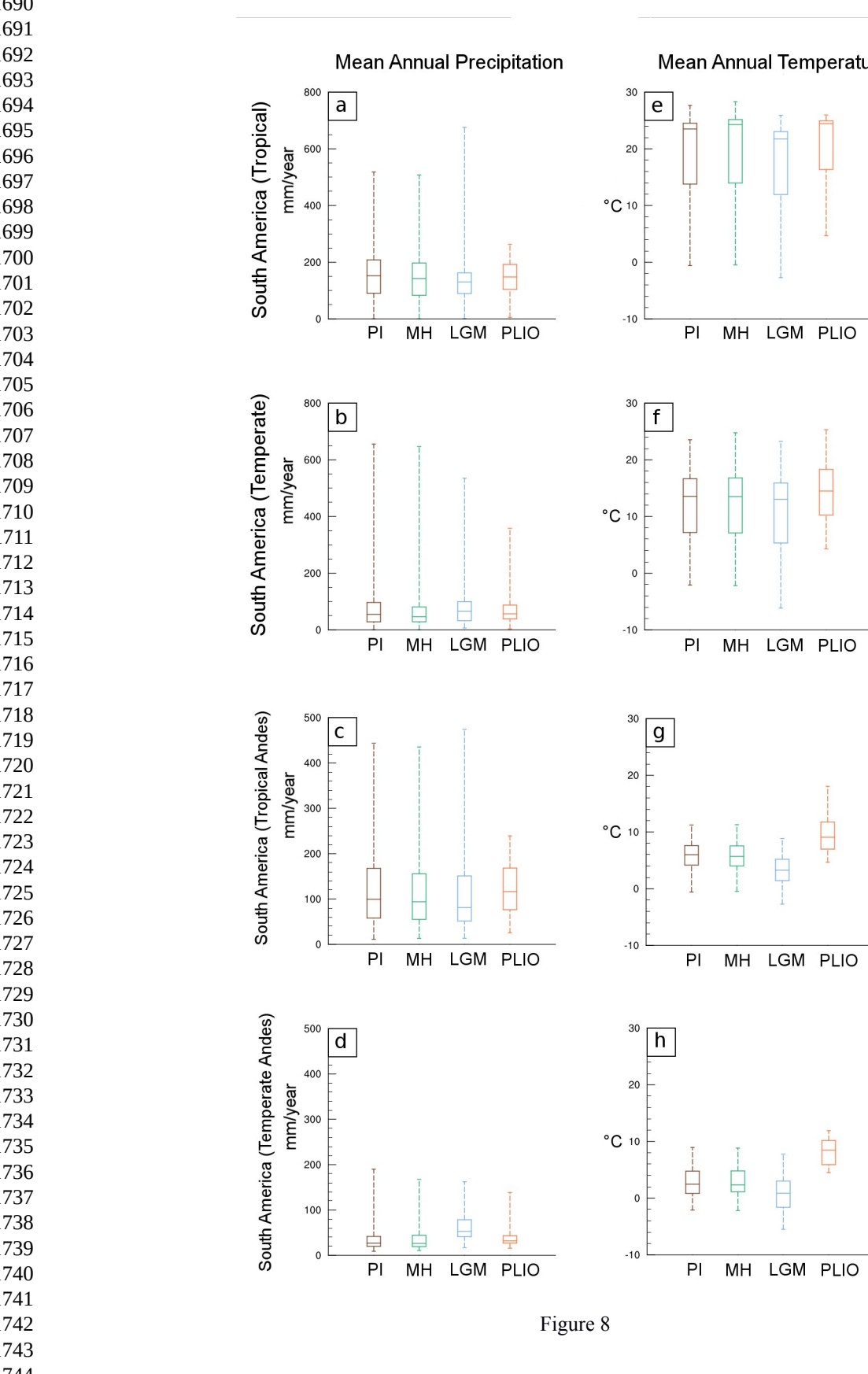

Figure 8

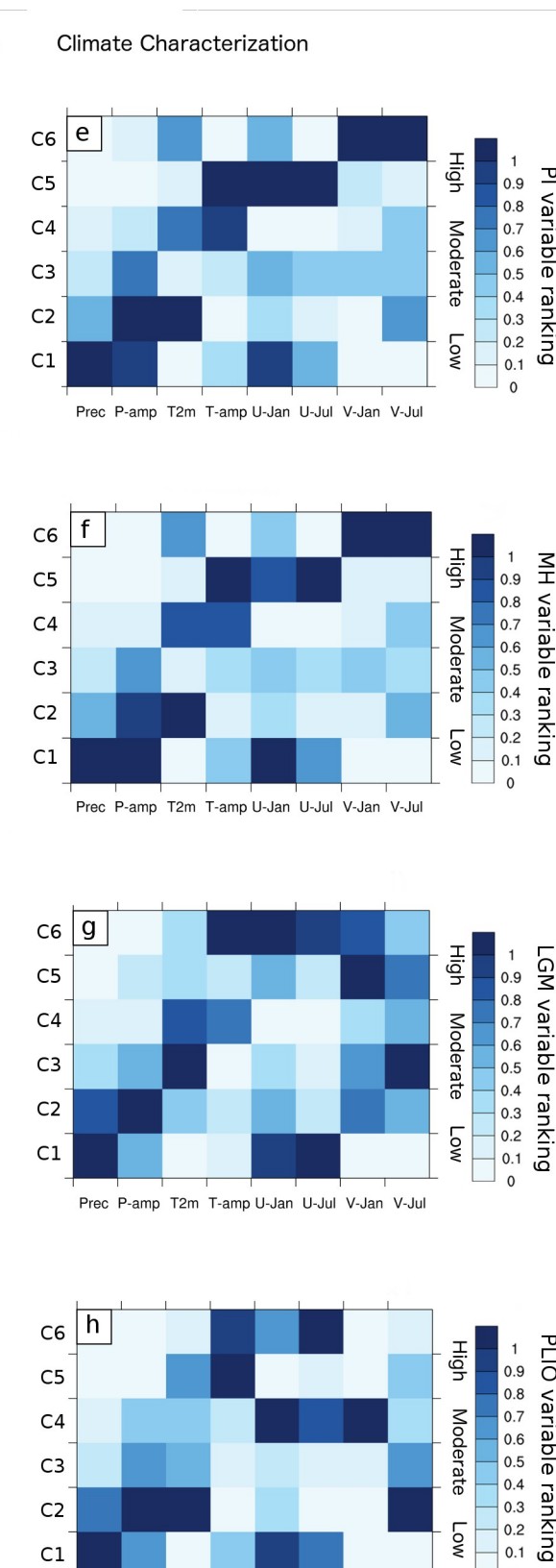

Figure 9

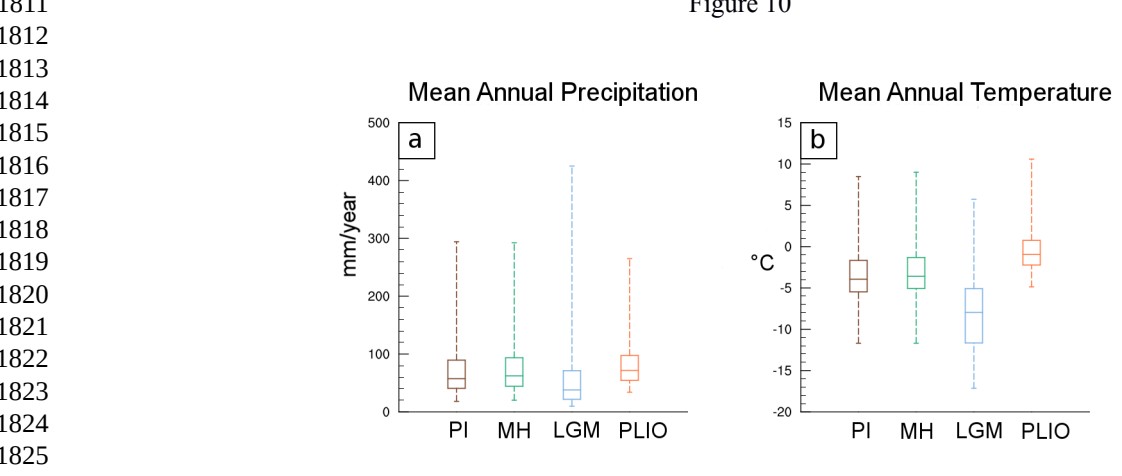

(a)

PI

Annual Temperature (°C)

Annual Precipitation (mm/year)

(b)

MH

LGM

PLIO

colder than PI          warmer than PI

Degrees Celsius (C°)

drier than PI          wetter than PI

mm/year

ice cover

Figure 10

Mean Annual Precipitation

Mean Annual Temperature

Figure 11

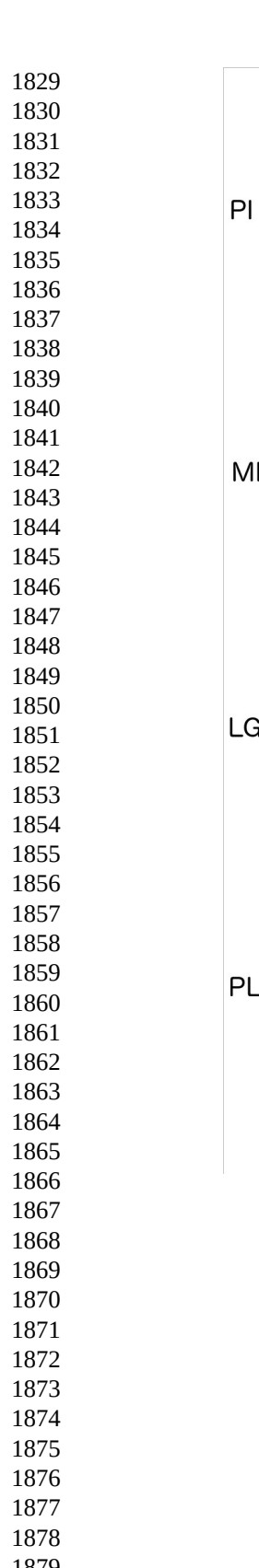

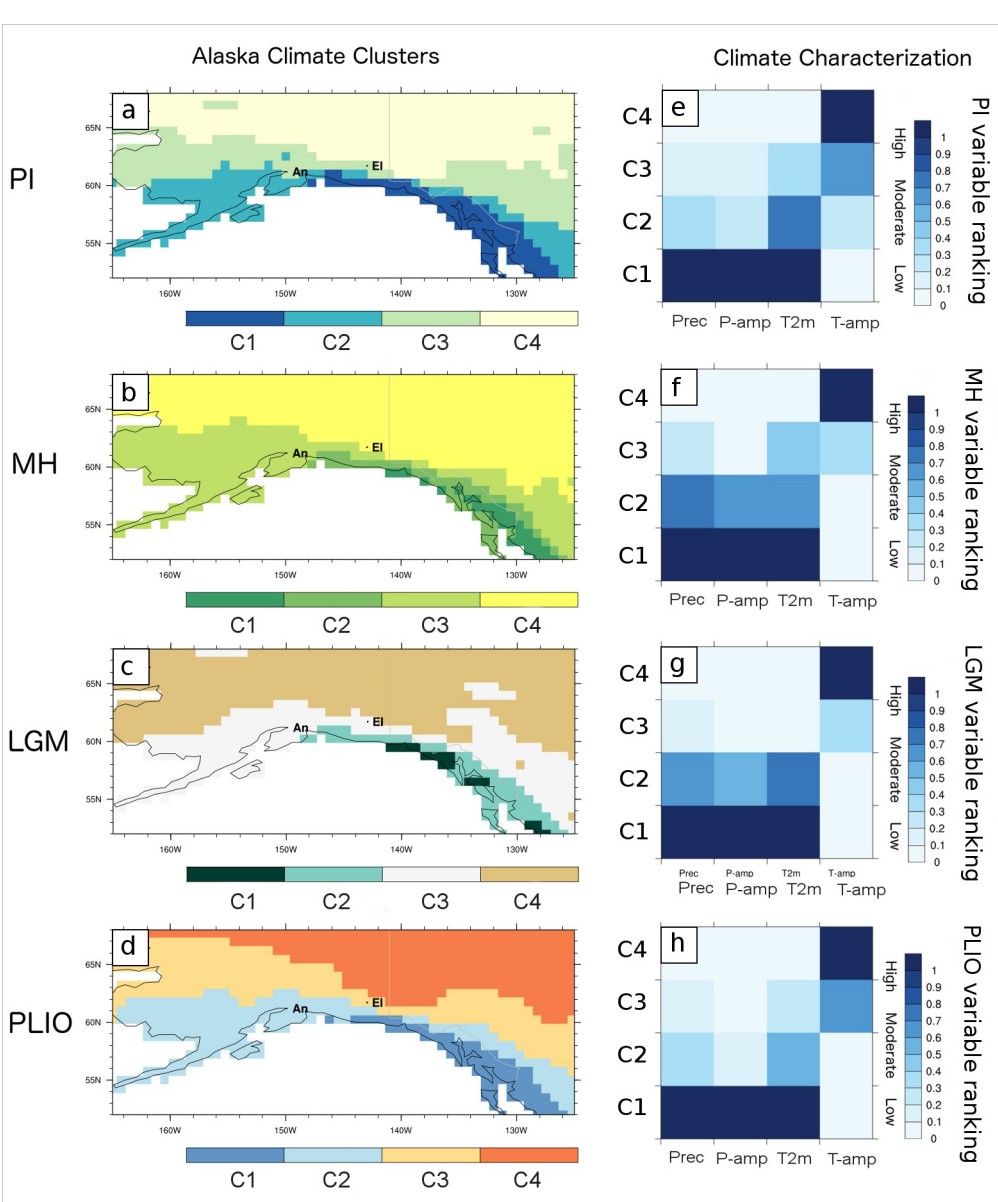

Figure 12

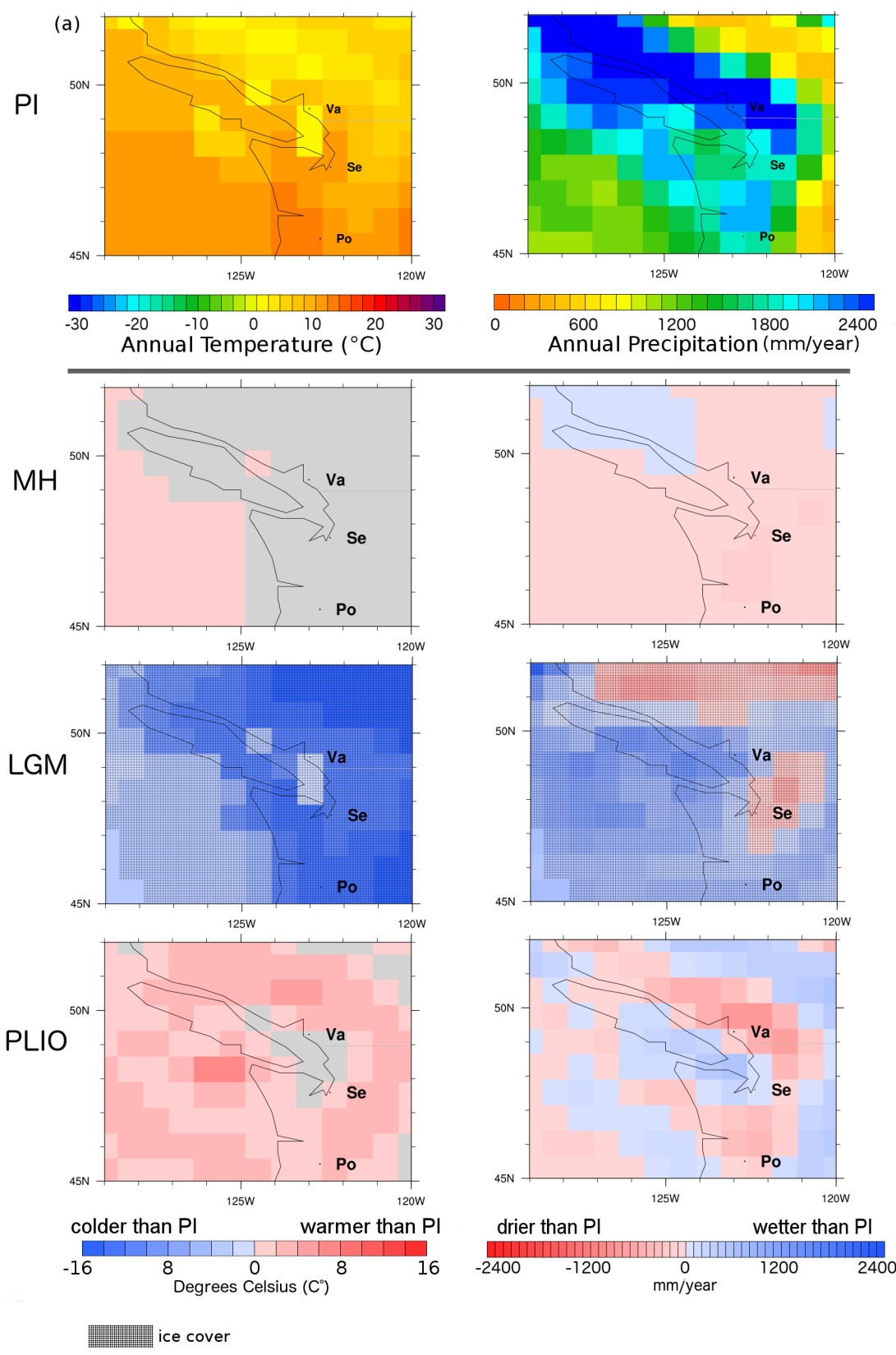

Figure 13

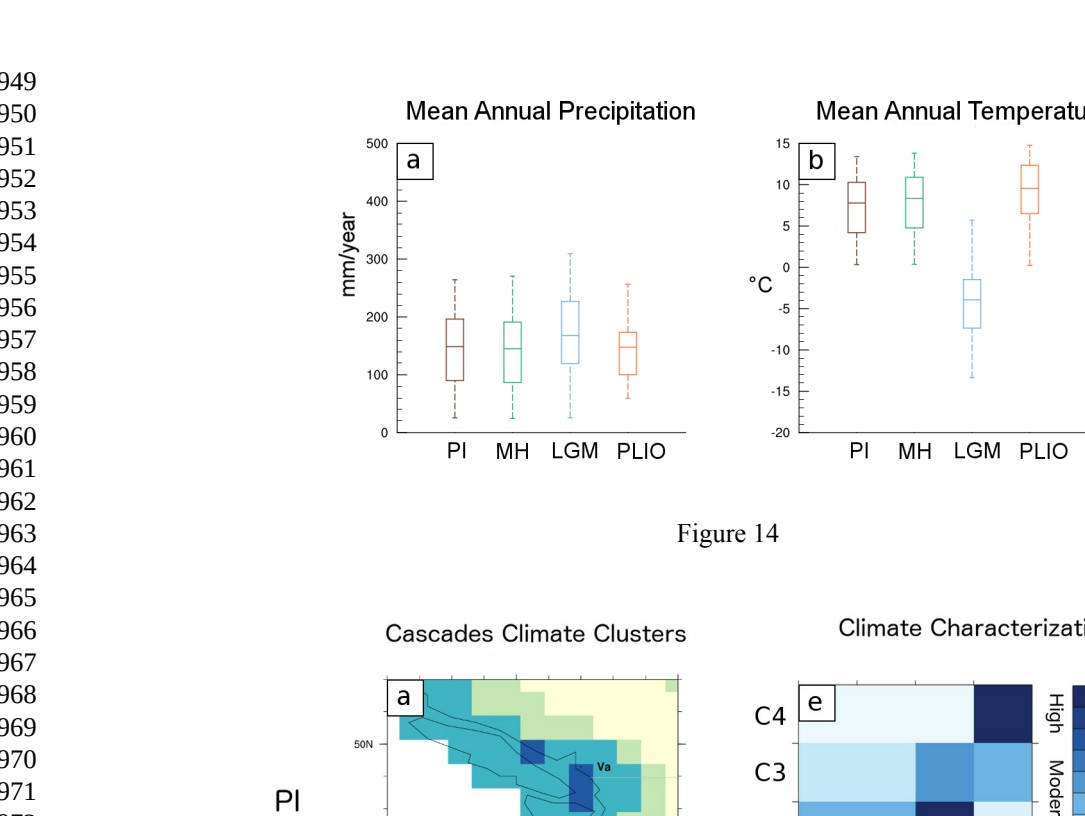

Figure 14

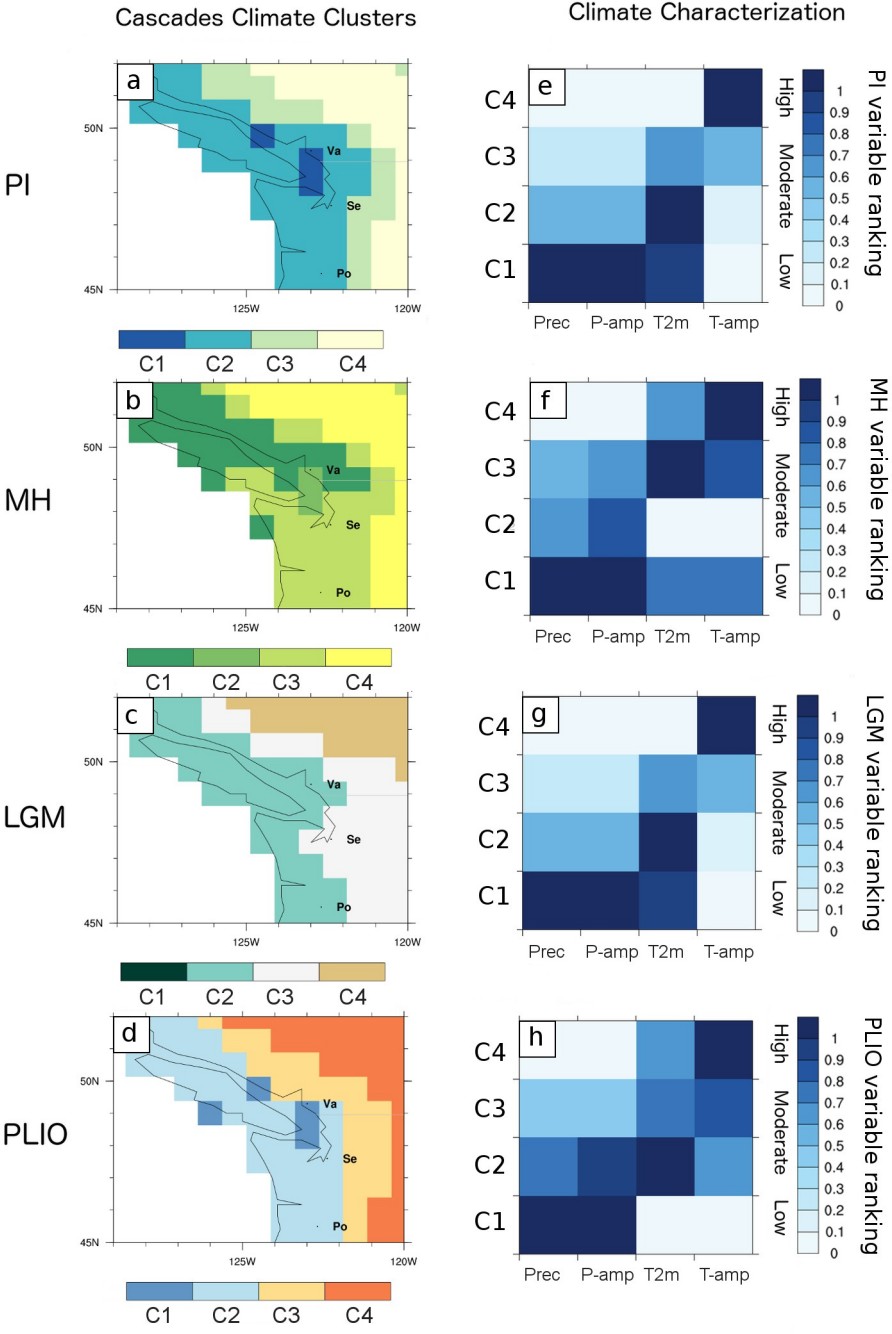

Figure 15

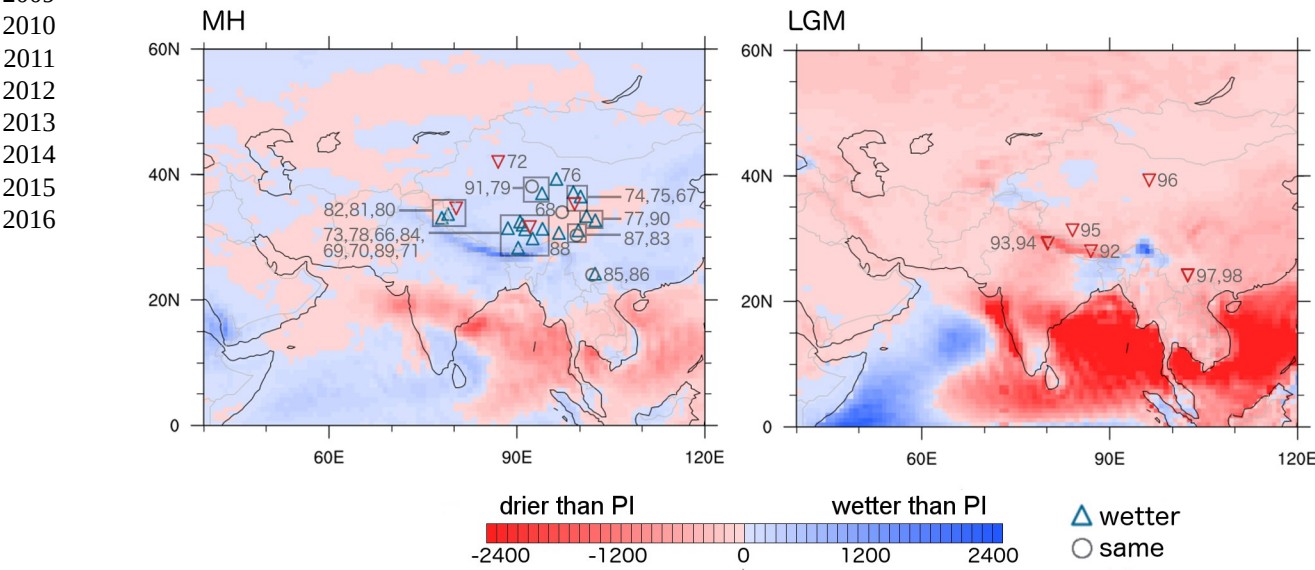

Figure 16

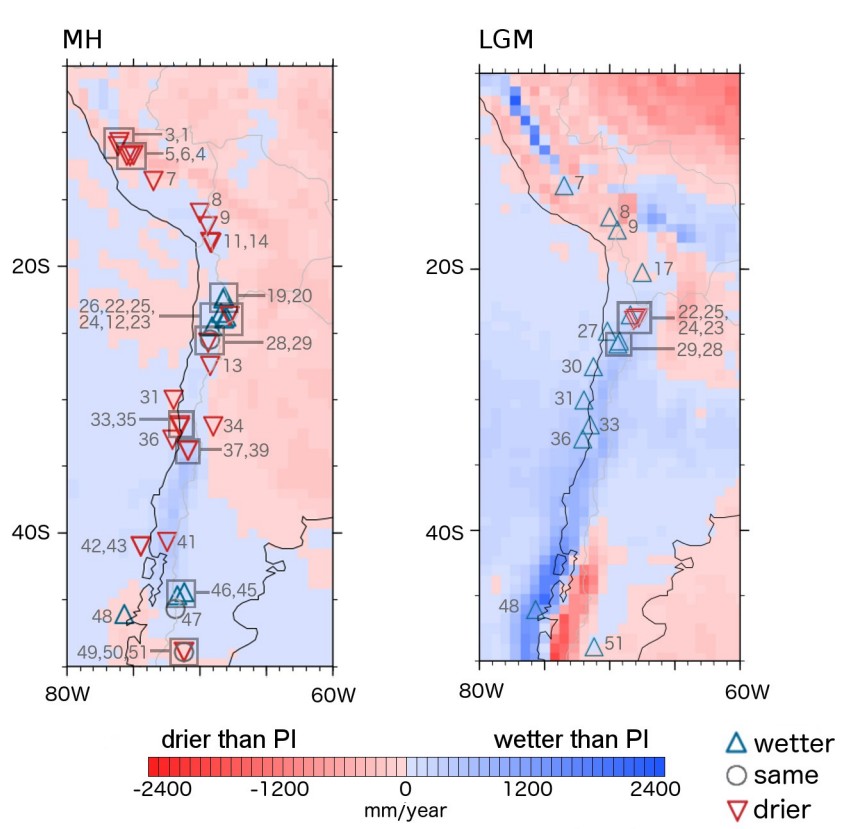

Figure 17