# Peer review of "Estimates of Late Cenozoic climate change relevant to Earth"

_Earth Surface Dynamics, 2017_

## Referee Comment (RC1) · A. Wickert (Referee) · 23 Sep 2017

**Summary:**

Mutz et al. look to use paleoclimate GCMs to identify drivers of past geomorphic change. This is a topic for which I hold great interest, and I feel that the authors have crafted a very useful set of model results that they leave underutilized. As such, I feel inclined to accept the paper on the basis of the useful results, but to request major revisions such that they do their own work justice.

My major concerns, which will become clear in the line-by-line comments (please feel free to respond to similar comments en masse) are as follows:

1. The paper is motivated by denudation rates and landscape evolution, but really

includes this as a speculative wrapper that is not substantiated. I suggest that instead you propose testable hypotheses surrounding your findings.

2. Related to #1, much of the text is a litany of "temperature was X here ...". I find such statements of results useful only insofar as they expand upon a figure (associated with a supplementary data set) that presents the results. While these sections are written clearly, I would suggest that the authors focus on a set of geomorphic questions (if this be their motivation) and how the model-data set informs those questions.

3. Many of the discussions of model results are of ice-covered regions, yet no consideration of direct glacial erosion is given. Furthermore, no reference to the changes in the statistics of discharge or catchment area in ice-covered regions is given. This seems a disservice to this relatively high-resolution paleoclimate AGCM: the geologic setting *must* be considered, otherwise it seems that the authors' pushing on the modeling end has not been matched by a simple geological history sanity check. I would suggest that either significantly glaciated regions and the catchments that they feed be masked out, or that glacial erosion and its associated processes be included in the discussion.

4. (Discussed only here): You have not compared your models against any data. I understand that this may be simply a modeling exercise that you do compare to other models. However, I think that such a comparison could assuage skepticism about your results and lend support to your case, especially if you include it as part of a local case study (see the third point below).

The core of these three points is that, with a bit more care, I think your results could say something really useful to the geomorphic community. Currently, the paper seems to be more a statement of, "this is important to geomorphology", followed by a long list of the model results. I challenge the authors to demonstrate (rather than simply stating) the importance of their work to geomorphology in a way that includes how it may impact the way scientists view Quaternary landscape evolution. Ideas include:

- Changes in means (done)

- Changes in statistical distributions of temperature and precipitation – think extreme events, frost-cracking window, etc.

- A focus on a few iconic regions while *explicitly* ignoring significantly ice-covered domains (I think this would be easiest, though obviously would be thrilled if you decided to tackle glacial processes)

- Using this focus to build a template for how to use paleoclimate GCM outputs to advance the field of geomorphology.

Currently, I think that the work is acceptable following changes for internal consistency and geological accuracy (see #1 and #3), but I think that you could be selling yourselves short if you don't dig just a tiny bit deeper to investigate your forcings and their impact on geomorphology.

I hope that you find these comments helpful in continuing to craft an insightful piece of work out of what seems to be a strong modeling approach.

**Line-by-line:**

23. US Pacific Northwest Pacific → drop second "Pacific"

29. future observational studies interested in quantifying → future observational studies **that quantify** (studies can't be interested in things, strictly speaking)

53. orogen scale → orogen-scale

~57. A couple of recent studies from the climate science community shed light on the impacts of the Andes (first ref below) and continents in general (second ref below). In case these are interesting to you, I'm pasting the bibliographic information here:

Maroon, E. A., D. M. W. Frierson, and D. S. Battisti (2015), The tropical precipitation response to Andes topography and ocean heat fluxes in an aquaplanet model, J. Clim., 28(1), 381–398, doi:10.1175/JCLI-D-14-00188.1.

[Figure]

Maroon, E. A., D. M. W. Frierson, S. M. Kang, and J. Scheff (2016), The precipitation response to an idealized subtropical continent, J. Clim., 29(12), 4543–4564, doi:10.1175/JCLI-D-15-0616.1.

73-75. "Furthermore, recent controversy exists concerning the spatial and temporal scales over which geologic and geochemical observations can record climate-driven changes in weathering and erosion [e.g. Whipple, 2009; von Blanckenburg et al., 2015; Braun, 2016].": I see that you do not return to this point later, so could you describe the controversy for those who are not familiar with it?

81. I see that later you discuss a little about what an AOGCM may do, but I will be looking for justification about how an AGCM may suffice. Is this in part because you prescribe the b.c.'s and you are running it for 17 years only? If so, could you discuss potential systematic variations between this and an AOGCM?

89. "PLIO to the Last Glacial Maximum": as you include no time-slices between these, I suggest making these part of the list and dropping the "to the".

147. "This section describes the clustering method used in this study." You could drop this sentence – the section title should be enough for even an inattentive reader!

176-178. I was wondeirng how you picked the number of clusters: I am glad to see that you performed a thorough search.

Section 3: Much of this is information that I find better communicated through figures than with text. It is clearly written, however, and I am reluctant to suggest a rewrite for brevity in a length-unconstrained journal so long as the text can be co-located with the figure.

190-192. I see you have another "This section describes..." sentence. If this is your preferred way to write, you may keep it; here, the second sentence is not such a good topic-sentence replacement.

197-198. i.e. over the ice sheets. (This applies to other regions as well, and should be

important to point out if you are going to then discuss fluvial processes in orogens)

203-paragraph: Also because of local ice loss, presumably. So I think that the two prior paragraphs could have a new summary that "The greatest changes in temperature is observed where the greatest change in local ice extent occurs."

214-215. Have you considered discussions of the African Humid Period?

373-374. If you are looking at the influence of temperature and precipitation on erosion, and you are not including subglacial erosion, then your preceding text must indicate where your changes really are indicative of ice extent – both as a separate process domain and as a driver of fluvial processes and potential changes in the statistics of river discharge.

Section 4.1. Your first paragraph (weathering) differs from the content (comparing your model results with those published). These should be in different subsections, and the weathering paragraph may need to be expanded. Your "weathering and erosion" paragraph also neglects direct effects of glaciers, ice caps, and ice sheets, which were globallly significant.

Section 4.2. Once again, your discussion is often of formerly (or currently) ice-covered regions without explicitly acknowledging that this is a different process domain. In addition, as with the previous section, the body paragraphs are mostly about model comparison and regional changes with sparse link to the landscape-evolution factors indicated in the topic paragraph.

416-423. Please discuss the direct influence of glaciers on the erosion orogens in the context of changing precipitation (and therefore mass balance). Is it significant or not?

433-434. "Coastal North America"? Doesn't look like it: seems to be most of NA south of the ice margin.

Section 4.3. The authors describe the results here, but I find the connection to erosion rates to be insufficiently described compared to how they are highlighted in the topic

sentence, as well as in the abstract. I would like you to go one step beyond "ought to be considered" and actually posit how you expect the erosion rates – and therefore, the balance between erosion and exhumation and perhaps the equilibrium shapes of the mountains and their rivers – to vary. Otherwise, you are suggesting future work rather than actually describing the possible geomorphic significance – and I think underutilizing your results in a paper that is clearly targeted towards geomorphologists.

498. "which may favour frost driven weathering during glacial climate states" – the St. Elias range was covered by glaciers! Yes, there can be some frost-cracking around the ice, but don't you think this is important too? http://instaar.colorado.edu/groups/QGISL/ak_paleoglacier_atlas/gallery/index.html

508. "enhanced sediment production driven by frost processes" – same as above. Glaciers were there. Consider them.

Conclusions: Comparison to other models: is this match surprising or no? Did you (mostly) use the same inputs and simply increase the grid resolution? If so, could you comment on how the improved grid and possible variations in inputs and use of the ocean as a boundary condition may have affected (or not) your results as compared to those of earlier studies? This would be more useful to include in the discussion than a simple list of "Our temperature in place Y was $T_0$, and X et al. wrote that they found it was $T_1$, which is close to $T_0$. Think big-picture, in both process and numerics!

533. Did your 8-10 degC changes occur significantly over areas that would be affected by hillslope or fluvial processes? (i.e. unglaciated areas?)

---

## Referee Comment (RC2) · Anonymous Referee #2 · 14 Nov 2017

The authors simulate the pre-industrial, mid-Holocene, LGM, and Pliocene climates using the ECHAM5 general circulation model. The motivation is to understand how past climatic states may change regional climatologies, particularly over mountainous areas that have been the focus of much erosion and geomorphic work. The authors find that past climatic states (particularly the LGM and Pliocene) do produce changes in absolute temperature and precipitation and in the annual ranges of these two climatic variables.

Overall, I did not find the manuscript to be a particularly useful addition to the literature. Though the motivation is potentially novel, the analysis is not complete and needs substantially more work. Most of the manuscript focuses on simply describing climatic changes, while neglecting novel analyses. Consequently, at this point, I recommend

reject, though perhaps with substantial work (including more modeling, comparison with existing models, and/or comparison with data), it may become publishable in the future in ESurf.

First, the authors need to decide what the point of the paper is. Most of the paper reads as a description of climatological changes for 3 periods in the geologic past (MH, LGM, and Plio) (indeed, most of the text is written this way). However, this work has already been done, most prominently by the PMIP and PlioMIP set of model intercomparisons. What does this manuscript offer that these model intercomparisons have not already analyzed? A case could be made that these model intercomparisons are typically of a global nature (though there has been some work on changes in the Asian Monsoon systems using both PlioMIP and PMIP results (Jiang et al., 2013; Jiang and Lang, 2010; Zhang et al., 2013)), so that the analysis of the orogens in this study is useful. However, given that GCMs have difficulty simulating precipitation and in particular simulating precipitation over complex topography, the usefulness of simply describing changes over the Himalaya, Andes, Cascades, and St. Elias ranges is somewhat muted. For example, why should readers believe that ECHAM5 produces reasonable results over the St. Elias range? Why not use (or at least compare with) the existing model intercomparisons to look at changes in these locations? Most of the PlioMIP simulations are of a lower resolution than the model simulations presented here, though not all (Haywood et al., 2013). Many of the newer PlioMIP2 simulations are being run at a higher resolution and permit at least some comparison with the data here (Chandan and Peltier, 2017). Having myself tried to access PlioMIP data, I understand that it can be difficult to get access to the PlioMIP output, but if the point of the paper is to quantify actual climatological change, then comparison with other models is a must (or at least a thorough treatment of possible boundary condition uncertainties and additional ECHAM5 model runs to establish the sensitivity of precipitation/temperature in these areas); otherwise, we have no reason to believe that ECHAM5 presents anything resembling a proper picture of climatic change in the past.

[Figure]

Some parts of the paper address actual causes for climatic change (for example, discussion of the Pacific North American Teleconnection (no citation given). Is this related to the PDO? Lines 493-495). Again, if this is the primary point of the paper, then substantially more work needs to be done to address why precipitation, for example, increases in the Himalaya in the Pliocene. If this was the point of the paper, it would obviate the need to compare with other model simulations (see paragraph above), but would then require substantially more work to identify how various atmospheric phenomenon change through time. A generalized description of changes in the past is not particularly useful (most of this information can be communicated fairly effectively with figures), so addressing the causes of these changes (or comparison with proxies—see below) is worthwhile.

Parts of the discussion showcase comparisons with terrestrial proxy data, though this is limited to citing previous work and stating that there is general agreement with previous, proxy-model work. If a proxy-model comparison is the point of the paper, then more work needs to be done actually compiling the proxies and doing a proper statistical test to see if there is agreement between modelled precip/temperature changes in each of these orogens and existing data. This would be a useful contribution to the literature, but, again, would require substantially more work.

Much of the motivation for the manuscript appears to be to understand how climatic changes may change denudation/geomorphic analyses, but this is done in only a superficial way. If this is the point of the paper, then, again, much more work needs to be done, rather than simply stating that erosion depends on climate (lines 416-423 are a good example of statements that serve to motivate a paper, but don't provide any actual analysis). For example, can the authors take some of the climate model output and, given a potential 2000 mm/yr change in precipitation in the Himalaya since the Pliocene, actually re-interpret some of the existing exhumation/denudation data? If not, why not? What additional data is necessary? And if such a reanalysis isn't possible, then how does knowledge of such a change in precipitation facilitate future work?

I agree with the generalized statements made throughout the paper (i.e., that denudation and landscape evolution depend on climate), but these are somewhat self-evident and, as currently written, the manuscript does not make a fundamental contribution to improving our knowledge on this subject except to state that climate changed in the past. Assumptions of stationarity are indeed a problem in interpreting modern datasets that have a component of geologic history, but a really nice contribution of this paper would be to show how these assumptions can be mitigated when one knows the history of precipitation or temperature.

Because ESurf is not a climate modeling journal, more discussion needs to be given as to the limitations of ECHAM5 in a way that Earth surface process folks can understand. For example, what are the uncertainties associated with simulating orographic precipitation? Though T159 is high-resolution, it still requires substantially smoothing topography, which presumably introduces some uncertainty in to the results. What uncertainties are associated with the PRISM reconstruction? (on a side note, which PRISM reconstruction is used? PRISM3D? PRISM2? I mention this because the topographies between different PRISM reconstructions are substantially different.)

I found the use of the cluster analysis to be not intuitively helpful. If the authors want to keep using it, then the authors need to at least walk the reader through an example of how to understand Figures 6, 9, 12, and 15. Is C1 always the same climate zone in each figure and in each time-slice? If so, why are different colors used? Why don't the authors use something more intuitive, like Köppen's climate classification scheme (Peel et al., 2007) to classify climates? As best I understand it, the clustering analysis is used to show the spatial extent of a given climate in a given time-slice and in a given location, but it's not clear how one should interpret these results.

Minor Comments: Are the topographies for any of these ranges modified at all (it's unclear from the PRISM reference (Line 131), whether this has been done).

Lines 57-59: "Cold-temperature island" is not a climatic term in widespread use. What

precisely do you mean? Also, Boos and Kuang, 2010 specifically refute the idea that Plateau surface elevation matters for the South Asian Monsoon and rather focus on the Himalaya instead.

Line 61: Zhisheng et al. (2001) don't actually present any new geological data. Instead, it is all from cited literature. The focus of their study was GCM results. Dettman et al. (2003) is not the only study to look at this. Please see the following studies (which are just a sampling): (Caves et al., 2017; Kent-Corson et al., 2006; Lechler et al., 2013; Lechler and Niemi, 2011; Licht et al., 2016; Methner et al., 2016; Mulch et al., 2015, 2008; Pingel et al., 2016).

Lines 77 and 78: "documenting the magnitude" appears twice.

Lines 96-97: Though, importantly, several recent studies have run ECHAM5 at a higher resolution ((Feng et al., 2016; Feng and Poulsen, 2016).

Lines 102-103: This statement is somewhat odd, since the authors are specifically investigating climatological changes over mountain ranges, where resolution typically tends to matter.

Lines 114-133: For all simulations, stating the pCO2 used in the experiment would be most helpful, particularly since it won't take up much room. Also, how is the land-surface treated? For example, the authors state that they are using vegetation reconstructions, but it's unclear if this is then being fed into a "built-in" land-surface model or if they are explicitly using JSBACH.

Lines 126: "for the" used twice.

Lines 203-204: Changes in Greenland and Antarctica are almost certainly unreliable. Because PRISM uses a reconstructed ice-sheet extent, changes in temperature in Greenland and Antarctica are almost certainly reflecting the imposed boundary condition, which itself has quite a bit of uncertainty. It's hard to get around this, except to note that the change in temperature is entirely dependent upon the ice-sheet boundary

condition (see discussion in and of de Boer et al. (2015)).

Lines 416-423: Are runoff changes in these models coupled to precipitation changes? In all cases, does P–E (precipitation minus evaporation) scale with changes in precipitation. I'm not particularly familiar with JSBACH (presuming this is the land-surface model used), but if it has a $CO_2$ fertilization parameterization, then runoff may be decoupled from precip. Some of these erosion processes may depend more on runoff than precip.

Figure 1: Would be nice to also plot the topography of the St. Elias range and the Cascades.

Figure 7b-Precip-PLIO: Why does precipitation appear to follow a wave-like pattern over tropical South America? Is this due to the spectral nature of ECHAM5?

References used in review: An, Z., Kutzbach, J.E., Prell, W.L., Porter, S.C., 2001. Evolution of Asian monsoons and phased uplift of the Himalaya-Tibetan plateau since Late Miocene times. Nature 411, 62–66. doi:10.1038/35075035

Caves, J.K., Bayshashov, B.U., Zhamangara, A., Ritch, A.J., Ibarra, D.E., Sjostrom, D.J., Mix, H.T., Winnick, M.J., Chamberlain, C.P., 2017. Late Miocene uplift of the Tian Shan and Altai and reorganization of Central Asia climate. GSA Today 27, 19–26.

Chandan, D., Peltier, W.R., 2017. Regional and global climate for the mid-Pliocene using CCSM4 and PlioMIP2 boundary conditions. Clim. Past 13, 919–942. doi:10.5194/cp-2017-21

de Boer, B., Dolan, A.M., Bernales, J., Gasson, E., Goelzer, H., Golledge, N.R., Sutter, J., Huybrechts, P., Lohmann, G., Rogozhina, I., Abe-Ouchi, A., Saito, F., van de Wal, R.S.W., 2015. Simulating the Antarctic ice sheet in the Late-Pliocene warm period: PLISMIP-ANT, an ice-sheet model intercomparison project. Cryosph. 9, 881–903. doi:10.5194/tc-9-881-2015

Dettman, D.L., Fang, X., Garzione, C.N., Li, J., 2003. Uplift-driven climate change at

12 Ma: a long $\delta$18O record from the NE margin of the Tibetan plateau. Earth Planet. Sci. Lett. 214, 267–277. doi:10.1016/S0012-821X(03)00383-2

Feng, R., Poulsen, C.J., 2016. Refinement of Eocene lapse rates, fossil-leaf altimetry, and North American Cordilleran surface elevation estimates. Earth Planet. Sci. Lett. doi:10.1016/j.epsl.2015.12.022

Feng, R., Poulsen, C.J., Werner, M., 2016. Tropical circulation intensification and tectonic extension recorded by Neogene terrestrial d18O records of the western United States. Geology 44. doi:10.1130/G38212.1

Haywood, A.M., Hill, D.J., Dolan, A.M., Otto-Bliesner, B.L., Bragg, F., Chan, W.L., Chandler, M.A., Contoux, C., Dowsett, H.J., Jost, A., Kamae, Y., Lohmann, G., Lunt, D.J., Abe-Ouchi, A., Pickering, S.J., Ramstein, G., Rosenbloom, N.A., Salzmann, U., Sohl, L., Stepanek, C., Ueda, H., Yan, Q., Zhang, Z., 2013. Large-scale features of Pliocene climate: Results from the Pliocene Model Intercomparison Project. Clim. Past 9, 191–209. doi:10.5194/cp-9-191-2013

Jiang, D., Lang, X., 2010. Last glacial maximum East Asian monsoon: Results of PMIP simulations. J. Clim. 23, 5030–5038. doi:10.1175/2010JCLI3526.1

Jiang, D., Lang, X., Tian, Z., Ju, L., 2013. Mid-Holocene East Asian summer monsoon strengthening: Insights from Paleoclimate Modeling Intercomparison Project (PMIP) simulations. Palaeogeogr. Palaeoclimatol. Palaeoecol. 369, 422–429. doi:10.1016/j.palaeo.2012.11.007

Kent-Corson, M.L., Sherman, L.S., Mulch, A., Chamberlain, C.P., 2006. Cenozoic topographic and climatic response to changing tectonic boundary conditions in Western North America. Earth Planet. Sci. Lett. 252, 453–466. doi:10.1016/j.epsl.2006.09.049

Lechler, A.R., Niemi, N.A., 2011. Sedimentologic and isotopic constraints on the Paleogene paleogeography and paleotopography of the southern Sierra Nevada, California. Geology 39, 379–382. doi:10.1130/G31535.1

Lechler, A.R., Niemi, N. a., Hren, M.T., Lohmann, K.C., 2013. Paleoelevation estimates for the northern and central proto-Basin and Range from carbonate clumped isotope thermometry. Tectonics 32. doi:10.1002/tect.20016

Licht, A., Quade, J., Kowler, A., Santos, M. de los S., Hudson, A., Schauer, A., Huntington, K., Copeland, P., Lawton, T., 2016. Impact of the North American Monsoon on isotope paleoaltimeters: Implications for the paleoaltimetry of the American Southwest. Am. J. Sci. 317.

Methner, K., Fiebig, J., Wacker, U., Umhoefer, P., Chamberlain, C.P., Mulch, A., 2016. Eo-Oligocene proto-Cascades topography revealed by clumped ($\Delta$47) and oxygen isotope ($\delta$18O) geochemistry (Chumstick Basin, WA, USA). Tectonics 35, 546–564. doi:10.1002/2015TC003984

Mulch, A., Chamberlain, C.P., Cosca, M.A., Teyssier, C., Methner, K., Hren, M.T., Graham, S.A., 2015. Rapid change in high-elevation precipitation patterns of western North America during the Middle Eocene Climatic Optimum (MECO). Am. J. Sci. 315, 317–336. doi:10.2475/04.2015.02

Mulch, A., Sarna-Wojcicki, A.M., Perkins, M.E., Chamberlain, C.P., 2008. A Miocene to Pleistocene climate and elevation record of the Sierra Nevada (California). Proc. Natl. Acad. Sci. U. S. A. 105, 6819–6824. doi:10.1073/pnas.0708811105

Peel, M.C., Finlayson, B.L., McMahon, T.A., 2007. Updated world map of the Koppen-Geiger climate classification. Hydrol. Earth Syst. Sci. 11, 1633–1644.

Pingel, H., Mulch, A., Alonso, R.N., Cottle, J., Hynek, S.A., Poletti, J., Rohrmann, A., Schmitt, A.K., Stockli, D.F., Strecker, M.R., 2016. Surface uplift and convective rainfall along the southern Central Andes (Angastaco Basin, Argentina). Earth Planet. Sci. Lett. 440, 33–42. doi:10.1016/j.epsl.2016.02.009

Zhang, R., Yan, Q., Zhang, Z.S., Jiang, D., Otto-Bliesner, B.L., Haywood, a. M., Hill, D.J., Dolan, a. M., Stepanek, C., Lohmann, G., Contoux, C., Bragg, F., Chan, W.-L.,

Chandler, M. a., Jost, a., Kamae, Y., Abe-Ouchi, a., Ramstein, G., Rosenbloom, N. a., Sohl, L., Ueda, H., 2013. East Asian monsoon climate simulated in the PlioMIP. Clim. Past Discuss. 9, 1135–1164. doi:10.5194/cpd-9-1135-2013
* * *

---

## Editor Comment (EC1) · D. L. Egholm (Editor) · 17 Nov 2017

First, I would like to thank the authors and the reviewers for their contributions to ESurf Discussions. I suggest that the authors carefully consider the comments made by the two reviewers and provide point-by-point responses to them. I also recommend that the authors use the many constructive suggestions made by the reviewers to revise the manuscript.

---

## Author Comment (AC1) · 14 Jan 2018

**Response to RC1**
(responses in blue)

Mutz et al. look to use paleoclimate GCMs to identify drivers of past geomorphic change. This is a topic for which I hold great interest, and I feel that the authors have crafted a very useful set of model results that they leave underutilized. As such, I feel inclined to accept the paper on the basis of the useful results, but to request major revisions such that they do their own work justice.

My major concerns, which will become clear in the line-by-line comments (please feel free to respond to similar comments en masse) are as follows: 1. The paper is motivated by denudation rates and landscape evolution, but really includes this as a speculative wrapper that is not substantiated. I suggest that instead you propose testable hypotheses surrounding your findings.
2. Related to #1, much of the text is a litany of "temperature was X here ...". I find such statements of results useful only insofar as they expand upon a figure (associated with a supplementary data set) that presents the results. While these sections are written clearly, I would suggest that the authors focus on a set of geomorphic questions (if this be their motivation) and how the model-data set informs those questions.
3. Many of the discussions of model results are of ice-covered regions, yet no consideration of direct glacial erosion is given. Furthermore, no reference to the changes in the statistics of discharge or catchment area in ice-covered regions is given. This seems a disservice to this relatively high-resolution paleoclimate AGCM: the geologic setting *must* be considered, otherwise it seems that the authors' pushing on the modeling end has not been matched by a simple geological history sanity check. I would suggest that either significantly glaciated regions and the catchments that they feed be masked out, or that glacial erosion and its associated processes be included in the discussion.
4. (Discussed only here): You have not compared your models against any data. I understand that this may be simply a modeling exercise that you do compare to other models. However, I think that such a comparison could assuage skepticism about your results and lend support to your case, especially if you include it as part of a local case study (see the third point below). The core of these three points is that, with a bit more care, I think your results could say something really useful to the geomorphic community. Currently, the paper seems to be more a statement of, "this is important to geomorphology", followed by a long list of the model results. I challenge the authors to demonstrate (rather than simply stating) the importance of their work to geomorphology in a way that includes how it may impact the way scientists view Quaternary landscape evolution. Ideas include:
• Changes in means (done)
• Changes in statistical distributions of temperature and precipitation – think extreme events, frost-cracking window, etc.
• A focus on a few iconic regions while *explicitly* ignoring significantly ice-covered domains (I think this would be easiest, though obviously would be thrilled if you decided to tackle glacial processes)
• Using this focus to build a template for how to use paleoclimate GCM outputs to advance the field of geomorphology. Currently, I think that the work is acceptable following changes for internal consistency and geological accuracy (see #1 and #3), but I think that you could be selling yourselves short if you don't dig just a tiny bit deeper to investigate your forcings and their impact on geomorphology. I hope that you find these comments helpful in continuing to craft an insightful piece of work out of what seems to be a strong modeling approach.

We thank Prof. Andrew Wickert for his highly valuable review of our manuscript. Many important points were raised in the review and we hope that our appreciation for the input is sufficiently reflected in the revisions we made in response to it. We also encourage him to see our response to the second reviewer, where we provide additional geologic relevance of this study by now including a comparison of available terrestrial proxy data to our model results. We also explain throughout our response to the 2[nd] reviewer why an application of the predicted climate change to predict denudation rate changes is a large undertaking that can not be meaningfully conducted in this paper, but warrants more detailed applications of the models to individual areas (a topic of ongoing work/application for us). We refrain from using the model predicted runoff in the global GCM (even though it's conducted at relatively high resolution compared to previous work) to calculate changes in fluvial incision. This would be better done by mapping the predicted precipitation changes onto higher resolution (<90 m) DEMs and solving the kinematic wave equation for each fluvial erosion in each catchment, for the changes in precipitation. However, as we repeatedly mention above, this is not possible to include in this manuscript without first characterising how the precipitation has changed in each region (the current manuscript goals). Work in progress we are conducting is trying to apply the kinematic wave equation and palaeoprecipitation to selected areas, but it's proving difficult to implement meaningfully without temporally continuous (e.g. LGM to present) simulations of precipitation change. We hope this brings to the reviewers attention the complications associated with doing full erosion history calculations based on these results. We have expanded the last paragraph in the instruction to convey the above perspective better, and more clearly articulate (and justify) the scope and limitations of the manuscript.

We appreciate the importance of addressing specific sets of geomorphic questions and hypotheses (1 & 2) and we are currently taking an in-depth look at quantifying the potential for erosion by a variety of processes. These include different methods of quantifying frost cracking intensity and extreme precipitation events and how these changed over time. However, in order to include those in this manuscript, we fear that would have to seriously compromise the thoroughness with which we investigate these questions at the moment. Instead, we hope that we can convey the usefulness of our consistently set up palaeoclimate simulations as a framework for addressing any of these particular questions in detail, and have modified sections of this manuscript accordingly. This includes, but is not restricted to, extensive compilations of proxy-based precipitation reconstructions for our two larger study sites (South Asia and western South America) and comparison of this data to our model output. With this, we hope that we were also able to address the concerns raised in point 4. In order to address the important point raised about glaciated areas (3), we added an ice cover layer on all of our difference plots, included global maps of ice extent (as used for our simulations) in the supplementary material, and discussed where the large differences in temperature and precipitation we highlight in the manuscript are accompanied by changes in ice cover. Thus we hope to prevent that interpretations of the implications of our results are made without consideration of changes in ice cover (and consequently shifts in the process domain).

Line-by-line:
23. US Pacific Northwest Pacific → drop second "Pacific"
This has been corrected. Thank you for catching that.

29. future observational studies interested in quantifying → future observational studies that quantify (studies can't be interested in things, strictly speaking)
That is right of course. It has been corrected as suggested.

53. orogen scale → orogen-scale
It has been corrected as suggested.

∼57. A couple of recent studies from the climate science community shed light on the impacts of the Andes (first ref below) and continents in general (second ref below). In case these are interesting to you, I'm pasting the bibliographic information here:
Maroon, E. A., D. M. W. Frierson, and D. S. Battisti (2015), The tropical precipitation response to Andes topography and ocean heat fluxes in an aquaplanet model, J. Clim., 28(1), 381–398, doi:10.1175/JCLI-D-14-00188.1.
Maroon, E. A., D. M. W. Frierson, S. M. Kang, and J. Scheff (2016), The precip itation response to an idealized subtropical continent, J. Clim., 29(12), 4543–4564, doi:10.1175/JCLI-D-15-0616.1.
Many thanks for the references. These are indeed of much interest to us and we included them in the revised manuscript.

73-75. "Furthermore, recent controversy exists concerning the spatial and temporal scales over which geologic and geochemical observations can record climate-driven changes in weathering and erosion [e.g. Whipple, 2009; von Blanckenburg et al., 2015; Braun, 2016].": I see that you do not return to this point later, so could you describe the controversy for those who are not familiar with it?
Thank you for pointing this out. We described the controversy briefly for those unfamiliar with it after the sentence quoted above.

81. I see that later you discuss a little about what an AOGCM may do, but I will be looking for justification about how an AGCM may suffice. Is this in part because you prescribe the b.c.'s and you are running it for 17 years only? If so, could you discuss potential systematic variations between this and an AOGCM?
We prescribe sea surface temperature reconstructions (SSTs) boundary conditions, which allows us to bypass the computationally expensive coupled simulations. Because these are fixed climatologies (though with seasonality preserved), the simulation of fewer years suffices. As a consequence, however, we do not expect to see decadal scale variability as we would in case of coupled models or prescribed SSTs that vary from year to year (such as present day simulations using AMIP SST's). We discuss this in the revised manuscript.

89. "PLIO to the Last Glacial Maximum": as you include no time-slices between these, I suggest making these part of the list and dropping the "to the".
This has been corrected as suggested.

147. "This section describes the clustering method used in this study." You could drop this sentence – the section title should be enough for even an inattentive reader!
We followed this suggestion and dropped that sentence.

176-178. I was wondeirng how you picked the number of clusters: I am glad to see that you performed a thorough search.
Thank you. We added some text explaining this. We systematically increased the number of clusters from 3 to 10 and assessed the distinctiveness or similarities of resulting climate clusters. Once the increase in the number of cluster no longer resulted in the addition of another cluster that was distinctly different from the others, we used this as a cut off point and used the cluster number of the previous iteration as the optimal cluster number.

Section 3: Much of this is information that I find better communicated through figures than with text. It is clearly written, however, and I am reluctant to suggest a rewrite for brevity in a length-unconstrained journal so long as the text can be co-located with the figure.
190-192. I see you have another "This section describes..." sentence. If this is your preferred way to write, you may keep it; here, the second sentence is not such a good topic-sentence replacement.
Thank you. We kept the sentence in this instance as it also immediately draws attention to the relevant figures, which may also serve in addressing the previous point you raised.

197-198. i.e. over the ice sheets. (This applies to other regions as well, and should be important to point out if you are going to then discuss fluvial processes in orogens)
Many thanks for pointing this out. We are more mindful of this in the revised discussion.

203-paragraph: Also because of local ice loss, presumably. So I think that the two prior paragraphs could have a new summary that "The greatest changes in temperature is observed where the greatest change in local ice extent occurs."
Thank you. We followed your suggestion.

214-215. Have you considered discussions of the African Humid Period?
We had not considered discussion of precipitation changes in North Africa, since it lies outside the regions we focus on. However, we appreciate that Holocene precipitation changes in the region are important and may be of interest to many readers. We therefore included a short discussion of Holocene precipitation changes in Northern Africa in the revised manuscript.

373-374. If you are looking at the influence of temperature and precipitation on erosion, and you are not including subglacial erosion, then your preceding text must indicate where your changes really are indicative of ice extent – both as a separate process domain and as a driver of fluvial processes and potential changes in the statistics of river discharge.
Thank you. We are more mindful of this in our discussion.

Section 4.1. Your first paragraph (weathering) differs from the content (comparing your model results with those published). These should be in different subsections, and the weathering paragraph may need to be expanded. Your "weathering and erosion" paragraph also neglects direct effects of glaciers, ice caps, and ice sheets, which were globallly significant.

Thank you for this suggestion. We re-structured this section as suggested and took ice extent into consideration in the erosion section.

Section 4.2. Once again, your discussion is often of formerly (or currently) ice-covered regions without explicitly acknowledging that this is a different process domain. In addition, as with the previous section, the body paragraphs are mostly about model comparison and regional changes with sparse link to the landscape-evolution factors indicated in the topic paragraph.
As above, we restructured this section, took ice extent into account and chose a more fitting section title.

416-423. Please discuss the direct influence of glaciers on the erosion orogens in the context of changing precipitation (and therefore mass balance). Is it significant or not?
Although it is challenging to sufficiently quantify changes in glacier-related erosion due to differences in precipitation, we now include this point in our discussion.

433-434. "Coastal North America"? Doesn't look like it: seems to be most of NA south of the ice margin.
That is correct. We revised our descriptions accordingly.

Section 4.3. The authors describe the results here, but I find the connection to erosion rates to be insufficiently described compared to how they are highlighted in the topic sentence, as well as in the abstract. I would like you to go one step beyond "ought to be considered" and actually posit how you expect the erosion rates – and therefore, the balance between erosion and exhumation and perhaps the equilibrium shapes of the mountains and their rivers – to vary. Otherwise, you are suggesting future work rather than actually describing the possible geomorphic significance – and I think underutilizing your results in a paper that is clearly targeted towards geomorphologists.
As described in our response above, we believe that trying to address specific problems such as these or quantifying how differences would be expressed as erosion rates would be beyond the scope of this manuscript and come at the cost of not being able to address these as thoroughly as we are currently attempting in other ongoing work.

498. "which may favour frost driven weathering during glacial climate states" – the St. Elias range was covered by glaciers! Yes, there can be some frost-cracking around the ice, but don't you think this is important too? http://instaar.colorado.edu/groups/QGISL/ak_paleoglacier_atlas/gallery/index.html
Thank you for drawing our attention to this. We consider this in our discussion now and revised the manuscript accordingly.

508. "enhanced sediment production driven by frost processes" – same as above. Glaciers were there. Consider them.
As for the comment above, we also considered glaciers here in the revised manuscript.

Conclusions: Comparison to other models: is this match surprising or no? Did you (mostly) use the same inputs and simply increase the grid resolution? If so, could you comment on how the improved grid and possible variations in inputs and use of them ocean as a boundary condition may have affected (or not) your results as compared to those of earlier studies? This would be more useful to include in the discussion than a simple list of "Our temperature in place Y was $T_0$, and X et al. wrote that they found it was $T_1$, which is close to $T_0$. Think big-picture, in both process and numerics!
Due to model-specific parameterisation, deviation is possible. In the revised manuscript, we comment on this as well as on the model resolution and implications of using ocean as boundary conditions instead of an ocean model.

533. Did your 8-10 degC changes occur significantly over areas that would be affected by hillslope or fluvial processes? (i.e. unglaciated areas?)
Some unglaciated areas experience large differences in temperature, but the maxima of 8-10 degC geographically coincide with ice cover changes. We acknowledge and discuss this in the revised manuscript.

[revised manuscript text omitted]

Figure 1

[Figure]

Figure 2

[Figure]

Figure 3

[Figure]

Figure 4

[Figure]

[Figure]

Figure 5

[Figure]

Figure 6

[Figure]

**(a) PI**

MH   LGM   PLIO

(b)

**Annual Temperature (°C)**
-30  -15  0  15  30

colder than PI        warmer than PI
-12    6    0    6    12
Degrees Celsius (C°)

**Annual Precipitation (mm/year)**
800  1600  2400

drier than PI        wetter than PI
-2400   -1200   0   1200   2400
mm/year ice cover

Figure 7

[Figure]

Figure 8

Andes Climate Clusters

Climate Characterization

PI

MH

LGM

PLIO

Figure 9

[Figure]

PI

Annual Temperature (°C)

Annual Precipitation (mm/year)

(b)

MH

LGM

PLIO

colder than PI          warmer than PI

drier than PI          wetter than PI

Degrees Celsius (C°)

mm/year ice cover

Figure 10

Mean Annual Precipitation

Mean Annual Temperature

Figure 11

[Figure]

[Figure]

Figure 12

[Figure]

Figure 13

[Figure]

[Figure]

Figure 14

[Figure]

Figure 15

[Figure]

Figure 16

[Figure]

Figure 17

[Figure]

Supplemental Figure S1

---

## Author Comment (AC2) · 14 Jan 2018

**Response to RC2**

(responses in blue)

The authors simulate the pre-industrial, mid-Holocene, LGM, and Pliocene climates using the ECHAM5 general circulation model. The motivation is to understand how past climatic states may change regional climatologies, particularly over mountainous areas that have been the focus of much erosion and geomorphic work. The authors find that past climatic states (particularly the LGM and Pliocene) do produce changes in absolute temperature and precipitation and in the annual ranges of these two climatic variables. Overall, I did not find the manuscript to be a particularly useful addition to the literature. Though the motivation is potentially novel, the analysis is not complete and needs substantially more work. Most of the manuscript focuses on simply describing climatic changes, while neglecting novel analyses. Consequently, at this point, I recommend reject, though perhaps with substantial work (including more modeling, comparison with existing models, and/or comparison with data), it may become publishable in the future in Esurf.

We thank the reviewer for their constructive feedback on our manuscript. We found the reviewer's comment very useful in the improvement of this manuscript and hope that the modifications we made do the reviewer's thorough review justice. Most importantly, we followed the reviewer's suggestion to deepen the comparison of our model simulation results with data, specifically proxy-based reconstructions, as we believe this suggestion is of most relevance to the earth surface science community targeted by ESurf. While it was not possible to create a complete overview of all proxy-based studies everywhere and for all time periods, we included an extensive compilation of site specific reconstructions of precipitation for our largest areas (South America and Tibet). In prioritizing the compilation of precipitation reconstructions, we also hope to address the reviewer's concerns about the GCM's performance regarding this variable, and are happy to report that the model shows satisfactory to good performance in predicting the direction of changes in MH and LGM.

First, the authors need to decide what the point of the paper is. Most of the paper reads as a description of climatological changes for 3 periods in the geologic past (MH, LGM, and Plio) (indeed, most of the text is written this way). However, this work has already been done, most prominently by the PMIP and PlioMIP set of model intercomparisons. What does this manuscript offer that these model intercomparisons have not already analyzed? A case could be made that these model intercomparisons are typically of a global nature (though there has been some work on changes in the Asian Monsoon systems using both PlioMIP and PMIP results (Jiang et al., 2013; Jiang and Lang,2010; Zhang et al., 2013)), so that the analysis of the orogens in this study is useful. However, given that GCMs have difficulty simulating precipitation and in particular simulating precipitation over complex topography, the usefulness of simply describing changes over the Himalaya, Andes, Cascades, and St. Elias ranges is somewhat muted. For example, why should readers believe that ECHAM5 produces reasonable results over the St. Elias range? Why not use (or at least compare with) the exist- ing model intercomparisons to look at changes in these locations? Most of the PlioMIP simulations are of a lower resolution than the model simulations presented here, though not all (Haywood et al., 2013). Many of the newer PlioMIP2 simulations are being run at a higher resolution and permit at least some comparison with the data here (Chandan and Peltier, 2017). Having myself tried to access PlioMIP data, I understand that it can be difficult to get access to the PlioMIP output, but if the point of the paper is to quantify actual climatological change, then comparison with other models is a must (or at least a thorough treatment of possible boundary condition uncertainties and additional ECHAM5 model runs to establish the sensitivity of precipitation/temperature in these areas); otherwise, we have no reason to believe that ECHAM5 presents anything resembling a proper picture of climatic change in the past.

**Summary Response**: We appreciate this assessment and hope to have enhanced the usefulness of the manuscript with modifications we've made throughout the results and discussion sections. While GCM simulations for the time slices we chose already exist and individual studies were also conducted at a relatively high resolution, we believe (contrary to the reviewer's suggestion) that the usefulness of our results lies in both the high resolution combined with the consistency in model choice, resolution, output frequency (and methods of descriptive statistics). We emphasise this point in the revised manuscript. We acknowledge in the text (methods section 2.1 and introduction) the shortcomings of GCMs in predicting orographic precipitation in the discussion and in the revised text now compare the simulated precipitation with readily available proxy-based reconstructions for specific locations in Tibet and South America where we were able to compile data (this was a large undertaking). The revised manuscript now demonstrates the shortcomings of GCMs and provides an additional data compilation that ESurf readers interested in palaeoclimate effects on denudation may find useful. The changes we have made to address this reviewers suggestions are contained in the revised introduction, and extensively throughout the Discussion section (4.0), as well as the addition of two new (concluding figures).

**In More Detail:** While we appreciated the reviewer's comments, we note that what he/she is asking for in the above comments is a manuscript that either focuses on an inter-model comparisons, or a comparison to proxy data. We see merits in both, but both are not possible in a single manuscript. An inter-model comparison is not really well suited to the aims of this journal and our target community to bring attention to the magnitude of palaeoclimate changes that have occurred in different active orogens. Our intended audience is not the palaeoclimate modelling community (and journals associated with it), but rather the surface processes community. We specifically chose this journal to provide this community with palaeoclimate predictions that may be of interest to them. In our own experience in trying to interpret palaeodenudation rates from data we produce, our first goal is always trying to find predictions or observations of climate change for a region. Thus, we set out to write this paper to provide what we find in our research as the first, most useful, step in understanding the surface process history of a region. Furthermore, one major aspect of this study that we want to retain is the statistical analysis of the climatology of each time slice as this is what is most useful for the ESurf community in terms of knowing if a region they are working is has experienced a signal change in climate through time.

Thus, we are left with the conundrum that what the review requests and we want would normally be three full manuscripts that include: 1) a statistical analysis of climatological changes over active orogens (our previous focus of the text), or 2) a comparison of a suite of similarly set up and common code (ECHAM5) simulations of palaeoclimate time slices to other GCM models (one of the suggestions by this reviewer), or 3) a model-data comparisons for model evaluation at the time slices investigated (also suggested by this reviewer).

***In an attempt to hopefully reach a compromise with this reviewer – the revised manuscript is now structured around points 1 and 3 above (statistical analysis of climatological changes, and model-data comparison where possible).*** These changes have been manifested in a now expanded discussion section, and through the addition of model-data comparison figures at the end of the text (see section 4.1.2 , 4.2.2, and 4.4). These changes hopefully reach a happy middle-ground between our aims and the reviewer's suggestions. The changes have significantly expanded the manuscript, and we hope that the reviewer also recognizes that implementation of all the suggested changes is not possible within a single manuscript of typical length for this journal. The revised manuscript now has 18 figures (including 1 in the supplemental section) and 21 pages of manuscript text (not including references / captions).

Some parts of the paper address actual causes for climatic change (for example, discussion of the Pacific North American Teleconnection (no citation given). Is this related to the PDO? Lines 493-495). Again, if this is the primary point of the paper, then substantially more work needs to be done to address why precipitation, for example, increases in the Himalaya in the Pliocene. If this was the point of the paper, it would obviate the need to compare with other model simulations (see paragraph above), but would then require substantially more work to identify how various atmospheric phe- nomenon change through time. A generalized description of changes in the past is not particularly useful (most of this information can be communicated fairly effectively with figures), so addressing the causes of these changes (or comparison with prox- iesâ˘T˘ see below) is worthwhile.

Parts of the discussion showcase comparisons with terrestrial proxy data, though this is limited to citing previous work and stating that there is general agreement with previous, proxy-model work. If a proxy-model comparison is the point of the paper, then more work needs to be done actually compiling the proxies and doing a proper statistical test to see if there is agreement between modelled precip/temperature changes in each of these orogens and existing data. This would be a useful contribution to the literature, but, again, would require substantially more work.

We address the suggestions in the previous 2 paragraphs in the following ways: **First,** as was mentioned earlier, we maintain that for this journal and the community that reads it there is value in demonstrating climate change events in commonly studied active orogens where denudation studies (sensitive to climate change) are conducted. This requires that we maintain our current statistical characterisations of the climate change. With this approach, it then becomes intractable to focus on the climate dynamics associated with change in each region because typically a thorough investigation of the climate dynamics associate with change in each region could be a paper for each region in itself, and the broader picture of change in a range would be missed. We try to meet the reviewer half way in this suggestion by including additonal comments on possible causes for climatic change in different regions.

The PNA (citation now included) is related to the PDO, and Alaskan precipitation and temperature is to some degree controlled by it. In order to properly address these questions, however, further analyses looking at dominant atmospheric variability modes, trajectories and other aspects of atmosphere would have to be carried out, which are beyond the scope of this study. We have modified the manuscript in text at the end of section 2.1 to clarify (and justify) why we do not conduct an inter model comparison, as well as some of the caveats associated with our approach.

However, (**second**), we like the reviewer's suggestions of adding additional comparisons to proxy data, and we heavily invested time in compiling this information in this revised version for locations where spatially distributed data are available for the time slices available.
Unfortunately, changes from proxy studies are often reported in terms of relative changes compared to modern (e.g., wetter/drier, warmer/colder) and are sometimes contradictory for the same location, making the application of otherwise suitable statistical tests (e.g. t-test and even non-parametric tests) difficult. There are, in some cases (see final figures in paper), even contradictions between the proxy data themselves in neighbouring locations, a reality often underappreciated by the surface processes community. Furthermore, terrestrial proxy data are not available for all the time slices and locations we investigate. This is particularly true for the Pliocene time slice. **With these limitations of available proxies, we do the best we can to compare available observations to proxy data. These comparisons are now provided in revisions to the discussion section, and the addition of 2 figures at the end of the manuscript**.

We note that for the Alaska and Cascadia (NW USA) study areas, which are heavily and repeatedly glaciated, that limited - to no proxy data are available (due to poor preservation of proxies over glacial-interglacial cycles). Although some marine proxy records and records of past glaciations have been published for these regions, they are not useful for our purposes because they do not record if terrestrial locations in the regions were wetter/drier or warmer/colder during our time slices. Marine proxy records are of limited use for comparison to the terrestrial changes investigated in this study. Thus, despite the lack of terrestrial proxy comparisons available for two of the regions (Alaska, Cascadia), we maintain that presentation of the model predicted changes for these data poor regions is extremely useful because the model predictions augment existing terrestrial data gaps, and provide a starting point for future studies to formulate testable hypotheses of climate change and potential denudation impacts.

Much of the motivation for the manuscript appears to be to understand how climatic changes may change denudation/geomorphic analyses, but this is done in only a superficial way. If this is the point of the paper, then, again, much more work needs to be done, rather than simply stating that erosion depends on climate (lines 416-423 are a good example of statements that serve to motivate a paper, but don't provide any actual analysis). For example, can the authors take some of the climate model output and, given a potential 2000 mm/yr change in precipitation in the Himalaya since the Pliocene, actually re-interpret some of the existing exhumation/denudation data? If not, why not? What additional data is necessary? And if such a reanalysis isn't possible, then how does knowledge of such a change in precipitation facilitate future work?

This is a great suggestion and we are indeed doing this as ongoing work. But what the reviewer is asking for here simply cannot be included in this paper without making it an extremely long manuscript. For example, in the surface processes community, comparisons of river profiles to modern precipitation, or temperature changes to frost cracking histories for **one** catchment / study area (e.g. <100x100 km2) is typically a full manuscript in itself (e.g. Marhsall et al., 2015 Science Advances; Schaller et al. 2004 Journal of Geology). So, we respectfully disagree with the reviewer that this should be done within this manuscript. It's simply not possible to do in a way that would be convincing to the surface processes community.

I agree with the generalized statements made throughout the paper (i.e., that denudation and landscape evolution depend on climate), but these are somewhat self-evident and, as currently written, the manuscript does not make a fundamental contribution to improving our knowledge on this subject except to state that climate changed in the past. Assumptions of stationarity are indeed a problem in interpreting modern datasets that have a component of geologic history, but a really nice contribution of this paper would be to show how these assumptions can be mitigated when one knows the history of precipitation or temperature.

We agree with the reviewer that it is "self-evident" that palaeoclimate can impact the denudation history of an orogen, but how many palaeodenudation rate studies actually make a comparison to spatial distributed predictions of palaeoclimate? Very few in our experience, and we cite the robust ones in the manuscript. What most palaeodenudation rate studies do is compare some set of observation to the nearest set of proxy observations, but this approach does not provide a sense of spatial variability in terrestrial climate change that models predict (furthermore, proxy observations are often far removed from the denudation rate observations). In our manuscript, we specifically avoid picking on previous publications that follow this 'conventional' approach to show how this commonly used approach ignores spatial variability in climate change. We don't think this is a productive way to advance the science. Instead, we maintain that providing spatially continuous model predictions at different time slices in a self-consistent set of simulations is extremely useful for documenting the magnitude of climate change (which is not self-evident), and for formulating testable hypotheses for future work and to identify where the best locations are in the world to investigate climate change impacts on a regions denudation history. Thus, the way in which our research group usually conducts palaeodenudation rate studies is to first run model simulations to formulate testable hypotheses, and then to investigate/test these hypotheses with field studies and geochemical observations of denudation histories. This approach to investigating palaeoclimate-palaeodenudation interactions is definitely not self-evident in the literature in our opinion.

We very much agree that in order for our simulations to ultimately be useful in actually quantifying denudation much more work needs to be done. Translating any of the changes observed here (or by other studies) to erosion rates remains a big problem in the Earth surface science community. It is one that we are actively working on and includes the application and comparison of different models for quantifying frost cracking [Anderson et al., 1998, Hales and Roering, 2007, and Andersen et al. 2015] and possible improvements on them, different measures for precipitation extremes and testing how well they are captured in modern day simulations (by comparison with indices derived from observation based datasets), etc.. However, the format and length of a typical article does not allow thorough investigation of any of these as part of this manuscript and we believe that an arbitrary selection of these efforts would ultimately be very misleading. Instead, we hope to offer these consistently set up simulations as a useful framework for the earth surface science community to build on. We added further descriptions and discussion to convey this.

Because ESurf is not a climate modeling journal, more discussion needs to be given as to the limitations of ECHAM5 in a way that Earth surface process folks can understand. For example, what are the uncertainties associated with simulating orographic precipitation? Though T159 is high-resolution, it still requires substantially smoothing topography, which presumably introduces some uncertainty in to the results. What uncertainties are associated with the PRISM reconstruction? (on a side note, which PRISM reconstruction is used? PRISM3D? PRISM2? I mention this because the topographies between different PRISM reconstructions are substantially different.)

We thank the reviewer for raising this important point. We emphasised the problem of simulating orographic precipitation, and included the recommendation for downscaling where it may be required. We used the PRISM3D reconstruction (as in Haywood et al. 2010), and commented on some of its uncertainties.

I found the use of the cluster analysis to be not intuitively helpful. If the authors want to keep using it, then the authors need to at least walk the reader through an example of how to understand Figures 6, 9, 12, and 15. Is C1 always the same climate zone in each figure and in each time-slice? If so, why are different colors used? Why don't the authors use something more intuitive, like Köppen's climate classification scheme (Peel et al., 2007) to classify climates? As best I understand it, the clustering analysis is used to show the spatial extent of a given climate in a given time-slice and in a given location, but it's not clear how one should interpret these results.

We have modified the text in the results and discussion sections to help clarify this. In short, the clustering analysis essentially fulfills a similar synoptic purpose, but optimises classification and is more fine-tuned to this study's purpose in its selection of variables. This is now reflected in text changes we made to the end of the methods section 2.2. The climate clusters do indeed show the spatial extent of a given climate (described by the mean vectors represented graphically in the raster plots and by numbers in the table included in the supplementary material). The idea is to provide an overview of regional climate without the need to study maps of individual variables, on which these patterns and the climatic homogeneity may not be seen as easily. Each plot represents an optimal classification and thus cluster 1, for example, is not always described by the same mean vector (though usually is usually very similar). The different colours are used to avoid the interpretation that cluster 1, for example, is always characterised by the exact same mean vector. We have included a more elaborate explanation in our revised manuscript to avoid confusion. While readers may be more familiar with the Köppen climate classification scheme, we are more interested in providing an overview not forcefully tied to the categories of this classification scheme. Clustering by various methods (such as this one or PCA) as a synoptic tool are not uncommon (e.g. Paeth 2004, Mannig et al. 2013), but we acknowledge that many readers may not be use to these tools and therefore elaborate explanations.

Minor Comments: Are the topographies for any of these ranges modified at all (it's unclear from the PRISM reference (Line 131), whether this has been done).

Yes, the topographies are different and we now specify the reconstruction we use (PRISM3D) in the methods section.

Lines 57-59: "Cold-temperature island" is not a climatic term in widespread use. What precisely do you mean? Also, Boos and Kuang, 2010 specifically refute the idea that Plateau surface elevation matters for the South Asian Monsoon and rather focus on the Himalaya instead.

Yes, this matters only regionally, but not for the South Asian Monsoon. We have corrected this sentence.

Line 61: Zhisheng et al. (2001) don't actually present any new geological data. Instead, it is all from cited literature. The focus of their study was GCM results. Dettman et al. (2003) is not the only study to look at this. Please see the following studies (which are just a sampling): (Caves et al., 2017; Kent-Corson et al., 2006; Lechler et al., 2013; Lechler and Niemi, 2011; Licht et al., 2016; Methner et al., 2016; Mulch et al., 2015,
2008; Pingel et al., 2016).

Thank you for pointing out this inaccuracy and pointing us to additional studies we ought to list here. We modified the text accordingly.

Lines 77 and 78: "documenting the magnitude" appears twice.
Thank you for pointing this out. We corrected this.

Lines 96-97: Though, importantly, several recent studies have run ECHAM5 at a higher resolution ((Feng et al., 2016; Feng and Poulsen, 2016).
That is true, of course. We mention these in our revised manuscript.

Lines 102-103: This statement is somewhat odd, since the authors are specifically investigating climatological changes over mountain ranges, where resolution typically tends to matter.
Thank you for noticing this apparent inconsistency. We corrected the text accordingly.

Lines 114-133: For all simulations, stating the pCO2 used in the experiment would be most helpful, particularly since it won't take up much room. Also, how is the land-surface treated? For example, the authors state that they are using vegetation reconstructions, but it's unclear if this is then being fed into a "built-in" land-surface model or if they are explicitly using JSBACH.
This is a good point. We included these values in the text. We used the built-in land surface scheme (LSS) and clarified this in the revised text.

Lines 126: "for the" used twice.
This has been corrected.

Lines 203-204: Changes in Greenland and Antarctica are almost certainly unreliable. Because PRISM uses a reconstructed ice-sheet extent, changes in temperature in Greenland and Antarctica are almost certainly reflecting the imposed boundary condition, which itself has quite a bit of uncertainty. It's hard to get around this, except to note that the change in temperature is entirely dependent upon the ice-sheet boundary condition (see discussion in and of de Boer et al. (2015)).
Thank you for pointing this out. We comment on the Pliocene uncertainty in ice sheet reconstructions in the method section of the revised manuscript.

Lines 416-423: Are runoff changes in these models coupled to precipitation changes? In all cases, does P–E (precipitation minus evaporation) scale with changes in precipitation. I'm not particularly familiar with JSBACH (presuming this is the land-surface model used), but if it has a CO2 fertilization parameterization, then runoff may be decoupled from precip. Some of these erosion processes may depend more on runoff than precip.
We use the built in LSS and the runoff is coupled to precipitation. Also, ECHAM5's runoff is not particularly useful in river discharge modelling (see Weiland et al. 2011), which would be of interest in context of erosion. Given this, we refrain from using the model predicted runoff in the global GCM (even though it's conducted at relatively high resolution compared to previous work) to calculate changes in fluvial incision. This would be better done by mapping the predicted precipitation changes onto higher resolution (<90 m) DEMs and solving the kinematic wave equation for each fluvial erosion in each catchment, for the changes in precipitation. However, as we repeatedly mention above, this is not possible to include in this manuscript without first characterising how the precipitation has changed in each region (the current manuscript goals). Work in progress we are conducting is trying to apply the kinematic wave equation and palaeoprecipitation to selected areas, but it's proving difficult to implement meaningfully without temporally continuous (e.g. LGM to present) simulations of precipitation change. We hope this brings to the readers attention the complications associated with doing full erosion history calculations based on these results. We have expanded the last paragraph in the instruction to convey the above perspective better, and more clearly articulate (and justify) the scope and limitations of the manuscript.

Figure 1: Would be nice to also plot the topography of the St. Elias range and the Cascades.
This is a good idea. We included ECHAM5 topographies for Alaska and the Pacific Northwest in Fig. 1 in the revised manuscript.

Figure 7b-Precip-PLIO: Why does precipitation appear to follow a wave-like pattern over tropical South America? Is this due to the spectral nature of ECHAM5?

This may indeed be due to the spectral nature of ECHAM5.

References used in review:

Thank you for being thoughtful enough to provide these. We have added many of them to the revised text.

An, Z., Kutzbach, J.E., Prell, W.L., Porter, S.C., 2001. Evolution of Asian monsoons and phased uplift of the Himalaya-Tibetan plateau since Late Miocene times. Nature 411, 62–66. doi:10.1038/35075035

Caves, J.K., Bayshashov, B.U., Zhamangara, A., Ritch, A.J., Ibarra, D.E., Sjostrom, D.J., Mix, H.T., Winnick, M.J., Chamberlain, C.P., 2017. Late Miocene uplift of the Tian Shan and Altai and reorganization of Central Asia climate. GSA Today 27, 19–26.

Chandan, D., Peltier, W.R., 2017. Regional and global climate for the mid-Pliocene using CCSM4 and PlioMIP2 boundary conditions. Clim. Past 13, 919–942. doi:10.5194/cp-2017-21

de Boer, B., Dolan, A.M., Bernales, J., Gasson, E., Goelzer, H., Golledge, N.R., Sutter, J., Huybrechts, P., Lohmann, G., Rogozhina, I., Abe-Ouchi, A., Saito, F., van de Wal, R.S.W., 2015. Simulating the Antarctic ice sheet in the Late-Pliocene warm period: PLISMIP-ANT, an ice-sheet model intercomparison project. Cryosph. 9, 881–903. doi:10.5194/tc-9-881-2015

Dettman, D.L., Fang, X., Garzione, C.N., Li, J., 2003. Uplift-driven climate change at
Ma: a long δ18O record from the NE margin of the Tibetan plateau. Earth Planet. Sci. Lett. 214, 267–277. doi:10.1016/S0012-821X(03)00383-2

Feng, R., Poulsen, C.J., 2016. Refinement of Eocene lapse rates, fossil-leaf altimetry, and North American Cordilleran surface elevation estimates. Earth Planet. Sci. Lett. doi:10.1016/j.epsl.2015.12.022

Feng, R., Poulsen, C.J., Werner, M., 2016. Tropical circulation intensification and tectonic extension recorded by Neogene terrestrial d18O records of the western United States. Geology 44. doi:10.1130/G38212.1

Haywood, A.M., Hill, D.J., Dolan, A.M., Otto-Bliesner, B.L., Bragg, F., Chan, W.L., Chandler, M.A., Contoux, C., Dowsett, H.J., Jost, A., Kamae, Y., Lohmann, G., Lunt, D.J., Abe-Ouchi, A., Pickering, S.J., Ramstein, G., Rosenbloom, N.A., Salzmann, U., Sohl, L., Stepanek, C., Ueda, H., Yan, Q., Zhang, Z., 2013. Large-scale features of Pliocene climate: Results from the Pliocene Model Intercomparison Project. Clim. Past 9, 191–209. doi:10.5194/cp-9-191-2013

Jiang, D., Lang, X., 2010. Last glacial maximum East Asian monsoon: Results of PMIP simulations. J. Clim. 23, 5030–5038. doi:10.1175/2010JCLI3526.1

Jiang, D., Lang, X., Tian, Z., Ju, L., 2013. Mid-Holocene East Asian summer mon-soon strengthening: Insights from Paleoclimate Modeling Intercomparison Project (PMIP) simulations. Palaeogeogr. Palaeoclimatol. Palaeoecol. 369, 422–429. doi:10.1016/j.palaeo.2012.11.007

Kent-Corson, M.L., Sherman, L.S., Mulch, A., Chamberlain, C.P., 2006. Cenozoic to- pographic and climatic response to changing tectonic boundary conditions in Western North America. Earth Planet. Sci. Lett. 252, 453–466. doi:10.1016/j.epsl.2006.09.049

Lechler, A.R., Niemi, N.A., 2011. Sedimentologic and isotopic constraints on the Pale- ogene paleogeography and paleotopography of the southern Sierra Nevada, California. Geology 39, 379–382. doi:10.1130/G31535.1

Lechler, A.R., Niemi, N. a., Hren, M.T., Lohmann, K.C., 2013. Paleoelevation estimates for the northern and central proto-Basin and Range from carbonate clumped isotope thermometry. Tectonics 32. doi:10.1002/tect.20016

Licht, A., Quade, J., Kowler, A., Santos, M. de los S., Hudson, A., Schauer, A., Hunt- ington, K., Copeland, P., Lawton, T., 2016. Impact of the North American Monsoon on isotope paleoaltimeters: Implications for the paleoaltimetry of the American Southwest. Am. J. Sci. 317.

Methner, K., Fiebig, J., Wacker, U., Umhoefer, P., Chamberlain, C.P., Mulch, A., 2016. Eo-Oligocene proto-Cascades topography revealed by clumped (Δ47) and oxygen isotope (δ18O) geochemistry (Chumstick Basin, WA, USA). Tectonics 35, 546–564. doi:10.1002/2015TC003984

Mulch, A., Chamberlain, C.P., Cosca, M.A., Teyssier, C., Methner, K., Hren, M.T., Gra- ham, S.A., 2015. Rapid change in high-elevation precipitation patterns of western North America during the Middle Eocene Climatic Optimum (MECO). Am. J. Sci. 315,
317–336. doi:10.2475/04.2015.02

Mulch, A., Sarna-Wojcicki, A.M., Perkins, M.E., Chamberlain, C.P., 2008. A Miocene to Pleistocene climate and elevation record of the Sierra Nevada (California). Proc. Natl. Acad. Sci. U. S. A. 105, 6819–6824. doi:10.1073/pnas.0708811105

Peel, M.C., Finlayson, B.L., McMahon, T.A., 2007. Updated world map of the Koppen- Geiger climate classification. Hydrol. Earth Syst. Sci. 11, 1633–1644.

Pingel, H., Mulch, A., Alonso, R.N., Cottle, J., Hynek, S.A., Poletti, J., Rohrmann, A., Schmitt, A.K., Stockli, D.F., Strecker, M.R., 2016. 
[revised manuscript text omitted]

[Figure]

Figure 1

[Figure]

Figure 2

[Figure]

Figure 3

[Figure]

Figure 4

[Figure]

Figure 5

[Figure]

Figure 6

[Figure]

Figure 7

[Figure]

Figure 8

Figure 9

[Figure]

(a)

PI

Annual Temperature (°C)

Annual Precipitation (mm/year)

(b)

MH

LGM

PLIO

colder than PI          warmer than PI

Degrees Celsius (C°)

drier than PI          wetter than PI

mm/year

▦ ice cover

Figure 10

Mean Annual Precipitation a

Mean Annual Temperature b

Figure 11

[Figure]

Figure 12

[Figure]

Figure 13

[Figure]

Figure 14

[Figure]

Figure 15

[Figure]

Figure 16

[Figure]

Figure 17

[Figure]

Supplemental Figure S1

---

## Author Response (AR3)

**Response to RC1 (1st revision)**

(responses in blue)

Mutz et al. look to use paleoclimate GCMs to identify drivers of past geomorphic change. This is a topic for which I hold great interest, and I feel that the authors have crafted a very useful set of model results that they leave underutilized. As such, I feel inclined to accept the paper on the basis of the useful results, but to request major revisions such that they do their own work justice.

My major concerns, which will become clear in the line-by-line comments (please feel free to respond to similar comments en masse) are as follows: 1. The paper is motivated by denudation rates and landscape evolution, but really includes this as a speculative wrapper that is not substantiated. I suggest that instead you propose testable hypotheses surrounding your findings.
2. Related to #1, much of the text is a litany of "temperature was X here ...". I find such statements of results useful only insofar as they expand upon a figure (associated with a supplementary data set) that presents the results. While these sections are written clearly, I would suggest that the authors focus on a set of geomorphic questions (if this be their motivation) and how the model-data set informs those questions.
3. Many of the discussions of model results are of ice-covered regions, yet no consideration of direct glacial erosion is given. Furthermore, no reference to the changes in the statistics of discharge or catchment area in ice-covered regions is given. This seems a disservice to this relatively high-resolution paleoclimate AGCM: the geologic setting *must* be considered, otherwise it seems that the authors' pushing on the modeling end has not been matched by a simple geological history sanity check. I would suggest that either significantly glaciated regions and the catchments that they feed be masked out, or that glacial erosion and its associated processes be included in the discussion.
4. (Discussed only here): You have not compared your models against any data. I understand that this may be simply a modeling exercise that you do compare to other models. However, I think that such a comparison could assuage skepticism about your results and lend support to your case, especially if you include it as part of a local case study (see the third point below). The core of these three points is that, with a bit more care, I think your results could say something really useful to the geomorphic community. Currently, the paper seems to be more a statement of, "this is important to geomorphology", followed by a long list of the model results. I challenge the authors to demonstrate (rather than simply stating) the importance of their work to geomorphology in a way that includes how it may impact the way scientists view Quaternary landscape evolution. Ideas include:
• Changes in means (done)
• Changes in statistical distributions of temperature and precipitation – think extreme events, frost-cracking window, etc.
• A focus on a few iconic regions while *explicitly* ignoring significantly ice-covered domains (I think this would be easiest, though obviously would be thrilled if you decided to tackle glacial processes)
• Using this focus to build a template for how to use paleoclimate GCM outputs to advance the field of geomorphology. Currently, I think that the work is acceptable following changes for internal consistency and geological accuracy (see #1 and #3), but I think that you could be selling yourselves short if you don't dig just a tiny bit deeper to investigate your forcings and their impact on geomorphology. I hope that you find these comments helpful in continuing to craft an insightful piece of work out of what seems to be a strong modeling approach.

We thank Prof. Andrew Wickert for his highly valuable review of our manuscript. Many important points were raised in the review and we hope that our appreciation for the input is sufficiently reflected in the revisions we made in response to it. We also encourage him to see our response to the second reviewer, where we provide additional geologic relevance of this study by now including a comparison of available terrestrial proxy data to our model results. We also explain throughout our response to the 2nd reviewer why an application of the predicted climate change to predict denudation rate changes is a large undertaking that can not be meaningfully conducted in this paper, but warrants more detailed applications of the models to individual areas (a topic of ongoing work/application for us). We refrain from using the model predicted runoff in the global GCM (even though it's conducted at relatively high resolution compared to previous work) to calculate changes in fluvial incision.  This would be better done by mapping the predicted precipitation changes onto higher resolution (<90 m) DEMs and solving the kinematic wave equation for each fluvial erosion in each catchment, for the changes in precipitation. However, as we repeatedly mention above, this is not possible to include in this
manuscript without first characterising how the precipitation has changed in each region (the
current manuscript goals). Work in progress we are conducting is trying to apply the kinematic
wave equation and palaeoprecipitation to selected areas, but it's proving difficult to implement
meaningfully without temporally continuous (e.g. LGM to present) simulations of precipitation
change. We hope this brings to the reviewers attention the complications associated with doing
full erosion history calculations based on these results. We have expanded the last paragraph in
the instruction to convey the above perspective better, and more clearly articulate (and justify)
the scope and limitations of the manuscript.

We appreciate the importance of addressing specific sets of geomorphic questions and hypotheses (1 & 2)
and we are currently taking an in-depth look at quantifying the potential for erosion by a variety of
processes. These include different methods of quantifying frost cracking intensity and extreme
precipitation events and how these changed over time. However, in order to include those in this
manuscript, we fear that would have to seriously compromise the thoroughness with which we investigate
these questions at the moment. Instead, we hope that we can convey the usefulness of our consistently set
up palaeoclimate simulations as a framework for addressing any of these particular questions in detail, and
have modified sections of this manuscript accordingly. This includes, but is not restricted to, extensive
compilations of proxy-based precipitation reconstructions for our two larger study sites (South Asia and
western South America) and comparison of this data to our model output. With this, we hope that we were
also able to address the concerns raised in point 4. In order to address the important point raised about
glaciated areas (3), we added an ice cover layer on all of our difference plots, included global maps of ice
extent (as used for our simulations) in the supplementary material, and discussed where the large
differences in temperature and precipitation we highlight in the manuscript are accompanied by changes in
ice cover. Thus we hope to prevent that interpretations of the implications of our results are made without
consideration of changes in ice cover (and consequently shifts in the process domain).

Line-by-line:
23. US Pacific Northwest Pacific→drop second "Pacific"
This has been corrected. Thank you for catching that.

29. future observational studies interested in quantifying→future observational studies that quantify
(studies can't be interested in things, strictly speaking)
That is right of course. It has been corrected as suggested.

53. orogen scale → orogen-scale
It has been corrected as suggested.

~57. A couple of recent studies from the climate science community shed light on the impacts of the Andes
(first ref below) and continents in general (second ref below). In case these are interesting to you, I'm
pasting the bibliographic information here:
Maroon, E. A., D. M. W. Frierson, and D. S. Battisti (2015), The tropical precipitation response to Andes
topography and ocean heat fluxes in an aquaplanet model, J. Clim., 28(1), 381–398, doi:10.1175/JCLI-D-
14-00188.1.
Maroon, E. A., D. M. W. Frierson, S. M. Kang, and J. Scheff (2016), The precip itation response to an
idealized subtropical continent, J. Clim., 29(12), 4543–4564, doi:10.1175/JCLI-D-15-0616.1.
Many thanks for the references. These are indeed of much interest to us and we included them in the
revised manuscript.

73-75. "Furthermore, recent controversy exists concerning the spatial and temporal scales over which
geologic and geochemical observations can record climate-driven changes in weathering and erosion [e.g.
Whipple, 2009; von Blanckenburg et al., 2015; Braun, 2016].": I see that you do not return to this point
later, so could you describe the controversy for those who are not familiar with it?
Thank you for pointing this out. We described the controversy briefly for those unfamiliar with it after the
sentence quoted above.

81. I see that later you discuss a little about what an AOGCM may do, but I will be looking for justification about how an AGCM may suffice. Is this in part because you prescribe the b.c.'s and you are running it for
17 years only? If so, could you discuss potential systematic variations between this and an AOGCM?
We prescribe sea surface temperature reconstructions (SSTs) boundary conditions, which allows us to
bypass the computationally expensive coupled simulations. Because these are fixed climatologies (though
with seasonality preserved), the simulation of fewer years suffices. As a consequence, however, we do not
expect to see decadal scale variability as we would in case of coupled models or prescribed SSTs that vary
from year to year (such as present day simulations using AMIP SST's). We discuss this in the revised
manuscript.
89. "PLIO to the Last Glacial Maximum": as you include no time-slices between these, I suggest making
these part of the list and dropping the "to the".
This has been corrected as suggested.
147. "This section describes the clustering method used in this study." You could drop this sentence – the
section title should be enough for even an inattentive reader!
We followed this suggestion and dropped that sentence.
176-178. I was wondeirng how you picked the number of clusters: I am glad to see that you performed a
thorough search.
Thank you. We added some text explaining this. We systematically increased the number of clusters from 3
to 10 and assessed the distinctiveness or similarities of resulting climate clusters. Once the increase in the
number of cluster no longer resulted in the addition of another cluster that was distinctly different from the
others, we used this as a cut off point and used the cluster number of the previous iteration as the optimal
cluster number.
Section 3: Much of this is information that I find better communicated through figures than with text. It is
clearly written, however, and I am reluctant to suggest a rewrite for brevity in a length-unconstrained
journal so long as the text can be co-located with the figure.
190-192. I see you have another "This section describes..." sentence. If this is your preferred way to write,
you may keep it; here, the second sentence is not such a good topic-sentence replacement.
Thank you. We kept the sentence in this instance as it also immediately draws attention to the relevant
figures, which may also serve in addressing the previous point you raised.
197-198. i.e. over the ice sheets. (This applies to other regions as well, and should be important to point
out if you are going to then discuss fluvial processes in orogens)
Many thanks for pointing this out. We are more mindful of this in the revised discussion.
203-paragraph: Also because of local ice loss, presumably. So I think that the two prior paragraphs could
have a new summary that "The greatest changes in temperature is observed where the greatest change in
local ice extent occurs."
Thank you. We followed your suggestion.
214-215. Have you considered discussions of the African Humid Period?
We had not considered discussion of precipitation changes in North Africa, since it lies outside the regions
we focus on. However, we appreciate that Holocene precipitation changes in the region are important and
may be of interest to many readers. We therefore included a short discussion of Holocene precipitation
changes in Northern Africa in the revised manuscript.
373-374. If you are looking at the influence of temperature and precipitation on erosion, and you are not
including subglacial erosion, then your preceding text must indicate where your changes really are
indicative of ice extent – both as a separate process domain and as a driver of fluvial processes and
potential changes in the statistics of river discharge.
Thank you. We are more mindful of this in our discussion.
Section 4.1. Your first paragraph (weathering) differs from the content (comparing your model results with
those published). These should be in different subsections, and the weathering paragraph may need to be
expanded. Your "weathering and erosion" paragraph also neglects direct effects of glaciers, ice caps, and
ice sheets, which were globallly significant.

Thank you for this suggestion. We re-structured this section as suggested and took ice extent into
consideration in the erosion section.
Section 4.2. Once again, your discussion is often of formerly (or currently) ice-covered regions without
explicitly acknowledging that this is a different process domain. In addition, as with the previous section,
the body paragraphs are mostly about model comparison and regional changes with sparse link to the
landscape-evolution factors indicated in the topic paragraph.
As above, we restructured this section, took ice extent into account and chose a more fitting section title.
416-423. Please discuss the direct influence of glaciers on the erosion orogens in the context of changing
precipitation (and therefore mass balance). Is it significant or not?
Although it is challenging to sufficiently quantify changes in glacier-related erosion due to differences in
precipitation, we now include this point in our discussion.
433-434. "Coastal North America"? Doesn't look like it: seems to be most of NA south of the ice margin.
That is correct. We revised our descriptions accordingly.
Section 4.3. The authors describe the results here, but I find the connection to erosion rates to be
insufficiently described compared to how they are highlighted in the topic sentence, as well as in the
abstract. I would like you to go one step beyond "ought to be considered" and actually posit how you
expect the erosion rates – and therefore, the balance between erosion and exhumation and perhaps the
equilibrium shapes of the mountains and their rivers – to vary. Otherwise, you are suggesting future work
rather than actually describing the possible geomorphic significance – and I think underutilizing your
results in a paper that is clearly targeted towards geomorphologists.
As described in our response above, we believe that trying to address specific problems such as these or
quantifying how differences would be expressed as erosion rates would be beyond the scope of this
manuscript and come at the cost of not being able to address these as thoroughly as we are currently
attempting in other ongoing work.
498. "which may favour frost driven weathering during glacial climate states" – the St. Elias range was
covered by glaciers! Yes, there can be some frost-cracking around the ice, but don't you think this is
important too? http://instaar.colorado.edu/groups/QGISL/ak_paleoglacier_atlas/gallery/index.html
Thank you for drawing our attention to this. We consider this in our discussion now and revised the
manuscript accordingly.
508. "enhanced sediment production driven by frost processes" – same as above. Glaciers were there.
Consider them.
As for the comment above, we also considered glaciers here in the revised manuscript.
Conclusions: Comparison to other models: is this match surprising or no? Did you (mostly) use the same
inputs and simply increase the grid resolution? If so, could you comment on how the improved grid and
possible variations in inputs and use of them ocean as a boundary condition may have affected (or not)
your results as compared to those of earlier studies? This would be more useful to include in the discussion
than a simple list of "Our temperature in place Y was T0 , and X et al. wrote that they found it was T1 ,
which is close to T0 . Think big-picture, in both process and numerics!
Due to model-specific parameterisation, deviation is possible. In the revised manuscript, we comment on
this as well as on the model resolution and implications of using ocean as boundary conditions instead of
an ocean model.
533. Did your 8-10 degC changes occur significantly over areas that would be affected by hillslope or
fluvial processes? (i.e. unglaciated areas?)
Some unglaciated areas experience large differences in temperature, but the maxima of 8-10 degC
geographically coincide with ice cover changes. We acknowledge and discuss this in the revised
manuscript.

**Response to RC2 (1ˢᵗ revision)**

(responses in blue)

The authors simulate the pre-industrial, mid-Holocene, LGM, and Pliocene climates using the ECHAM5 general circulation model. The motivation is to understand how past climatic states may change regional climatologies, particularly over mountainous areas that have been the focus of much erosion and geomorphic work. The authors find that past climatic states (particularly the LGM and Pliocene) do produce changes in absolute temperature and precipitation and in the annual ranges of these two climatic variables. Overall, I did not find the manuscript to be a particularly useful addition to the literature. Though the motivation is potentially novel, the analysis is not complete and needs substantially more work. Most of the manuscript focuses on simply describing climatic changes, while neglecting novel analyses. Consequently, at this point, I recommend reject, though perhaps with substantial work (including more modeling, comparison with existing models, and/or comparison with data), it may become publishable in the future in Esurf.

We thank the reviewer for their constructive feedback on our manuscript. We found the reviewer's comment very useful in the improvement of this manuscript and hope that the modifications we made do the reviewer's thorough review justice. Most importantly, we followed the reviewer's suggestion to deepen the comparison of our model simulation results with data, specifically proxy-based reconstructions, as we believe this suggestion is of most relevance to the earth surface science community targeted by ESurf. While it was not possible to create a complete overview of all proxy-based studies everywhere and for all time periods, we included an extensive compilation of site specific reconstructions of precipitation for our largest areas (South America and Tibet). In prioritizing the compilation of precipitation reconstructions, we also hope to address the reviewer's concerns about the GCM's performance regarding this variable, and are happy to report that the model shows satisfactory to good performance in predicting the direction of changes in MH and LGM.

First, the authors need to decide what the point of the paper is. Most of the paper reads as a description of climatological changes for 3 periods in the geologic past (MH, LGM, and Plio) (indeed, most of the text is written this way). However, this work has already been done, most prominently by the PMIP and PlioMIP set of model intercomparisons. What does this manuscript offer that these model intercomparisons have not already analyzed? A case could be made that these model intercomparisons are typically of a global nature (though there has been some work on changes in the Asian Monsoon systems using both PlioMIP and PMIP results (Jiang et al., 2013; Jiang and Lang,2010; Zhang et al., 2013)), so that the analysis of the orogens in this study is useful. However, given that GCMs have difficulty simulating precipitation and in particular simulating precipitation over complex topography, the usefulness of simply describing changes over the Himalaya, Andes, Cascades, and St. Elias ranges is somewhat muted. For example, why should readers believe that ECHAM5 produces reasonable results over the St. Elias range? Why not use (or at least compare with) the exist- ing model intercomparisons to look at changes in these locations? Most of the PlioMIP simulations are of a lower resolution than the model simulations presented here, though not all (Haywood et al., 2013). Many of the newer PlioMIP2 simulations are being run at a higher resolution and permit at least some comparison with the data here (Chandan and Peltier, 2017). Having myself tried to access PlioMIP data, I understand that it can be difficult to get access to the PlioMIP output, but if the point of the paper is to quantify actual climatological change, then comparison with other models is a must (or at least a thorough treatment of possible boundary condition uncertainties and additional ECHAM5 model runs to establish the sensitivity of precipitation/temperature in these areas); otherwise, we have no reason to believe that ECHAM5 presents anything resembling a proper picture of climatic change in the past.

**Summary Response**: We appreciate this assessment and hope to have enhanced the usefulness of the manuscript with modifications we've made throughout the results and discussion sections. While GCM simulations for the time slices we chose already exist and individual studies were also conducted at a relatively high resolution, we believe (contrary to the reviewer's suggestion) that the usefulness of our results lies in both the high resolution combined with the consistency in model choice, resolution, output frequency (and methods of descriptive statistics). We emphasise this point in the revised manuscript. We acknowledge in the text (methods section 2.1 and introduction) the shortcomings of GCMs in predicting orographic precipitation in the discussion and in the revised text now compare the simulated precipitation with readily available proxy-based reconstructions for specific locations in Tibet and South America where we were able to compile data (this was a large undertaking). The revised manuscript now demonstrates the shortcomings of GCMs and provides an additional data compilation that ESurf readers interested in palaeoclimate effects on denudation may find useful. The changes we have made to address this reviewers suggestions are contained in the revised introduction, and extensively throughout the Discussion section (4.0), as well as the addition of two new (concluding figures).

**In More Detail:** While we appreciated the reviewer's comments, we note that what he/she is asking for in the above comments is a manuscript that either focuses on an inter-model comparisons, or a comparison to proxy data. We see merits in both, but both are not possible in a single manuscript. An inter-model comparison is not really well suited to the aims of this journal and our target community to bring attention to the magnitude of palaeoclimate changes that have occurred in different active orogens. Our intended audience is not the palaeoclimate modelling community (and journals associated with it), but rather the surface processes community. We specifically chose this journal to provide this community with palaeoclimate predictions that may be of interest to them. In our own experience in trying to interpret palaeodenudation rates from data we produce, our first goal is always trying to find predictions or observations of climate change for a region. Thus, we set out to write this paper to provide what we find in our research as the first, most useful, step in understanding the surface process history of a region. Furthermore, one major aspect of this study that we want to retain is the statistical analysis of the climatology of each time slice as this is what is most useful for the ESurf community in terms of knowing if a region they are working is has experienced a signal change in climate through time.

Thus, we are left with the conundrum that what the review requests and we want would normally be three full manuscripts that include: 1) a statistical analysis of climatological changes over active orogens (our previous focus of the text), or 2) a comparison of a suite of similarly set up and common code (ECHAM5) simulations of palaeoclimate time slices to other GCM models (one of the suggestions by this reviewer), or 3) a model-data comparisons for model evaluation at the time slices investigated (also suggested by this reviewer).

*In an attempt to hopefully reach a compromise with this reviewer – the revised manuscript is now structured around points 1 and 3 above (statistical analysis of climatological changes, and model-data comparison where possible).* These changes have been manifested in a now expanded discussion section, and through the addition of model-data comparison figures at the end of the text (see section 4.1.2 , 4.2.2, and 4.4). These changes hopefully reach a happy middle-ground between our aims and the reviewer's suggestions. The changes have significantly expanded the manuscript, and we hope that the reviewer also recognizes that implementation of all the suggested changes is not possible within a single manuscript of typical length for this journal. The revised manuscript now has 18 figures (including 1 in the supplemental section) and 21 pages of manuscript text (not including references / captions).

Some parts of the paper address actual causes for climatic change (for example, discussion of the Pacific North American Teleconnection (no citation given). Is this related to the PDO? Lines 493-495). Again, if this is the primary point of the paper, then substantially more work needs to be done to address why precipitation, for example, increases in the Himalaya in the Pliocene. If this was the point of the paper, it would obviate the need to compare with other model simulations (see paragraph above), but would then require substantially more work to identify how various atmospheric phe- nomenon change through time. A generalized description of changes in the past is not particularly useful (most of this information can be communicated fairly effectively with figures), so addressing the causes of these changes (or comparison with prox- iesâ˘T˘ see below) is worthwhile.

Parts of the discussion showcase comparisons with terrestrial proxy data, though this is limited to citing previous work and stating that there is general agreement with previous, proxy-model work. If a proxy-model comparison is the point of the paper, then more work needs to be done actually compiling the proxies and doing a proper statistical test to see if there is agreement between modelled precip/temperature changes in each of these orogens and existing data. This would be a useful contribution to the literature, but, again, would require substantially more work.

We address the suggestions in the previous 2 paragraphs in the following ways: **First,** as was mentioned earlier, we maintain that for this journal and the community that reads it there is value in demonstrating climate change events in commonly studied active orogens where denudation studies (sensitive to climate change) are conducted. This requires that we maintain our current statistical characterisations of the climate change. With this approach, it then becomes intractable to focus on the climate dynamics associated with change in each region because typically a thorough investigation of the climate dynamics associate with change in each region could be a paper for each region in itself, and the broader picture of change in a range
would be missed. We try to meet the reviewer half way in this suggestion by including additonal comments
on possible causes for climatic change in different regions.
The PNA (citation now included) is related to the PDO, and Alaskan precipitation and temperature is to
some degree controlled by it. In order to properly address these questions, however, further analyses looking
at dominant atmospheric variability modes, trajectories and other aspects of atmosphere would have to be
carried out, which are beyond the scope of this study. We have modified the manuscript in text at the end of
section 2.1 to clarify (and justify) why we do not conduct an inter model comparison, as well as some of the
caveats associated with our approach.
However, (**second**), we like the reviewer's suggestions of adding additional comparisons to proxy data, and
we heavily invested time in compiling this information in this revised version for locations where spatially
distributed data are available for the time slices available.
Unfortunately, changes from proxy studies are often reported in terms of relative changes compared to
modern (e.g., wetter/drier, warmer/colder) and are sometimes contradictory for the same location, making
the application of otherwise suitable statistical tests (e.g. t-test and even non-parametric tests) difficult.
There are, in some cases (see final figures in paper), even contradictions between the proxy data themselves
in neighbouring locations, a reality often underappreciated by the surface processes community.
Furthermore, terrestrial proxy data are not available for all the time slices and locations we investigate. This
is particularly true for the Pliocene time slice. **With these limitations of available proxies, we do the best**
**we can to compare available observations to proxy data. These comparisons are now provided in**
**revisions to the discussion section, and the addition of 2 figures at the end of the manuscript**.
We note that for the Alaska and Cascadia (NW USA) study areas, which are heavily and repeatedly
glaciated, that limited - to no proxy data are available (due to poor preservation of proxies over glacial-
interglacial cycles). Although some marine proxy records and records of past glaciations have been
published for these regions, they are not useful for our purposes because they do not record if terrestrial
locations in the regions were wetter/drier or warmer/colder during our time slices. Marine proxy records are
of limited use for comparison to the terrestrial changes investigated in this study. Thus, despite the lack of
terrestrial proxy comparisons available for two of the regions (Alaska, Cascadia), we maintain that
presentation of the model predicted changes for these data poor regions is extremely useful because the
model predictions augment existing terrestrial data gaps, and provide a starting point for future studies to
formulate testable hypotheses of climate change and potential denudation impacts.
Much of the motivation for the manuscript appears to be to understand how climatic changes may change
denudation/geomorphic analyses, but this is done in only a superficial way. If this is the point of the paper,
then, again, much more work needs to be done, rather than simply stating that erosion depends on climate
(lines 416-423 are a good example of statements that serve to motivate a paper, but don't provide any
actual analysis). For example, can the authors take some of the climate model output and, given a
potential 2000 mm/yr change in precipitation in the Himalaya since the Pliocene, actually re-interpret some
of the existing exhumation/denudation data? If not, why not? What additional data is necessary? And if such a
reanalysis isn't possible, then how does knowledge of such a change in precipitation facilitate future work?
This is a great suggestion and we are indeed doing this as ongoing work. But what the reviewer is asking
for here simply cannot be included in this paper without making it an extremely long manuscript. For
example, in the surface processes community, comparisons of river profiles to modern precipitation, or
temperature changes to frost cracking histories for **one** catchment / study area (e.g. <100x100 km2) is
typically a full manuscript in itself (e.g. Marhsall et al., 2015 Science Advances; Schaller et al. 2004
Journal of Geology). So, we respectfully disagree with the reviewer that this should be done within this
manuscript. It's simply not possible to do in a way that would be convincing to the surface processes
community.
I agree with the generalized statements made throughout the paper (i.e., that denudation and landscape
evolution depend on climate), but these are somewhat self-evident and, as currently written, the manuscript
does not make a fundamental contribution to improving our knowledge on this subject except to state that
climate changed in the past. Assumptions of stationarity are indeed a problem in interpreting modern datasets
that have a component of geologic history, but a really nice contribution of this paper would be to show how
these assumptions can be mitigated when one knows the history of precipitation or temperature.

We agree with the reviewer that it is "self-evident" that palaeoclimate can impact the denudation history of
an orogen, but how many palaeodenudation rate studies actually make a comparison to spatial distributed
predictions of palaeoclimate? Very few in our experience, and we cite the robust ones in the manuscript.
What most palaeodenudation rate studies do is compare some set of observation to the nearest set of proxy
observations, but this approach does not provide a sense of spatial variability in terrestrial climate change
that models predict (furthermore, proxy observations are often far removed from the denudation rate
observations). In our manuscript, we specifically avoid picking on previous publications that follow this
'conventional' approach to show how this commonly used approach ignores spatial variability in climate
change. We don't think this is a productive way to advance the science. Instead, we maintain that providing
spatially continuous model predictions at different time slices in a self-consistent set of simulations is
extremely useful for documenting the magnitude of climate change (which is not self-evident), and for
formulating testable hypotheses for future work and to identify where the best locations are in the world to
investigate climate change impacts on a regions denudation history. Thus, the way in which our research
group usually conducts palaeodenudation rate studies is to first run model simulations to formulate testable
hypotheses, and then to investigate/test these hypotheses with field studies and geochemical observations of
denudation histories. This approach to investigating palaeoclimate-palaeodenudation interactions is
definitely not self-evident in the literature in our opinion.

We very much agree that in order for our simulations to ultimately be useful in actually quantifying
denudation much more work needs to be done. Translating any of the changes observed here (or by other
studies) to erosion rates remains a big problem in the Earth surface science community. It is one that we are
actively working on and includes the application and comparison of different models for quantifying frost
cracking [Anderson et al., 1998, Hales and Roering, 2007, and Andersen et al. 2015] and possible
improvements on them, different measures for precipitation extremes and testing how well they are captured
in modern day simulations (by comparison with indices derived from observation based datasets), etc..
However, the format and length of a typical article does not allow thorough investigation of any of these as
part of this manuscript and we believe that an arbitrary selection of these efforts would ultimately be very
misleading. Instead, we hope to offer these consistently set up simulations as a useful framework for the
earth surface science community to build on. We added further descriptions and discussion to convey this.

Because ESurf is not a climate modeling journal, more discussion needs to be given as to the limitations of
ECHAM5 in a way that Earth surface process folks can understand. For example, what are the
uncertainties associated with simulating orographic precipitation? Though T159 is high-resolution, it still
requires substantially smoothing topography, which presumably introduces some uncertainty in to the results.
What uncertainties are associated with the PRISM reconstruction? (on a side note, which PRISM
reconstruction is used? PRISM3D? PRISM2? I mention this because the topographies between different
PRISM reconstructions are substantially different.)

We thank the reviewer for raising this important point. We emphasised the problem of simulating orographic
precipitation, and included the recommendation for downscaling where it may be required. We used the
PRISM3D reconstruction (as in Haywood et al. 2010), and commented on some of its uncertainties.

I found the use of the cluster analysis to be not intuitively helpful. If the authors want to keep using it, then the
authors need to at least walk the reader through an example of how to understand Figures 6, 9, 12, and 15.
Is C1 always the same climate zone in each figure and in each time-slice? If so, why are different colors
used? Why don't the authors use something more intuitive, like Köppen's climate classification scheme (Peel
et al., 2007) to classify climates? As best I understand it, the clustering analysis is used to show the spatial
extent of a given climate in a given time-slice and in a given location, but it's not clear how one should
interpret these results.

We have modified the text in the results and discussion sections to help clarify this. In short, the clustering
analysis essentially fulfills a similar synoptic purpose, but optimises classification and is more fine-tuned to
this study's purpose in its selection of variables. This is now reflected in text changes we made to the end of
the methods section 2.2. The climate clusters do indeed show the spatial extent of a given climate (described
by the mean vectors represented graphically in the raster plots and by numbers in the table included in the
supplementary material). The idea is to provide an overview of regional climate without the need to study
maps of individual variables, on which these patterns and the climatic homogeneity may not be seen as
easily. Each plot represents an optimal classification and thus cluster 1, for example, is not always described
by the same mean vector (though usually is usually very similar). The different colours are used to avoid the interpretation that cluster 1, for example, is always characterised by the exact same mean vector. We have included a more elaborate explanation in our revised manuscript to avoid confusion. While readers may be more familiar with the Köppen climate classification scheme, we are more interested in providing an overview not forcefully tied to the categories of this classification scheme. Clustering by various methods (such as this one or PCA) as a synoptic tool are not uncommon (e.g. Paeth 2004, Mannig et al. 2013), but we acknowledge that many readers may not be use to these tools and therefore elaborate explanations.

Minor Comments: Are the topographies for any of these ranges modified at all (it's unclear from the PRISM reference (Line 131), whether this has been done).

Yes, the topographies are different and we now specify the reconstruction we use (PRISM3D) in the methods section.

Lines 57-59: "Cold-temperature island" is not a climatic term in widespread use. What precisely do you mean? Also, Boos and Kuang, 2010 specifically refute the idea that Plateau surface elevation matters for the South Asian Monsoon and rather focus on the Himalaya instead.

Yes, this matters only regionally, but not for the South Asian Monsoon. We have corrected this sentence.

Line 61: Zhisheng et al. (2001) don't actually present any new geological data. Instead, it is all from cited literature. The focus of their study was GCM results. Dettman et al. (2003) is not the only study to look at this. Please see the following studies (which are just a sampling): (Caves et al., 2017; Kent-Corson et al., 2006; Lechler et al., 2013; Lechler and Niemi, 2011; Licht et al., 2016; Methner et al., 2016; Mulch et al., 2015, 2008; Pingel et al., 2016).

Thank you for pointing out this inaccuracy and pointing us to additional studies we ought to list here. We modified the text accordingly.

Lines 77 and 78: "documenting the magnitude" appears twice.

Thank you for pointing this out. We corrected this.

Lines 96-97: Though, importantly, several recent studies have run ECHAM5 at a higher resolution ((Feng et al., 2016; Feng and Poulsen, 2016).

That is true, of course. We mention these in our revised manuscript.

Lines 102-103: This statement is somewhat odd, since the authors are specifically investigating climatological changes over mountain ranges, where resolution typically tends to matter.

Thank you for noticing this apparent inconsistency. We corrected the text accordingly.

Lines 114-133: For all simulations, stating the pCO2 used in the experiment would be most helpful, particularly since it won't take up much room. Also, how is the land-surface treated? For example, the authors state that they are using vegetation reconstructions, but it's unclear if this is then being fed into a "built-in" land-surface model or if they are explicitly using JSBACH.

This is a good point. We included these values in the text. We used the built-in land surface scheme (LSS) and clarified this in the revised text.

Lines 126: "for the" used twice.

This has been corrected.

Lines 203-204: Changes in Greenland and Antarctica are almost certainly unreliable. Because PRISM uses a reconstructed ice-sheet extent, changes in temperature in Greenland and Antarctica are almost certainly reflecting the imposed boundary condition, which itself has quite a bit of uncertainty. It's hard to get around this, except to note that the change in temperature is entirely dependent upon the ice-sheet boundary condition (see discussion in and of de Boer et al. (2015)).

Thank you for pointing this out. We comment on the Pliocene uncertainty in ice sheet reconstructions in the method section of the revised manuscript.

Lines 416-423: Are runoff changes in these models coupled to precipitation changes? In all cases, does P–E (precipitation minus evaporation) scale with changes in precipitation. I'm not particularly familiar with JSBACH (presuming this is the land-surface model used), but if it has a CO2 fertilization parameterization, then runoff may be decoupled from precip. Some of these erosion processes may depend more on runoff than precip.

We use the built in LSS and the runoff is coupled to precipitation. Also, ECHAM5's runoff is not particularly useful in river discharge modelling (see Weiland et al. 2011), which would be of interest in context of erosion. Given this, we refrain from using the model predicted runoff in the global GCM (even though it's conducted at relatively high resolution compared to previous work) to calculate changes in fluvial incision. This would be better done by mapping the predicted precipitation simulation changes onto higher resolution (<90 m) DEMs and solving the kinematic wave equation for each fluvial erosion in each catchment, for the changes in precipitation. However, as we repeatedly mention above, this is not possible to include in this manuscript without first characterising how the precipitation has changed in each region (the current manuscript goals). Work in progress we are conducting is trying to apply the kinematic wave equation and palaeoprecipitation to selected areas, but it's proving difficult to implement meaningfully without temporally continuous (e.g. LGM to present) simulations of precipitation change. We hope this brings to the readers attention the complications associated with doing full erosion history calculations based on these results. We have expanded the last paragraph in the instruction to convey the above perspective better, and more clearly articulate (and justify) the scope and limitations of the manuscript.

Figure 1: Would be nice to also plot the topography of the St. Elias range and the Cascades.

This is a good idea. We included ECHAM5 topographies for Alaska and the Pacific Northwest in Fig. 1 in the revised manuscript.

Figure 7b-Precip-PLIO: Why does precipitation appear to follow a wave-like pattern over tropical South America? Is this due to the spectral nature of ECHAM5?

This may indeed be due to the spectral nature of ECHAM5.

References used in review:

Thank you for being thoughtful enough to provide these. We have added many of them to the revised text.

An, Z., Kutzbach, J.E., Prell, W.L., Porter, S.C., 2001. Evolution of Asian monsoons and phased uplift of the Himalaya-Tibetan plateau since Late Miocene times. Nature 411, 62–66. doi:10.1038/35075035

Caves, J.K., Bayshashov, B.U., Zhamangara, A., Ritch, A.J., Ibarra, D.E., Sjostrom, D.J., Mix, H.T., Winnick, M.J., Chamberlain, C.P., 2017. Late Miocene uplift of the Tian Shan and Altai and reorganization of Central Asia climate. GSA Today 27, 19–26.

Chandan, D., Peltier, W.R., 2017. Regional and global climate for the mid-Pliocene using CCSM4 and PlioMIP2 boundary conditions. Clim. Past 13, 919–942. doi:10.5194/cp-2017-21

de Boer, B., Dolan, A.M., Bernales, J., Gasson, E., Goelzer, H., Golledge, N.R., Sutter, J., Huybrechts, P., Lohmann, G., Rogozhina, I., Abe-Ouchi, A., Saito, F., van de Wal, R.S.W., 2015. Simulating the Antarctic ice sheet in the Late-Pliocene warm period: PLISMIP-ANT, an ice-sheet model intercomparison project. Cryosph. 9, 881–903. doi:10.5194/tc-9-881-2015

Dettman, D.L., Fang, X., Garzione, C.N., Li, J., 2003. Uplift-driven climate change at 12 Ma: a long δ18O record from the NE margin of the Tibetan plateau. Earth Planet. Sci. Lett. 214, 267–277. doi:10.1016/S0012-821X(03)00383-2

Feng, R., Poulsen, C.J., 2016. Refinement of Eocene lapse rates, fossil-leaf altimetry, and North American Cordilleran surface elevation estimates. Earth Planet. Sci. Lett. doi:10.1016/j.epsl.2015.12.022

Feng, R., Poulsen, C.J., Werner, M., 2016. Tropical circulation intensification and tectonic extension recorded by Neogene terrestrial d18O records of the western United States. Geology 44. doi:10.1130/G38212.1

Haywood, A.M., Hill, D.J., Dolan, A.M., Otto-Bliesner, B.L., Bragg, F., Chan, W.L., Chandler, M.A., Contoux, C., Dowsett, H.J., Jost, A., Kamae, Y., Lohmann, G., Lunt, D.J., Abe-Ouchi, A., Pickering, S.J., Ramstein, G., Rosenbloom, N.A., Salzmann, U., Sohl, L., Stepanek, C., Ueda, H., Yan, Q., Zhang, Z., 2013. Large-scale features of Pliocene climate: Results from the Pliocene Model Intercomparison Project. Clim. Past 9, 191–209. doi:10.5194/cp-9-191-2013

Jiang, D., Lang, X., 2010. Last glacial maximum East Asian monsoon: Results of PMIP simulations. J. Clim. 23, 5030–5038. doi:10.1175/2010JCLI3526.1

Jiang, D., Lang, X., Tian, Z., Ju, L., 2013. Mid-Holocene East Asian summer mon-soon strengthening: Insights from Paleoclimate Modeling Intercomparison Project (PMIP) simulations. Palaeogeogr. Palaeoclimatol. Palaeoecol. 369, 422–429. doi:10.1016/j.palaeo.2012.11.007

Kent-Corson, M.L., Sherman, L.S., Mulch, A., Chamberlain, C.P., 2006. Cenozoic to-pographic and
climatic response to changing tectonic boundary conditions in Western North America. Earth Planet. Sci.
Lett. 252, 453–466. doi:10.1016/j.epsl.2006.09.049

Lechler, A.R., Niemi, N.A., 2011. Sedimentologic and isotopic constraints on the Pale-ogene
paleogeography and paleotopography of the southern Sierra Nevada, California. Geology 39, 379–382.
doi:10.1130/G31535.1

Lechler, A.R., Niemi, N. a., Hren, M.T., Lohmann, K.C., 2013. Paleoelevation estimates for the northern and
central proto-Basin and Range from carbonate clumped isotope thermometry. Tectonics 32.
doi:10.1002/tect.20016

Licht, A., Quade, J., Kowler, A., Santos, M. de los S., Hudson, A., Schauer, A., Hunt-ington, K., Copeland,
P., Lawton, T., 2016. Impact of the North American Monsoon on isotope paleoaltimeters: Implications for the
paleoaltimetry of the American Southwest. Am. J. Sci. 317.

Methner, K., Fiebig, J., Wacker, U., Umhoefer, P., Chamberlain, C.P., Mulch, A., 2016. Eo-Oligocene proto-
Cascades topography revealed by clumped ($\Delta$47) and oxygen isotope ($\delta$18O) geochemistry (Chumstick
Basin, WA, USA). Tectonics 35, 546–564. doi:10.1002/2015TC003984

Mulch, A., Chamberlain, C.P., Cosca, M.A., Teyssier, C., Methner, K., Hren, M.T., Gra-ham, S.A., 2015.
Rapid change in high-elevation precipitation patterns of western North America during the Middle Eocene
Climatic Optimum (MECO). Am. J. Sci. 315,
317–336. doi:10.2475/04.2015.02

Mulch, A., Sarna-Wojcicki, A.M., Perkins, M.E., Chamberlain, C.P., 2008. A Miocene to Pleistocene climate
and elevation record of the Sierra Nevada (California). Proc. Natl. Acad. Sci. U. S. A. 105, 6819–6824.
doi:10.1073/pnas.0708811105

Peel, M.C., Finlayson, B.L., McMahon, T.A., 2007. Updated world map of the Koppen-Geiger climate
classification. Hydrol. Earth Syst. Sci. 11, 1633–1644.

Pingel, H., Mulch, A., Alonso, R.N., Cottle, J., Hynek, S.A., Poletti, J., Rohrmann, A., Schmitt, A.K., Stockli,
D.F., Strecker, M.R., 2016. Surface uplift and convective rainfall along the southern Central Andes
(Angastaco Basin, Argentina). Earth Planet. Sci. Lett. 440, 33–42. doi:10.1016/j.epsl.2016.02.009

Zhang, R., Yan, Q., Zhang, Z.S., Jiang, D., Otto-Bliesner, B.L., Haywood, a. M., Hill, D.J., Dolan, a. M.,
Stepanek, C., Lohmann, G., Contoux, C., Bragg, F., Chan, W.-L.,

Chandler, M.a., Jost, a., Kamae, Y., Abe-Ouchi, a., Ramstein, G., Rosenbloom, N. a., Sohl, L., Ueda, H.,
2013. East Asian monsoon climate simulated in the PlioMIP. Clim. Past Discuss. 9,1135-1164.
doi:10.5194/cpd-9-1135-2013

**Response to RC1 (2nd revision)**

(responses in blue)

The revised manuscript by Mutz et al. is a clear improvement over its predecessor, but does not sufficiently address substantial comments that I made on the last version of the manuscript. I have given more thought to specific suggestions on how to rectify this. Some of these issues are major, and I have therefore suggested "Major Revisions", though I believe that Mutz et al. may be able to complete them in short order. I will, therefore, also keep this short and to the point. Once these major issues are resolved, I will be happy to review the revised manuscript thoroughly.

The primary items of concern are:
1. A title that is not representative of the article content
2. A lack of clarity from the abstract and introduction on what the authors actually do.

We thank Andrew Wickert for the review of the revised manuscript, and for the concise and clear communication of concerns and suggestions. We hope that we managed to address these concerns adequately. Below, we describe how we addressed these concerns.

In short, the authors must communicate up-front something along the lines of:
"Motivated by the need to better understand climate impacts on the denudation of orogens, we model paleoclimate at 4 time slices, and qualitatively compare how changes in temperature and precipitation may impact fluvial and/or hillslope erosion."

After the description of the scientific background to this study in the introduction, we included a slightly modified version of the suggested sentence to begin to describe our work. We modified the sentence to be as precise as possible in our communication of what we attempt in the study (line 85): *"Motivated by the need to better understand climate impacts on Earth surface processes, especially the denudation of orogens, we model palaeoclimate for four time slices in the Late Cenozoic, use descriptive statistics to identify the extent of different regional climates, quantify changes in temperature and precipitation, and discuss the potential impacts on fluvial and/or hillslope erosion."*

Title:
"Where is Late Cenozoic climate change most likely to impact denudation?"

The authors do not answer this question. There are two issues to address here. The former can be addressed in the text. The latter will require a change in scope or title.
1. The text does not answer this question in regards to fluvial and hillslope erosion. The discussion provides some ideas on how changes in P and T may impact erosion rates in different regions, but offer no concrete proposals or answers.
2. The authors state that glacial erosion is beyond the scope of their study. There is no way to discuss late Cenozoic erosion (let alone climate change and erosion) without considering glacial erosion.

I would suggest a title that relates more closely to the authors' work, as a modeling study intended to provide environmental conditions that may impact fluvial and/or hillslope erosion. Such a title will be less broad in scope, but may be more powerful in precision.

We agree with Andrew Wickert and modified the title to more precisely describe our work. We have changed the title to *"Estimates of Late Cenozoic climate change relevant to Earth surface processes in tectonically active orogens"*. We included "at active orogens" to better communicate the regional focus of our work. Since the intention of this study is also to provide a GCM-simulation framework for studies investigating a variety of surface processes, we decided to add "relevant to Earth surface processes". This way, the title does not promise to answer questions pertaining to all types of erosional processes, but instead explains that the work presented here, specifically the simulation themselves, provides a basis for more in-depth studies of a variety of surface processes. For example, since our original submission of the manuscript, the GCM simulation framework has been used to investigate vegetation response to differences in palaeoclimate (Werner et al., in review).
We also modified part of the introduction (line 111) in order to not create false expectations, and communicate more clearly what is to be expected in the manuscript.

Abstract:
1. Do you discuss vegetation gradients in the results of your work? In other words, is it worth including here? Otherwise, the abstract is clear.

While we belief our simulations are of benefit to the vegetation modelling community, as is demonstrated by the study mentioned above, we do not discuss this much in our manuscript. We therefore omitted "vegetation gradients" in the abstract.

Introduction:
It is not clear in the introduction how you will link your model results to landscape evolution. Making clear the focus on the model and its inspiration (landscape evolution), along with the limited scope of the actual application to landscape evolution you provide here, will help the readers see the paper for what it is.

We now clarify in the introduction (line 102) that our GCM output may directly be used as boundary conditions for vegetation and landscape evolution models, such as LPJ-GUESS and Landlab respectively, to bridge the gap between palaeoclimate change and quantitative estimates for Earth surface system responses.

One specific point: you mention glacial erosion in the introduction as being important, but then note in line 434 that you will not address it. Up until this point, the reader may reasonably think that you are going to discuss glacial erosion, as it is a dominant process in the late Cenozoic.

We now clarify in the introduction (line 109) that merited discussion of glacial erosion is beyond the scope of our study to avoid readers looking for such discussion in our manuscript.

References:
Werner, C., Schmid, M., Ehlers, T. A., Fuentes-Espoz, J. P., Steinkamp, J., Forrest, M., Liakka, J., Maldonado, A., and Hickler, T.: Effect of changing vegetation on denudation (part 1): Predicted vegetation composition and cover over the last 21 thousand years along the Coastal Cordillera of Chile, Earth Surf. Dynam. Discuss., https://doi.org/10.5194/esurf-2018-14, in review, 2018.

**List of Major Changes (1$^{st}$ revision)**

Overall, we found the reviewer's comments refreshingly useful. We greatly appreciate the time the editors have taken to find thoughtful reviewers, and the reviewers time in providing constructive comments. We were able to implement most of the suggested changes. However, some of the requested changes by reviewer 2 would require multiple manuscripts, and in our response to this reviewers comments we explain a (hopefully) happy middle ground between their requests and the reality of what we think can be realistically implemented in a manuscript for the ESurf community of readers. **The most extensive changes to the manuscript are in the introduction and discussion sections, as well as in the modification of previously presented figures (Fig. 1, 2, 3, 7, 10, 13)** - since modification of these sections and figures most directly addresses the reviewers concerns and they did not identify errors in our results required new simulations.

The general concern that resonated in both reviewers comments was the need to focus the manuscript more and add 'meat' to our arguments in one direction or another. We have taken this to heart, and **the revised manuscript presents an in-depth (and very time consuming) compilation of terrestrial proxy data observations for two of the regions (South Asia, South America)** which were not extensively glaciated and have good records available. These proxy data were compared to the model results and provide a new and focused dimension to the manuscript. **As a results of these efforts, we included two additional figures (Fig. 16, 17)**. As detailed in our response to reviewer 2 - we do not present a detailed inter-model comparison, although we have expanded our comparison to previous published work throughout the text (mostly in the discussion section). We also provide additional justification for our approach, and highlight more explicitly what is novel about our presentation of a series of experiments conducted with the same model at the same resolution.

The marked-up manuscript below highlights changes made to the manuscript in more detail.

**List of Major Changes (2$^{nd}$ revision)**

In accordance with suggestions made by the reviewer, major changes have been made to the title and parts of the introduction. Minor changes have been made to the abstract and acknowledgements.

[revised manuscript text omitted]